# DAVIS: OOD DETECTION VIA DOMINANT ACTIVATIONS AND VARIANCE FOR INCREASED SEPARATION

## ABSTRACT

Detecting out-of-distribution (OOD) inputs is a critical safeguard for deploying machine learning models in the real world. However, most post-hoc detection methods operate on penultimate feature representations derived from global average pooling (GAP) – a lossy operation that discards valuable distributional statistics from activation maps prior to global average pooling. We contend that these overlooked statistics, particularly channel-wise variance and dominant (maximum) activations, are highly discriminative for OOD detection. We introduce DAVIS, a simple and broadly applicable post-hoc technique that enriches feature vectors by incorporating these crucial statistics, directly addressing the information loss from GAP. Extensive evaluations show DAVIS sets a new benchmark across diverse architectures, including ResNet, DenseNet, and EfficientNet. It achieves significant reductions in the false positive rate (FPR95), with improvements of 48.26% on CIFAR-10 using ResNet-18, 38.13% on CIFAR-100 using ResNet-34, and 26.83% on ImageNet-1k benchmarks using MobileNet-v2. Our analysis reveals the underlying mechanism for this improvement, providing a principled basis for moving beyond the mean in OOD detection. Our code is available here: https://github.com/epsilon-2007/DAVIS

## 1 INTRODUCTION

Safe deployment of ML models in the open world hinges on a critical capability: recognizing and gracefully handling inputs that fall outside their training distribution. When faced with such out-of-distribution (OOD) data – samples from novel contexts or unknown classes – a robust model should signal its uncertainty rather than making a confident, and likely incorrect, prediction (Hendrycks & Gimpel, 2017; Nguyen et al., 2015; Hein et al., 2019). The ability to reliably detect OOD inputs is thus paramount for safety-critical systems, from medical diagnosis (Wang et al., 2017; Roy et al., 2021) to autonomous driving (Filos et al., 2020).

To address this challenge, researchers have developed a wide array of OOD detection techniques. While some approaches modify the model's training objective (Jeong & Kim, 2020; Ming et al., 2023; Ghosal et al., 2024), a particularly prominent line of work involves *post-hoc* methods, which do not require costly retraining. The resulting research has focused on scoring functions or modifying the penultimate layer features of a pre-trained network to distinguish between in-distribution (ID) and OOD samples (Hendrycks & Gimpel, 2017; Liu et al., 2020; Djurisic et al., 2023; Xu et al., 2024). These approaches share a unexamined, fundamental limitation: they operate on a feature vector produced by *global average pooling (GAP)*. While effective for classification, GAP is inherently a lossy operation as it summarizes each channel's entire spatial activation map – a rich distribution of spatial responses – into a single value. In doing so, it permanently discards crucial information cues about spread and peak intensity of activation distribution, which we contend are powerful signals for identifying anomalies.

Our work is motivated by the observation that the full raw distributions of pre-pooled activations contain highly discriminative OOD signals that are ignored by current post-hoc methods. As we show in Figure 1,

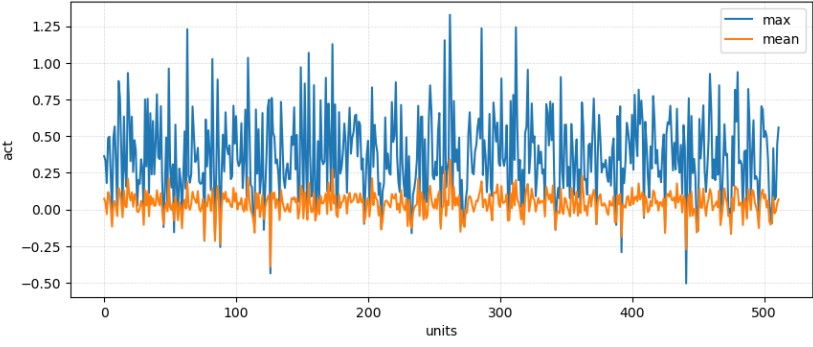

Figure 1: *Dominant activations provide a stronger OOD signal than mean activations. The plot shows the average activation gap between ID (CIFAR-10) and OOD (Texture) samples for each unit in the penultimate layer of a pre-trained ResNet-18. The gap derived from the dominant (maximum) activation (blue) is consistently and significantly larger than the gap from the standard mean activation (orange).*

the separation between ID and OOD samples is dramatically amplified when using statistics such as the dominant (maximum) activation instead of the mean (i.e., GAP). This increased separation directly translates to more reliable OOD scoring, as shown in Figure 2, where the distributions for ID and OOD data become more distinguishable via the decreased overlap between the two distributions. Based on this insight, we propose DAVIS *(Dominant Activations and Variance for Increased Separation)*, a simple, post-hoc plug and play method that directly counteracts the information loss of GAP. DAVIS enriches the penultimate feature vector by incorporating key overlooked statistics – namely the channel-wise maximum and variance. The primary strength of DAVIS lies in its synergistic versatility. Rather than replacing existing techniques, DAVIS complements standard baselines to unlock significant, additional performance.

In summary, our contributions are as follows:

1. We identify the information loss from global average pooling as a key, yet overlooked, weakness in post-hoc OOD detection. To address this, we propose DAVIS, a simple plug-and-play module that can be easily integrated into existing pipelines to enrich feature representations by extracting complementary, discriminative pre-pooling statistics.

2. We establish a new performance benchmark across a range of OOD benchmarks and diverse modern architectures, including ResNet, DenseNet, MobileNet-V2, and EfficientNet. DAVIS significantly reduces the FPR95, achieving relative improvements of up to 48% on CIFAR-10 using ResNet-18, 38% on the CIFAR-100 using ResNet-34, and 27% on ImageNet-1k using MobileNet-v2.

3. We provide a statistical analysis, grounded in empirical evidence, that reveals the underlying mechanism of DAVIS (see Appendix B). This provides a principled justification for moving beyond the mean and leveraging richer distributional statistics for robust OOD detection.

## 2 RELATED WORK

Existing work in OOD detection can be broadly grouped into three paradigms: post-hoc methods that operate on pre-trained classifiers, training-time regularization techniques that modify the learning objective, and generative models that learn the ID density. Our work, DAVIS, belongs to the post-hoc paradigm due to its composability with off-the-shelf models. In this section, we review each of the three paradigms.

**Post-Hoc Methods and the Reliance on Averaged Features.** The first focuses on designing effective scoring functions. While early methods like MSP (Hendrycks & Gimpel, 2017) were susceptible to overcon-

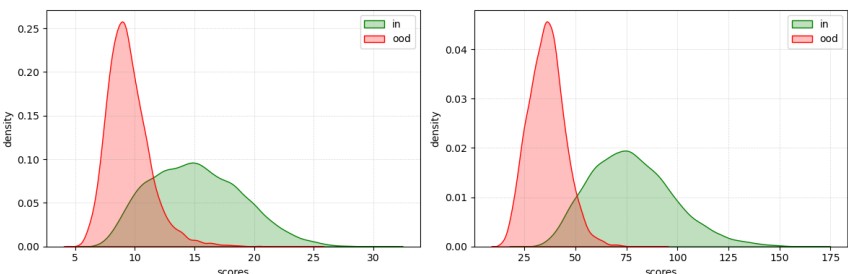

Figure 2: *Using dominant activations improves OOD score separation. Left: OOD scores based on standard mean activations show significant overlap between the ID (CIFAR-10) and OOD (Texture) distributions, leading to poor separability. Right: Leveraging dominant (maximum) activations shifts the OOD score distribution away from the ID scores. Both plots show energy scores from a ResNet-18.*

fidence (Nguyen et al., 2015), the field advanced significantly with two distinct paradigms. One approach introduced more robust logit-based signals, such as the energy score (Liu et al., 2020). A second approach developed feature-space methods, which use distances in the embedding space to detect outliers. Prominent examples include scores based on the Mahalanobis distance (Lee et al., 2018) and knn (Sun et al., 2022).

A second, more recent wave of methods seeks to improve these scores by directly modifying the penultimate layer feature vector. Techniques such as ReAct (Sun et al., 2021), DICE (Sun & Li, 2022), ASH (Djurisic et al., 2023), and SCALE (Xu et al., 2024) have shown that rectifying, sparsifying, or pruning these activations can significantly enhance the separation between ID and OOD score distributions. However, these methods all operate on a feature vector produced by GAP, which discards a wealth of discriminative information as discussed above. Our work directly addresses the information loss at its source, enriching this feature vector before any subsequent scoring or modification is applied.

**Training-Time Regularization.** This line of research focuses on improving OOD detection by modifying the model's training objective Ming et al. (2023); Ghosal et al. (2024). The resulting methods often incorporate an auxiliary dataset of outliers (Papadopoulos et al., 2021) or introduce regularization terms that encourage the model to produce less confident predictions on OOD samples (Hendrycks et al., 2019; Du et al., 2022; Ming et al., 2022; Huang et al., 2022). While effective, these approaches require a more complex training process and access to relevant outlier data, which may not always be available.

**Generative Models.** An alternative paradigm for OOD detection uses generative models to estimate the density of the ID data (Dinh et al., 2017; Choi & Jang, 2019; Kirichenko et al., 2020; Schirrmeister et al., 2020; Xiao et al., 2020; Ghosal et al., 2024; Liu & Qin, 2024). The intuition is that OOD samples will lie in low-density regions of the learned data manifold. However, it has been shown that deep generative models can paradoxically assign high likelihoods to structurally simple OOD samples (Nalisnick et al., 2019), making them less reliable than discriminative approaches for OOD detection. DAVIS builds on the strong empirical performance and reliability of discriminative classifiers.

## 3 METHOD

This section formally presents DAVIS. The core empirical insight is that typically discarded cues – specifically, peak intensity and variance – carry powerful discriminative cues between ID and OOD samples. By incorporating these properties, which are lost by simple averaging, DAVIS creates a more discriminative feature representation for separating ID from OOD samples, as illustrated in Figure 2.

### 3.1 DOMINANT ACTIVATIONS AND VARIANCE FOR INCREASED SEPARATION

DAVIS is a post-hoc module designed to counteract the information loss from GAP. We create a more discriminative feature representation by extracting richer statistics from the pre-pooling activation maps, before they are averaged. To formalize this, we consider a standard supervised classification setting. Let $\mathcal{X}$ denote the input space and $\mathcal{Y} = \{1, 2, \cdots, C\}$ the output label space for neural network $\theta$ trained on dataset $\mathcal{D}_{\text{in}}$.

For input $\mathbf{x}$, the network produces a set of n spatial activation maps $g(\mathbf{x}) \in \mathbb{R}^{n \times k \times k}$. The conventional penultimate layer feature vector, $h(\mathbf{x}) \in \mathbb{R}^n$, is obtained via GAP: $h(\mathbf{x}) = \text{Avg}(g(\mathbf{x}))$. DAVIS replaces or augments this vector by computing more descriptive statistics from each of the $n$ channels. For each channel, we compute the mean $\mu(\mathbf{x})$, the maximum (dominant) value $m(\mathbf{x})$, and the standard deviation $\sigma(\mathbf{x})$. We propose two primary formulations of the DAVIS feature vector, $h^{\text{DAVIS}}(\mathbf{x})$:[1]

1. DAVIS($m$) replaces the feature vector $h(\mathbf{x})$ by *dominant activation* $m(\mathbf{x})$ as follows:
$$h^{\text{DAVIS}(m)}(\mathbf{x}) = m(\mathbf{x}) \tag{1}$$

2. DAVIS($\mu, \sigma$) augments the mean activation with its corresponding channel-wise standard deviation $\sigma(\mathbf{x})$, scaled by a hyperparameter $\gamma$ as shown in Equation 2. We discuss a detailed hyperparameter selection process in Section 4.5 and Appendix E.
$$h^{\text{DAVIS}(\mu, \sigma)}(\mathbf{x}) = \mu(\mathbf{x}) + \gamma\sigma(\mathbf{x}) \tag{2}$$

This new feature vector, $h^{\text{DAVIS}}(\mathbf{x})$, is then passed through the original fully-connected layer (with weights $\mathbf{W}$ and bias $\mathbf{b}$) to produce a new set of logits as shown in Equation 3. The procedure described above modifies the activation level of the feature vector and aims to increase the separation between ID and OOD samples. Also, note that DAVIS can be used to complement existing downstream techniques such as Energy Liu et al. (2020), ReAct Sun et al. (2021), DICE Sun & Li (2022), ASH Djurisic et al. (2023), and SCALE Xu et al. (2024). We provide a description of these techniques in Appendix A.

$$f^{\text{DAVIS}}(\mathbf{x}) = \mathbf{W}^\top h^{\text{DAVIS}}(\mathbf{x}) + \mathbf{b} \tag{3}$$

### 3.2 OOD DETECTION WITH DAVIS

The goal of OOD detection is to learn a decision boundary $G_\lambda(\mathbf{x}; \theta)$ that classifies a test sample $\mathbf{x} \in \mathcal{X}$:

$$G_\lambda(\mathbf{x}; \theta) = \begin{cases} \text{ID} & \text{if } \mathbf{x} \sim \mathcal{D}_{\text{in}} \\ \text{OOD} & \text{if } \mathbf{x} \sim \mathcal{D}_{\text{out}} \end{cases} = \begin{cases} \text{ID} & \text{if } S(\mathbf{x}; \theta) \geq \lambda \\ \text{OOD} & \text{if } S(\mathbf{x}; \theta) < \lambda \end{cases} \tag{4}$$

where $S(\mathbf{x}; \theta)$ represents a downstream OOD scoring function, and by convention (Liu et al., 2020) $\lambda$ is a threshold calibrated such that 95% of ID data ($\mathcal{D}_{\text{in}}$) is correctly classified.

At inference time, the enriched logits from Equation 3 are used with a standard scoring function to perform OOD detection. While DAVIS can be seamlessly integrated with any downstream OOD scoring function that directly or indirectly uses the penultimate layer to derive the scores, we focus on the logit-based energy score $S_{\text{energy}}(\mathbf{x}; \theta)$, shown in Equation 5, due to its prevalence and superior performance. We also evaluate our methods using MSP (Hendrycks & Gimpel, 2017) and ODIN score (Liang et al., 2018) in Sections 4 and 6.

$$S_\theta(\mathbf{x}) = -\mathbf{E}_\theta(\mathbf{x}) = \log\left(\sum_{j=1}^{C} \exp^{(f_j^{\text{DAVIS}}(\mathbf{x};\theta))}\right) \tag{5}$$

Since our method complements existing techniques mentioned above, we evaluate DAVIS in conjunction with these approaches in Section 6. Subsequently, we formally characterize and explain why DAVIS improves the separability of the scores between ID and OOD data as part of detailed study in Appendix B.

---

[1]We use "DAVIS" to denote both versions. The context of our discussion will make clear any salient differences.

## 4 EMPIRICAL EVALUATION OF DAVIS

We conduct a comprehensive empirical evaluation to validate the efficacy of DAVIS. Our experimental evaluation provides a comprehensive assessment of our method, testing its performance on standard benchmarks, its scalability to large-scale datasets, its robustness across diverse architectures, and its behavior in challenging near-OOD scenarios. Throughout our evaluation, we use the energy score as the default scoring function for all methods. For brevity, a method such as DAVIS($m$) should be interpreted as DAVIS($m$) + Energy.

A core principle of our evaluation is to provide a fair and direct comparison against prior work. As many of the architectures used in our evaluation (e.g., ResNet-18, ResNet-34, and MobileNet-v2 for the CIFAR benchmarks; DenseNet-121 and EfficientNet-b0 for ImageNet) were not included in the original publications of foundational baselines like ReAct, DICE, ASH, and SCALE, we undertook a rigorous re-evaluation of these methods. In all cases, we carefully followed the official hyperparameter selection protocols and open-sourced implementations from their respective papers to ensure the integrity of our comparisons.

**Evaluation metrics.** In line with standard evaluation protocol in OOD detection (Liu et al., 2020), we evaluate the performance of our approach, DAVIS, using two key metrics: False Positive Rate (FPR95) and Area Under the ROC Curve (AUROC):

1. *FPR95* is the percentage of OOD samples incorrectly classified as ID when the threshold $\lambda$ is set to correctly classify 95% of ID samples. A lower value is better ($\downarrow$).

2. *AUROC* represents the probability that a random ID sample is assigned a higher score than a random OOD sample. A value of 1.0 is perfect, while 0.5 is random (a higher value is better, $\uparrow$).

### 4.1 CIFAR EVALUATION

Table 1: *OOD detection results on CIFAR benchmarks. All values are percentages and are averaged over six OOD test datasets. The full results for each evaluation dataset are provided in Appendix C. The symbol $\downarrow$ indicates lower values are better; $\uparrow$ indicates higher values are better. *In MobileNet-v2 on CIFAR-100, ASH is used instead of DICE as the combined method.*

| Dataset | Method | ResNet-18 | | ResNet-34 | | DenseNet-101 | | MobileNet-v2 | |
|---|---|---|---|---|---|---|---|---|---|
| | | FPR95 $\downarrow$ | AUROC $\uparrow$ | FPR95 $\downarrow$ | AUROC $\uparrow$ | FPR95 $\downarrow$ | AUROC $\uparrow$ | FPR95 $\downarrow$ | AUROC $\uparrow$ |
| CIFAR-10 | MSP | 58.43 | 91.23 | 54.86 | 91.96 | 45.36 | 92.43 | 59.88 | 90.82 |
| | ODIN | 28.98 | 95.16 | 23.06 | 95.53 | 19.37 | 96.06 | 35.25 | 93.37 |
| | Energy | 35.61 | 94.14 | 26.04 | 95.29 | 22.54 | 95.42 | 28.33 | 94.89 |
| | ReAct | 30.14 | 95.15 | 26.36 | 95.37 | 17.21 | 96.59 | 27.80 | 94.84 |
| | DICE | 30.92 | 94.69 | 23.00 | 95.84 | 14.51 | 96.74 | 22.67 | 95.78 |
| | ReAct+DICE | 20.22 | 96.40 | 21.75 | 96.17 | 10.32 | 97.94 | 22.21 | 95.88 |
| | ASH | 21.83 | 96.02 | 19.57 | 96.34 | 14.88 | 96.90 | 22.69 | 95.59 |
| | DAVIS($m$) | 18.21 | 96.69 | 17.12 | 96.81 | 18.96 | 96.41 | 29.22 | 94.59 |
| | DAVIS($m$) + DICE | **10.46** | **97.94** | **10.67** | **97.97** | 8.33 | 98.30 | **18.10** | **96.44** |
| | DAVIS($\mu,\sigma$) | 21.09 | 96.40 | 18.30 | 96.69 | 16.43 | 96.91 | 32.85 | 94.20 |
| | DAVIS($\mu,\sigma$) + DICE | 13.49 | 97.54 | 12.09 | 97.75 | **8.32** | **98.32** | 24.38 | 95.31 |
| CIFAR-100 | MSP | 80.40 | 76.16 | 79.68 | 78.08 | 78.04 | 73.96 | 83.83 | 72.78 |
| | ODIN | 66.06 | 84.78 | 67.50 | 84.71 | 57.67 | 84.00 | 70.10 | 83.63 |
| | Energy | 70.86 | 83.18 | 70.30 | 83.64 | 60.59 | 83.29 | 72.65 | 82.77 |
| | ReAct | 59.43 | 87.52 | 57.87 | 86.64 | 56.74 | 86.74 | 53.57 | 87.90 |
| | DICE | 56.90 | 85.39 | 55.17 | 86.28 | 44.91 | 87.40 | 64.78 | 82.93 |
| | ReAct+DICE | 50.44 | 87.94 | 58.83 | 83.16 | 38.90 | 90.64 | 53.77 | 83.59 |
| | ASH | 54.50 | 87.47 | 54.81 | 87.88 | 36.66 | 90.67 | 51.65 | 86.37 |
| | DAVIS($m$) | 53.32 | 89.41 | 52.26 | 89.54 | 44.55 | 89.98 | 55.28 | 87.04 |
| | DAVIS($m$) + DICE* | **33.38** | **92.51** | **33.91** | **92.33** | 38.32 | 91.74 | 46.35* | 86.96* |
| | DAVIS($\mu,\sigma$) | 56.72 | 88.36 | 55.26 | 88.78 | 44.97 | 89.88 | 55.05 | 87.23 |
| | DAVIS($\mu,\sigma$) + DICE* | 36.19 | 91.54 | 36.67 | 91.81 | **33.04** | **92.32** | **46.32*** | **87.23*** |

**Experimental Setup.** We evaluate on the widely-used CIFAR datasets (Krizhevsky et al., 2009). Following standard protocols (Liu et al., 2020; Sun et al., 2021; Djurisic et al., 2023), we use six common OOD datasets for evaluation: Textures (Cimpoi et al., 2014), SVHN (Netzer et al., 2011), Places365 (Zhou et al., 2017), LSUN-Crop (Yu et al., 2015), LSUN-Resize (Yu et al., 2015), and iSUN (Xu et al., 2015). To ensure fair comparison with prior work, we use a DenseNet-101 backbone (Huang et al., 2017). To demonstrate architectural generality, we extend our evaluation to ResNet-18, ResNet-34 (He et al., 2016), and MobileNet-v2 (Sandler et al., 2018). Model and training details are detailed in Appendix E.

**Results.** Table 1 summarizes our results on the CIFAR benchmarks. The table clearly shows the two key benefits of our method: (1) DAVIS used standalone (e.g., rows DAVIS($m$)) significantly outperforms the standard energy score baseline, proving its inherent value. (2) When composed with existing methods like DICE or ASH, DAVIS establishes a new benchmark. For instance, on CIFAR-10, DAVIS($m$) + DICE reduces FPR95 by 48.27%, 45.47%, 19.27%, and 18.50% with ResNet-18, ResNet-34, DenseNet-101, and MobileNet-v2, respectively.

On CIFAR-100, DAVIS($m$) + DICE reduces FPR95 by 38.82% and 38.14% with ResNet-18 and ResNet-34, while DAVIS($\mu, \sigma$) + DICE achieves a 9.88% reduction with DenseNet-101, and DAVIS($\mu, \sigma$) + ASH achieves a 10.32% reduction with MobileNet-v2. The detailed evaluation is provided in Appendix C.

## 4.2 IMAGENET EVALUATION

Table 2: *OOD detection results on ImageNet benchmarks. All values are percentages, averaged over four common OOD benchmark datasets. The full results for each evaluation dataset are provided in Appendix C. The symbol ↓ indicates lower values are better; ↑ indicates higher values are better. *In MobileNet-v2, ReAct+DICE is used instead of ASH as the combined method.*

| Method | DenseNet-121 | | ResNet-50 | | MobileNet-v2 | | EfficientNet-b0 | |
|---|---|---|---|---|---|---|---|---|
| | FPR95 ↓ | AUROC ↑ | FPR95 ↓ | AUROC ↑ | FPR95 ↓ | AUROC ↑ | FPR95 ↓ | AUROC ↑ |
| MSP | 63.46 | 82.65 | 65.06 | 82.75 | 70.42 | 80.64 | 67.72 | 81.92 |
| ODIN | 49.45 | 87.48 | 56.48 | 85.41 | 54.20 | 85.81 | 68.60 | 78.02 |
| Energy | 50.68 | 87.60 | 58.41 | 86.17 | 59.03 | 86.56 | 81.49 | 74.87 |
| ReAct | 40.98 | 91.01 | 31.43 | 92.95 | 48.94 | 88.74 | 60.08 | 85.21 |
| DICE | 38.67 | 89.65 | 34.75 | 90.78 | 41.93 | 89.60 | 97.70 | 45.79 |
| ReAct+DICE | 45.70 | 87.60 | 28.39 | 93.37 | 31.01 | 92.85 | 87.09 | 61.32 |
| ASH | 30.25 | 93.09 | 22.80 | 95.12 | 38.78 | 90.94 | 98.84 | 55.42 |
| DAVIS($m$) | 42.30 | 89.34 | 43.44 | 88.10 | 45.84 | 89.11 | 57.00 | 82.85 |
| DAVIS($m$) + ASH* | 30.80 | 92.97 | 24.36 | 94.74 | 31.61* | 90.34* | 56.32 | 82.84 |
| DAVIS($\mu, \sigma$) | 40.62 | 90.38 | 43.60 | 89.33 | 46.25 | 89.41 | 48.40 | 87.39 |
| DAVIS($\mu, \sigma$) + ASH* | **28.44** | **93.66** | **22.32** | **95.16** | **24.45*** | **93.65*** | **47.90** | **87.56** |

**Experimental Setup.** To assess scalability and performance in a more realistic, large-scale setting, we evaluate on the ImageNet-1k benchmark. We use four challenging OOD datasets: iNaturalist (Van Horn et al., 2018), SUN (Xiao et al., 2010), Places365 (Zhou et al., 2017), and Textures (Cimpoi et al., 2014). These datasets are carefully curated to avoid class overlap with ImageNet, while spanning distinct semantic domains to rigorously assess generalization performance (Liu et al., 2020; Sun et al., 2021). Our evaluation showcases broad architectural robustness by using pre-trained DenseNet-121, ResNet-50, MobileNet-v2, and EfficientNet-b0, for which we re-evaluated all baselines to ensure a fair comparison.

**Results.** As shown in Table 2, DAVIS delivers consistent improvements on the ImageNet scale. In direct comparison with energy score, which allows us to see the direct benefit of using DAVIS under the same scoring function. DAVIS($\mu, \sigma$) reduces the FPR95 by 19.85%, 25.35%, 21.65%, and 40.60% using DenseNet-121, ResNet-50, MobileNet-v2, and EfficientNet-b0 architectures respectively.

The most significant gains are achieved when composing `DAVIS` with existing foundational methods like ASH. Specifically, `DAVIS`$(\mu, \sigma)$ combined with `ASH` improves FPR95 by 5.98%, 2.11%, 20.27% compared to previous best results using DenseNet-121, ResNet-50, and EfficientNet-b0 respectively. Using MobileNet-v2, `DAVIS`$(\mu, \sigma)$ combined with `ReAct+DICE` achieves 26.83% improvement in FPR95. These results validate that the principles of `DAVIS` are effective in complex, large-scale scenarios and across diverse architectural families. The detailed evaluation is provided in Appendix C.

As pointed out in Sun et al. (2021); Sun & Li (2022); Djurisic et al. (2023), Mahalanobis (Lee et al., 2018) exhibits limiting performance, while being computationally expensive due to estimating the inverse of the covariance matrix. For this reason, we omit their results from this paper.

### 4.3 COMPARISON WITH OTHER BASELINES

To situate `DAVIS` against the contemporary methods, we provide a detailed comparison in Appendix D against three other methods: NCI (Liu & Qin, 2025), fDBD (Liu & Qin, 2024), and SCALE (Xu et al., 2024). The results confirm the superiority of our approach, particularly when used as a synergistic module. Against NCI's reported results, our combined method reduces the average FPR95 by a significant 43.90% on CIFAR-10 and 43.30% on ImageNet. This advantage is even more pronounced against fDBD, where our method achieves a substantial FPR95 reduction of 56.40% on ImageNet.

Our comprehensive evaluation of the SCALE baseline, extending its analysis beyond the limited architectures reported in its original publication, and found that our method, `DAVIS`, provides a significant and consistent performance boost. For instance, it reduces the average FPR95 of SCALE by to 36.84% on CIFAR-10 using ResNet-18, 22.07% on CIFAR-100 using ResNet-18, and 7.44% on ImageNet using DenseNet-121. More critically, on ImageNet, our method rescues the SCALE baseline on EfficientNet-b0 from a near-total failure (98.56% FPR95), achieving a functional 61.50% FPR95. This demonstrates the powerful synergistic effect of our approach and its ability to complement the robustness of other methods. A detailed breakdown of this comparison is provided in Appendix D.3.

### 4.4 NEAR-OOD EVALUATION

We further evaluate `DAVIS` on the challenging near-OOD detection task of separating CIFAR-10 (ID) from CIFAR-100 (OOD) (Ghosal et al., 2024). On this difficult benchmark, our method provides consistent improvements over all baselines baselines across every tested architecture. For instance, on a ResNet-18, `DAVIS` reduces the FPR95 by 7.7% relative to the strongest baseline, demonstrating its robustness even when the semantic gap between ID and OOD data is small. The complete results of this evaluation are provided in Table 6 in Appendix C.3.

### 4.5 HYPERPARAMETER SELECTION

The hyperparameter $\gamma$, which scales the standard deviation in Equation 2, is crucial for performance. Following established protocols (Sun et al., 2021; Sun & Li, 2022), we select $\gamma$ by creating a proxy OOD validation set. This set is generated by adding pixel-wise Gaussian noise, sampled from $\mathcal{N}(0, 0.2)$, to images from the ID validation set. Through this process, we identified optimal values of $\gamma = 3.0$ for all CIFAR models and $\gamma = 0.5$ for ImageNet models except EfficientNet-b0. A notable exception is EfficientNet-b0, which required a higher $\gamma = 4.0$. We attribute this to its use of the SiLU activation function, which, unlike ReLU, does not aggressively sparsify activations, leading to different statistical properties in the feature maps. We provide a deeper analysis of this effect in Section 5. Full details on all hyperparameters used for baseline re-evaluations are available in Appendix E.

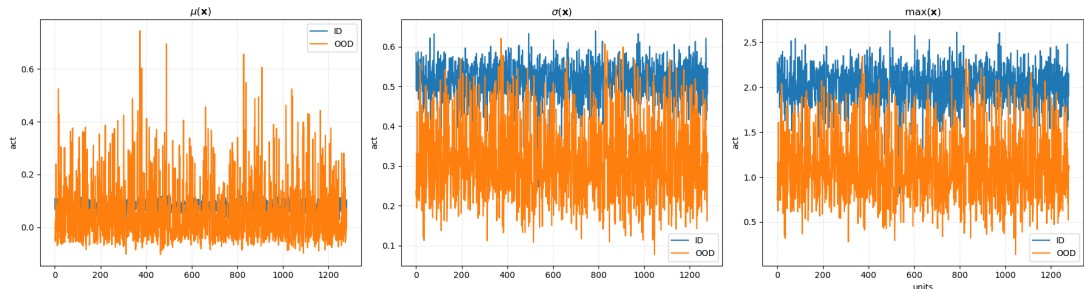

Figure 3: *Feature statistics for ID (ImageNet) vs. OOD (Texture) samples on an efficientNet-b0 backbone. While mean $\mu(\mathbf{x})$ show poor separation, both the standard deviation $\sigma(\mathbf{x})$ and maximum $m(\mathbf{x})$ statistics maintain a clear separation between ID and OOD activations.*

## 5 DISCUSSION

This section discusses the broader implications of DAVIS, analyzing its robustness against modern architectures and its practical advantages. We then justify our experimental focus on prevalent CNNs, address the method's current limitations, and outline promising avenues for future research.

**Evaluation using EfficientNet.** A critical observation from our experiments is the widespread failure of foundational baselines (e.g., DICE, ASH, SCALE) on the EfficientNet-b0 architecture (Table 5 and Table 11). We attribute this to its use of the dense SiLU activation function instead of the sparse ReLU common in other backbones, as the resulting dense features exhibit a high degree of overlap when aggregated using the standard mean (GAP). To strengthen this claim, we evaluated ResNet-18 with SiLU instead of ReLU on the CIFAR benchmarks. On this ResNet18-SiLU model, we observed that these same baselines struggled to outperform the standard energy score (i.e, DAVIS($m$), DAVIS($\mu, \sigma$) ). This provides strong evidence that the dense feature distributions produced by SiLU are indeed challenging for methods originally benchmarked on sparse, ReLU-based models. Detailed results for this experiment are provided in Appendix J.

DAVIS remains robust because it leverages more fundamental statistics. This is empirically validated in Figure 3, which shows that while the standard mean ($\mu(\mathbf{x})$) are largely indistinguishable for ID and OOD samples on EfficientNet-b0, both standard deviation ($\sigma(\mathbf{x})$) and maximum ($m(\mathbf{x})$) maintain a better separation. This inherent robustness not only allows our method to succeed but also to critically rescue baselines like ASH and SCALE from near-total failure, demonstrating generality of DAVIS.

**Overhead and Classification Accuracy.** As a post-hoc technique, DAVIS is deployed in a two-branch pipeline: OOD detection is performed using our modified features, while the original, unmodified features are used for the final classification of any sample deemed ID. It ensures that our OOD detection improvements come at no cost to the ID accuracy. On the other hand, computational overhead is negligible; for instance, on a ResNet-50, our method increases the total GFLOPs by less than 0.1%. A detailed ID classification accuracy using modified feature vector $h^{\text{DAVIS}}(\mathbf{x})$ is provided in Appendix F.

**Scope and Architectural Focus.** Our empirical evaluation of DAVIS is mainly focused on CNN-based architectures. The motivation behind this is two-fold: *(a)* Competitive baselines in the OOD detection literature extensively use CNN-based architectures (Sun & Li, 2022; Djurisic et al., 2023; Ming et al., 2023; Xu et al., 2024; Liu & Qin, 2024; 2025). For fair comparison, we adopt similar architectures to evaluate DAVIS. *(b)* CNN-based architectures continue to be widely used in both the research community and real-world applications. A comprehensive benchmark study carried out in prior work (Goldblum et al., 2023) has shown convolutional networks such as ResNet (He et al., 2016) and ConvNeXt (Liu et al., 2022b; Woo et al., 2023)

remain the default choice in real-world vision systems (including object detection, segmentation, retrieval, and classification) due to their strong inductive bias (translation invariance), computational efficiency, strong performance on moderate-scale data, and extensive ecosystem of pretrained models.

**Limitations and Future Work**. Our work opens several promising avenues for future research, stemming from the current architectural and methodological scope of DAVIS. *(a)* While the principles of DAVIS are general, its current implementation is tailored to architectures that use a spatial aggregation operation (e.g, GAP), such as CNNs. This makes it fundamentally incompatible with early Vision Transformers (ViTs) (Dosovitskiy et al., 2021) that bypass global aggregation, instead relying on a [CLS] token for classification. However, this opens a key future direction: extending our method to modern hierarchical models, like the Swin Transformer (Liu et al., 2022a), which have re-introduced a final aggregation layer. This is non-trivial, as it requires investigating the unique statistical properties of their feature maps. *(b)* Our evaluation deliberately centered on logit-based scoring functions, as methods like the energy score represent the state-of-the-art in terms of the performance-efficiency trade-off for post-hoc OOD detection. While alternatives like distance-based (Mahalanobis (Lee et al., 2018)) or density-based (KNN (Sun et al., 2022)) methods exist, they carry significant computational overhead. A valuable future study would thus be to investigate if the gains from combining DAVIS with these methods justify their additional cost.

## 6 ABLATION STUDIES

To provide a deeper understanding of DAVIS, this section summarizes three ablation studies. We present the main findings here and provide a detailed analysis in the appendix. These studies investigate our method's synergistic effect, justify our choice of statistics, and analyze its robustness to the hyperparameter $\gamma$.

**Synergy with Existing Methods.** To validate the synergistic effect of our method, we systematically evaluated the performance of baselines both with and without DAVIS. The results show that DAVIS consistently and significantly enhances all tested methods, including strong baselines like ASH and SCALE, on both CIFAR and ImageNet benchmarks. This confirms that our enriched feature representation provides a more robust foundation for existing OOD detectors. A detailed breakdown is provided in Appendix G.

**Analysis of Alternate Statistics.** To justify our choice of statistics, we performed an ablation study using the median and Shannon entropy Shannon (1948). Both alternatives performed poorly, with FPR95 scores often exceeding 95%, because they fail to produce a sufficiently distinctive and separable signal between ID and OOD samples. This analysis confirms that maximum and variance are superior choices as they produce a quantitatively stronger and more separable signal. The full results are presented in Appendix H.

**Sensitivity to Hyperparameter $\gamma$.** We analyzed the sensitivity of the DAVIS$(\mu, \sigma)$ method to its hyperparameter $\gamma$. Our findings indicate that the method is robust to the specific choice of $\gamma$. While performance generally improves as $\gamma$ increases from zero, the gains quickly saturate, showing that the model is not overly sensitive to precise tuning and performs well across a reasonable range of values. A detailed analysis is available in Appendix I.

## 7 CONCLUSION

In this work, we introduced DAVIS, a simple yet powerful technique that enhances OOD detection by leveraging statistical cues – specifically the channel-wise maximum and variance, that are typically discarded by GAP. Our extensive experiments demonstrate that DAVIS is a versatile, complementary tool that significantly boosts the performance of existing techniques across diverse datasets and architectures. Notably, it shows remarkable robustness on modern models like EfficientNet, where many conventional methods fail. Future work could explore extending these principles to transformer-based architectures or investigating a broader range of statistical measures.

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

## APPENDIX

## A  DESCRIPTION OF BASELINE METHODS

In resonance with existing work Liu et al. (2020); Sun et al. (2021); Sun & Li (2022); Djurisic et al. (2023), for the reader's convenience, we summarize in detail a few common techniques for defining OOD scores that measure the degree of ID-ness on the given sample. All the methods derive the score post hoc on neural networks trained with in-distribution data only. By convention, a higher score is indicative of being in-distribution, and vice versa.

**Softmax score**  One of the earliest works on OOD detection considered using the maximum softmax probability (MSP) to distinguish between $\mathcal{D}_{\text{in}}$ and $\mathcal{D}_{\text{out}}$ Hendrycks & Gimpel (2017). In detail, suppose the label space is $\mathcal{Y} = \{1, 2, \cdots, C\}$. We assume the classifier $f$ is defined in terms of a feature extractor $f : \mathcal{X} \to \mathbb{R}^m$ and a linear multinomial regressor with weight matrix $W \in \mathbb{R}^{C \times m}$ and bias vector $\mathbf{b} \in \mathbb{R}^C$. The prediction probability for each class is given by :

$$\mathbb{P}(y = c | \mathbf{x}) = \text{Softmax}(W h(\mathbf{x}) + \mathbf{b})_c \tag{6}$$

The softmax score is defined as $S_{\text{MSP}}(\mathbf{x}; f) := \max_c \mathbb{P}(y = c | \mathbf{x})$.

**ODIN**  Liang et al. (2018) This method introduced temperature scaling and input perturbation to improve the separation of MSP for ID and OOD data. $\tilde{\mathbf{x}}$ denotes perturbed input.

$$\mathbb{P}(y = c | \tilde{\mathbf{x}}) = \text{Softmax}[(W h(\tilde{\mathbf{x}}) + \mathbf{b})/T]_c \tag{7}$$

the ODIN score is defined as $S_{\text{ODIN}}(\mathbf{x}; f) := \max_c \mathbb{P}(y = c | \tilde{\mathbf{x}})$.

**Energy score**  The energy function Liu et al. (2020) maps the output logit to a scalar $S_{\text{Energy}}(\mathbf{x}; f) \in \mathbb{R}$, which is relatively lower for ID data:

$$S_{\text{Energy}}(\mathbf{x}; f) = -\text{Energy}(\mathbf{x}; f) = \log\left(\sum_{c=1}^{C} \exp(f_c(\mathbf{x}))\right) \tag{8}$$

They used the *negative energy score* for OOD detection, in order to align with the convention that $S(\mathbf{x}; f)$ is higher for ID data and vice versa.

**ReAct**  They perform post hoc modification of penultimate layer of the neural network. It works by truncating the feature activations at a threshold $c$, i.e., replacing each activation with $\min(x, c)$. This limits the influence of abnormally large activations often caused by OOD inputs. The truncation threshold is set with the validation strategy in Sun et al. (2021). Formally,

$$h^{\text{ReAct}}(\mathbf{x}) = \text{ReAct}(h(\mathbf{x}); c) = \min(h(\mathbf{x}), c) \quad \text{(applied element-wise)}$$

The final model output becomes:

$$f^{\text{ReAct}}(\mathbf{x}) = W^\top h^{\text{ReAct}}(\mathbf{x}) + \mathbf{b}$$

This method also uses energy score $S_{\text{Energy}}(\mathbf{x}; f^{\text{ReAct}}) \in \mathbb{R}$ for OOD detection.

**DICE**  Sun & Li (2022) It is a post hoc method to improve OOD detection by retaining only the most informative weights in the final layer of a pre-trained neural network. A *contribution matrix* $V \in \mathbb{R}^{m \times C}$ is computed, where each column is:

$$\mathbf{v}_c = \mathbb{E}_{\mathbf{x} \in \mathcal{D}}[\mathbf{w}_c \odot h(\mathbf{x})]$$

with $\odot$ denoting element-wise multiplication. Each entry in $V$ quantifies the average contribution of a feature unit to class $c$. A binary *masking matrix* $M \in \mathbb{R}^{m \times C}$ selects the top-$k$ highest-contributing weights, setting others to zero. The sparsified output is:

$$f^{\text{DICE}}(\mathbf{x}; \theta) = (M \odot W)^{\top} h(\mathbf{x}) + \mathbf{b}$$

This method also uses energy score $S_{\text{Energy}}(\mathbf{x}; f^{\text{DICE}}) \in \mathbb{R}$ for OOD detection.

**ASH** (Djurisic et al., 2023) It is also a post-hoc method that simplifies feature representations to improve OOD detection. They proposes three versions of ASH, we presented only the best performing version i.e, ASH-S. Given an input activation vector $h(\mathbf{x})$ and a pruning percentile $p$, ASH (Djurisic et al., 2023) proceeds as follows shaping the activation of penultimate layer $h(\mathbf{x})$ to get $h^{\text{ASH}}(\mathbf{x})$:

1. Compute the $p$-th percentile threshold $t$ of $h(\mathbf{x})$.
2. Let $s_1 = \sum h(\mathbf{x})$, the sum of all activation values before pruning.
3. Set all values in $h(\mathbf{x})$ less than $t$ to zero.
4. Let $s_2 = \sum h(\mathbf{x})$, the sum after pruning.
5. Scale all non-zero values in $h(\mathbf{x})$ by $\exp(s_1/s_2)$.

The final model output becomes, which is then used to compute energy score $S_{\text{Energy}}(\mathbf{x}; f^{\text{ASH}}) \in \mathbb{R}$ for OOD detection :

$$f^{\text{ASH}}(\mathbf{x}) = W^{\top} h^{\text{ASH}}(\mathbf{x}) + \mathbf{b}$$

**SCALE** (Xu et al., 2024) It is a post-hoc method designed to enhance out-of-distribution (OOD) detection by adaptively scaling the activation of the penultimate layer $h(\mathbf{x})$ before computing the final classifier output. Given an input activation vector $h(\mathbf{x})$ and a pruning percentile $p$, SCALE (Xu et al., 2024) proceeds as follows to obtain the scaled activation $h^{\text{SCALE}}(\mathbf{x})$:

1. Compute the $p$-th percentile threshold $t$ of $h(\mathbf{x})$.
2. Let $s_1 = \sum h(\mathbf{x})$, the sum of all activation values before pruning.
3. Construct a binary mask $\mathbf{1}_{\{h(\mathbf{x}) \geq t\}}$ that keeps only the top-$p$ activations.
4. Let $s_2 = \sum h(\mathbf{x}) \cdot \mathbf{1}_{\{h(\mathbf{x}) \geq t\}}$, the sum of the top-$p$ activations.
5. Compute the scaling ratio $r = \frac{s_1}{s_2}$.
6. Scale the original activations by $\exp(r)$:

$$h^{\text{SCALE}}(\mathbf{x}) = \exp(r) \cdot h(\mathbf{x}).$$

The final model output is then computed with the scaled activations, and the *energy score* is used for OOD detection:

$$f^{\text{SCALE}}(\mathbf{x}) = W^{\top} h^{\text{SCALE}}(\mathbf{x}) + \mathbf{b}, \quad S_{\text{Energy}}(\mathbf{x}; f^{\text{SCALE}}) \in \mathbb{R}.$$

## B  STATISTICAL ANALYSIS

In this section, we present a detailed statistical analysis of our method, DAVIS, demonstrating how it enhances the separation between in-distribution (ID) and out-of-distribution (OOD) samples. This increased separation leads to a sharper decision boundary between ID and OOD regions. Our analysis builds on key observations commonly made in prior work on OOD detection (Liu et al., 2020; Sun et al., 2021; Sun & Li, 2022; Djurisic et al., 2023; Xu et al., 2024), which we adopt as foundational to our analysis.

## B.1 SETUP

We consider a trained neural network parameterized by $\theta$, which encodes an input $\mathbf{x} \in \mathbb{R}^d$ to $n$ spatial activation maps, denoted by $g(\mathbf{x}) \in \mathbb{R}^{n \times k \times k}$. These activation maps are then transformed into $n$ dimensional feature vector $h(\mathbf{x}) \in \mathbb{R}^n$ (i.e., penultimate layer) via global average pooling (GAP) as shown in Equation 9, where Avg denotes the GAP operation applied independently to each of the $n$ activation maps in $g(\mathbf{x})$.

$$h(\mathbf{x}) = \texttt{Avg}\left(g(\mathbf{x})\right) \tag{9}$$

A weight matrix $\mathbf{W} \in \mathbb{R}^{n \times C}$ connects the feature vector $h(\mathbf{x}) \in \mathbb{R}^n$ to the output logit $f(\mathbf{x}) \in \mathbb{R}^C$ as shown in Equation 10, where $C$ is the total number of classes in $\mathcal{Y} = \{1, 2, \cdots, C\}$. The function maps the output logit $f(\mathbf{x})$ to a scalar energy $\mathbf{E}_\theta(\mathbf{x})$, which is relatively lower for ID data Liu et al. (2020) as shown in Equation 11.

$$f(\mathbf{x}) = \mathbf{W}^\top h(\mathbf{x}) + \mathbf{b} \tag{10}$$

$$S_\theta(\mathbf{x}) = -\mathbf{E}_\theta(\mathbf{x}) = \log\left(\sum_{c=1}^{C} \exp(f(\mathbf{x}; \theta))\right) \tag{11}$$

The goal of OOD detection is to learn a decision boundary $G_\lambda(\mathbf{x}; \theta)$ that classifies a test sample $\mathbf{x} \in \mathcal{X}$:

$$G_\lambda(\mathbf{x}; \theta) = \begin{cases} \text{in} & \text{if } S_\theta(\mathbf{x}) \geq \lambda \\ \text{out} & \text{if } S_\theta(\mathbf{x}) < \lambda \end{cases} \tag{12}$$

where a thresholding mechanism is employed to distinguish between ID and OOD samples. To align with the convention, samples with higher scores $S_\theta(\mathbf{x})$ are classified as ID while samples with lower scores are classified as OOD. By convention (Liu et al., 2020), the threshold $\lambda$ is typically chosen such that a high fraction of ID, (*e.g., 95%*) is correctly classified in practice.

As part of DAVIS, we retrieve mean $\mu(\mathbf{x}) \in \mathbb{R}^n$, variance $\sigma^2(\mathbf{x}) \in \mathbb{R}^n$, and maximum $m(\mathbf{x}) \in \mathbb{R}^n$ from the feature maps in $g(\mathbf{x})$. Up to this point, $\mu(\mathbf{x})$ and $h(\mathbf{x})$ are numerically equivalent. However, semantically, $\mu(\mathbf{x})$ denotes the extracted statistical features, while $h(\mathbf{x})$ refers to the penultimate layer representation. DAVIS modifies the penultimate layer $h(\mathbf{x})$ of the model using $\mu(\mathbf{x})$, $\sigma^2(\mathbf{x})$, and $m(\mathbf{x})$ for enhanced OOD detection. In this work, we extensively explored two versions of DAVIS:

- DAVIS($m$) replaces feature vector $h(\mathbf{x})$ by the maximum (dominant) $m(\mathbf{x})$ as shown in Equation 13.

$$h^{\texttt{DAVIS}(m)}(\mathbf{x}) = m(\mathbf{x}) \tag{13}$$

- DAVIS($\mu, \sigma$) augments the mean activation with its corresponding channel-wise standard deviation $\sigma(\mathbf{x})$, scaled by a hyperparameter $\gamma$ as shown in Equation 14.

$$h^{\texttt{DAVIS}(\mu,\sigma)}(\mathbf{x}) = \mu(\mathbf{x}) + \gamma\sigma(\mathbf{x}) \tag{14}$$

In effect, both DAVIS versions raise the activation level of feature vector $h(\mathbf{x})$. Our analysis focuses on understanding how modifying the penultimate feature representation $h(\mathbf{x})$ impacts the final OOD score. Recall that the logit vector $f(\mathbf{x})$ is a linear transformation of these features, i.e., $f(\mathbf{x}) = \mathbf{W}^\top h(\mathbf{x}) + \mathbf{b}$. Consequently, any logit-based scoring function, such as the energy score, is ultimately a function of $h(\mathbf{x})$.

## B.2 ANALYSIS

In this section, we provide a statistical analysis of our method. Our analysis is grounded in a foundational observation regarding the behavior of features extracted from well-trained classifiers, consistent with prior work (Liu et al., 2020; Sun et al., 2021; Djurisic et al., 2023; Xu et al., 2024). We denote in-distribution and out-of-distribution samples as $\mathbf{x}_{\text{in}}$ and $\mathbf{x}_{\text{out}}$, respectively.

**Observation 1.** *Given a well-trained model $\theta$, the statistical features extracted from $\mathbf{x}_{in}$ samples consistently exhibit higher magnitudes than those from $\mathbf{x}_{out}$ samples. This holds true for the channel-wise mean $\mu(\mathbf{x})$, maximum $m(\mathbf{x})$, and standard deviation $\sigma(\mathbf{x})$. Formally, we state this as shown in Equations 15. More precisely, these inequalities are characteristic of the majority of individual feature dimensions as shown in Figure 4.*

$$\mathbb{E}_{\mathbf{x}\sim\mathcal{D}_{\text{in}}}[\mu(\mathbf{x})] \geq \mathbb{E}_{\mathbf{x}\sim\mathcal{D}_{\text{out}}}[\mu(\mathbf{x})] \tag{15a}$$

$$\mathbb{E}_{\mathbf{x}\sim\mathcal{D}_{\text{in}}}[m(\mathbf{x})] \geq \mathbb{E}_{\mathbf{x}\sim\mathcal{D}_{\text{out}}}[m(\mathbf{x})] \tag{15b}$$

$$\mathbb{E}_{\mathbf{x}\sim\mathcal{D}_{\text{in}}}[\sigma(\mathbf{x})] \geq \mathbb{E}_{\mathbf{x}\sim\mathcal{D}_{\text{out}}}[\sigma(\mathbf{x})] \tag{15c}$$

*This fundamental property enables the network to perform both its primary classification task and OOD detection effectively.*

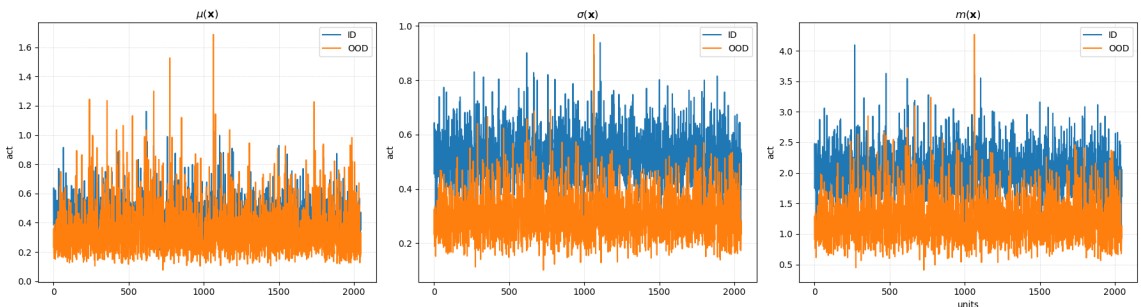

Figure 4: *Unit-wise comparison of statistical features for ID vs. OOD samples, with values averaged over the entire test set. Across a majority of feature dimensions, the mean ($\mu(\mathbf{x})$), standard deviation ($\sigma(\mathbf{x})$), and maximum ($m(\mathbf{x})$) statistics all exhibit consistently higher values for ID samples (blue) than for OOD samples (orange). Results are shown for a ResNet-50 model with ImageNet-1K as the ID dataset and Texture as the OOD dataset. This trend holds consistently across other architectures and data combinations.*

**Definition 1.** *To quantify the separation for a given feature vector $h(\mathbf{x})$, we define the separation gap $\Delta_h$, as the difference in its expected value across the ID and OOD distributions:*

$$\Delta_h = \mathbb{E}_{\mathbf{x}\sim\mathcal{D}_{\text{in}}}[h(\mathbf{x})] - \mathbb{E}_{\mathbf{x}\sim\mathcal{D}_{\text{out}}}[h(\mathbf{x})] = \mathbb{E}[h(\mathbf{x}_{\text{in}}) - h(\mathbf{x}_{\text{out}})]$$

*Specifically, the separation gaps for the mean, maximum, and our combined mean-and-standard-deviation feature are:*

$$\Delta_{\mu} := \mathbb{E}_{\mathbf{x}\sim\mathcal{D}_{\text{in}}}[\mu(\mathbf{x})] - \mathbb{E}_{\mathbf{x}\sim\mathcal{D}_{\text{out}}}[\mu(\mathbf{x})] = \mathbb{E}[\mu(\mathbf{x}_{\text{in}}) - \mu(\mathbf{x}_{\text{out}})], \tag{16a}$$

$$\Delta_{m} := \mathbb{E}_{\mathbf{x}\sim\mathcal{D}_{\text{in}}}[m(\mathbf{x})] - \mathbb{E}_{\mathbf{x}\sim\mathcal{D}_{\text{out}}}[m(\mathbf{x})] = \mathbb{E}[m(\mathbf{x}_{\text{in}}) - m(\mathbf{x}_{\text{out}})], \tag{16b}$$

$$\Delta_{\mu,\sigma} := \mathbb{E}_{\mathbf{x}\sim\mathcal{D}_{\text{in}}}[\mu(\mathbf{x})+\sigma(\mathbf{x})] - \mathbb{E}_{\mathbf{x}\sim\mathcal{D}_{\text{out}}}[\mu(\mathbf{x})+\sigma(\mathbf{x})] = \mathbb{E}[\mu(\mathbf{x}_{\text{in}})+\sigma(\mathbf{x}_{\text{in}}) - \mu(\mathbf{x}_{\text{out}}) - \sigma(\mathbf{x}_{\text{out}})] \tag{16c}$$

**Lemma 1.** *Given Observation 1, the separation gap of the combined mean-and-standard-deviation feature is greater than or equal to that of the mean feature alone:*

$$\Delta_{\mu,\sigma} \geq \Delta_{\mu}$$

*Proof.* By linearity of expectation, we can expand the definition of $\Delta_{\mu,\sigma}$ as follows:

$$\Delta_{\mu,\sigma} - \Delta_{\mu}$$
$$= \Big( \mathbb{E}_{\mathbf{x}\sim\mathcal{D}_{\mathrm{in}}}[\mu(\mathbf{x}) + \sigma(\mathbf{x})] - \mathbb{E}_{\mathbf{x}\sim\mathcal{D}_{\mathrm{out}}}[\mu(\mathbf{x}) + \sigma(\mathbf{x})] \Big) - \Big( \mathbb{E}_{\mathbf{x}\sim\mathcal{D}_{\mathrm{in}}}[\mu(\mathbf{x})] - \mathbb{E}_{\mathbf{x}\sim\mathcal{D}_{\mathrm{out}}}[\mu(\mathbf{x})] \Big)$$
$$= \mathbb{E}_{\mathbf{x}\sim\mathcal{D}_{\mathrm{in}}}[\mu(\mathbf{x})] + \mathbb{E}_{\mathbf{x}\sim\mathcal{D}_{\mathrm{in}}}[\sigma(\mathbf{x})] - \mathbb{E}_{\mathbf{x}\sim\mathcal{D}_{\mathrm{out}}}[\mu(\mathbf{x})] - \mathbb{E}_{\mathbf{x}\sim\mathcal{D}_{\mathrm{out}}}[\sigma(\mathbf{x})] - \mathbb{E}_{\mathbf{x}\sim\mathcal{D}_{\mathrm{in}}}[\mu(\mathbf{x})] + \mathbb{E}_{\mathbf{x}\sim\mathcal{D}_{\mathrm{out}}}[\mu(\mathbf{x})]$$
$$= \mathbb{E}_{\mathbf{x}\sim\mathcal{D}_{\mathrm{in}}}[\sigma(\mathbf{x})] - \mathbb{E}_{\mathbf{x}\sim\mathcal{D}_{\mathrm{out}}}[\sigma(\mathbf{x})]$$
$$\geq 0 \qquad\qquad \text{(Recall Equation } 15c \text{ of Observation 1)}$$

This result is empirically validated in leftmost plot of Figure 5, which shows that incorporating the standard deviation consistently increases the separation between ID and OOD samples. $\square$

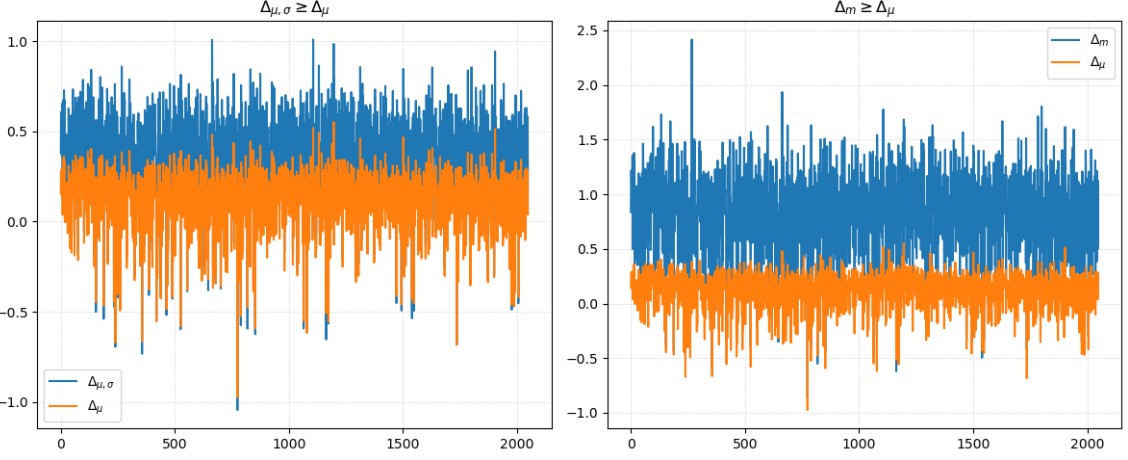

Figure 5: *Comparison of the separation gap $\Delta$ achieved by different statistical features, averaged over all test samples. Left: It demonstrate that incorporating the standard deviation $\Delta_{\mu,\sigma}$ yields a larger separation gap than using the mean activation alone $\Delta_{\mu}$. Right: It demonstrate that using the maximum activation $\Delta_m$ yields a larger separation gap than using the mean activation $\Delta_{\mu}$. Results are shown for a ResNet-50 model with ImageNet as the ID dataset and Texture as the OOD dataset. This finding holds consistently across other architectures and data combinations.*

**Observation 2.** *Our experiments consistently show that the maximum statistic provides a stronger separation signal than the mean statistic. We state this empirical finding as the following inequality:*

$$\Delta_m \geq \Delta_{\mu}$$

*This is consistent with the behavior of discriminative classifiers: ID samples are trained to elicit high-magnitude feature responses, while OOD samples tend to produce weaker, more uniform activations. This makes the maximum a more distinctive signal than the mean, which is empirically validated in right plot in Figure 5.*

**Assumption 1.** *Our analysis adopts a key assumption from ReAct (Sun et al., 2021), a principle also leveraged by subsequent methods like SCALE (Xu et al., 2024) and ASH (Djurisic et al., 2023). To formally analyze the effect of feature modifications on the output logits $f(\mathbf{x}) = \mathbf{W}^\top h(\mathbf{x}) + \mathbf{b}$ we assume a sufficient (though not strictly necessary) condition on the classifier's final weight matrix, $\mathbf{W}$. Specifically, we assume that $\mathbf{W}^\top \mathbf{1} \geq 0$ element-wise. As noted in (Sun et al., 2021), this property is often observed empirically and can be achieved by adding a positive constant to $\mathbf{W}$ without changing the final classification decisions.*

**Theorem 1.** *Let $h^\mu(\mathbf{x})$ be the baseline feature vector (from GAP) and $h^{DAVIS}(\mathbf{x})$ be an enhanced feature vector from our method. Let $\Delta_h^\mu = \mathbb{E}[h^\mu(\mathbf{x}_{\text{in}})] - \mathbb{E}[h^\mu(\mathbf{x}_{\text{out}})]$ and $\Delta_h^{DAVIS} = \mathbb{E}[h^{DAVIS}(\mathbf{x}_{\text{in}})] - \mathbb{E}[h^{DAVIS}(\mathbf{x}_{\text{out}})]$ be the respective feature separation gap vectors. Then, under the Assumption 1, the separation between the ID and OOD logits is also increased:*

$$\mathbb{E}[f^{DAVIS}(\mathbf{x}_{\text{in}}) - f^{DAVIS}(\mathbf{x}_{\text{out}})] \geq \mathbb{E}[f^\mu(\mathbf{x}_{\text{in}}) - f^\mu(\mathbf{x}_{\text{out}})]$$

*Proof.* To derive the effect on the distribution of model output, consider output logits $f(\mathbf{x}) = \mathbf{W}^\top h(\mathbf{x}) + \mathbf{b}$ as shown in Equation 10 and assume without loss of generality that element-wise $\mathbf{W}^\top \mathbf{1} \geq 0$. This can be achieved by adding a positive constant to $\mathbf{W}$ without changing the output probabilities or classification decision (Assumption 1).

*Case 1:* For notational clarity in the following analysis, let us denote the standard feature vector (from GAP) as $h^\mu(\mathbf{x})$ and our enhanced feature vector as $h^{\mu+\sigma}(\mathbf{x})$. The corresponding logits are then computed as $f^\mu(\mathbf{x}) = \mathbf{W}^\top h^\mu(\mathbf{x}) + \mathbf{b}$ and $f^{\mu+\sigma}(\mathbf{x}) = \mathbf{W}^\top h^{\mu+\sigma}(\mathbf{x}) + \mathbf{b}$, respectively.

Let $\delta = \mathbb{E}\left[h^{\mu+\sigma}(\mathbf{x}_{\text{in}}) - h^{\mu+\sigma}(\mathbf{x}_{\text{out}})\right] - \mathbb{E}\left[h^\mu(\mathbf{x}_{\text{in}}) - h^\mu(\mathbf{x}_{\text{out}})\right] \geq 0$ (recall Lemma 1)

$$
\begin{aligned}
&\mathbb{E}\left[f^{\mu+\sigma}(\mathbf{x}_{\text{in}}) - f^{\mu+\sigma}(\mathbf{x}_{\text{out}})\right] \\
=\ & \mathbb{E}\left[\mathbf{W}^\top(h^{\mu+\sigma}(\mathbf{x}_{\text{in}}) - h^{\mu+\sigma}(\mathbf{x}_{\text{out}}))\right] \\
=\ & \mathbf{W}^\top \mathbb{E}\left[h^{\mu+\sigma}(\mathbf{x}_{\text{in}}) - h^{\mu+\sigma}(\mathbf{x}_{\text{out}})\right] \\
=\ & \mathbf{W}^\top \left( \mathbb{E}\left[h^{\mu+\sigma}(\mathbf{x}_{\text{in}}) - h^{\mu+\sigma}(\mathbf{x}_{\text{out}})\right] - \mathbb{E}\left[h^\mu(\mathbf{x}_{\text{in}}) - h^\mu(\mathbf{x}_{\text{out}})\right] + \mathbb{E}\left[h^\mu(\mathbf{x}_{\text{in}}) - h^\mu(\mathbf{x}_{\text{out}})\right] \right) \\
=\ & \mathbf{W}^\top \left( \mathbb{E}\left[h^\mu(\mathbf{x}_{\text{in}}) - h^\mu(\mathbf{x}_{\text{out}})\right] + \Delta_{\mu,\sigma} - \Delta_\mu \right) \\
=\ & \mathbf{W}^\top \left( \mathbb{E}\left[h^\mu(\mathbf{x}_{\text{in}}) - h^\mu(\mathbf{x}_{\text{out}})\right] + \delta\mathbf{1} \right) \\
=\ & \mathbb{E}\left[\mathbf{W}^\top \left(h^\mu(\mathbf{x}_{\text{in}}) - h^\mu(\mathbf{x}_{\text{out}})\right)\right] + \delta\mathbf{W}^\top\mathbf{1} \\
\geq\ & \mathbb{E}\left[f^\mu(\mathbf{x}_{\text{in}}) - f^\mu(\mathbf{x}_{\text{out}})\right] \qquad (\because \mathbf{W}^\top\mathbf{1} \geq 0; \text{ Assumption 1})
\end{aligned}
$$

*Case 2:* Similar to above, for notational clarity, let us denote the standard feature vector as $h^\mu(\mathbf{x})$ and our dominant feature vector as $h^m(\mathbf{x})$. The corresponding logits are then computed as $f^\mu(\mathbf{x}) = \mathbf{W}^\top h^\mu(\mathbf{x}) + \mathbf{b}$ and $f^m(\mathbf{x}) = \mathbf{W}^\top h^m(\mathbf{x}) + \mathbf{b}$, respectively.

Let $\delta = \mathbb{E}\left[h^m(\mathbf{x}_{\text{in}}) - h^m(\mathbf{x}_{\text{out}})\right] - \mathbb{E}\left[h^\mu(\mathbf{x}_{\text{in}}) - h^\mu(\mathbf{x}_{\text{out}})\right] \geq 0$ (recall Observation 2)

$$\mathbb{E}\left[f^m(\mathbf{x}_{\text{in}}) - f^m(\mathbf{x}_{\text{out}})\right]$$

$$= \mathbb{E}\left[\mathbf{W}^\top(h^m(\mathbf{x}_{\text{in}}) - h^m(\mathbf{x}_{\text{out}}))\right]$$

$$= \mathbf{W}^\top \mathbb{E}\left[h^m(\mathbf{x}_{\text{in}}) - h^m(\mathbf{x}_{\text{out}})\right]$$

$$= \mathbf{W}^\top \left( \mathbb{E}\left[h^m(\mathbf{x}_{\text{in}}) - h^m(\mathbf{x}_{\text{out}})\right] - \mathbb{E}\left[h^\mu(\mathbf{x}_{\text{in}}) - h^\mu(\mathbf{x}_{\text{out}})\right] + \mathbb{E}\left[h^\mu(\mathbf{x}_{\text{in}}) - h^\mu(\mathbf{x}_{\text{out}})\right] \right)$$

$$= \mathbf{W}^\top \left( \mathbb{E}\left[h^\mu(\mathbf{x}_{\text{in}}) - h^\mu(\mathbf{x}_{\text{out}})\right] + \Delta_m - \Delta_\mu \right)$$

$$= \mathbf{W}^\top \left( \mathbb{E}\left[h^\mu(\mathbf{x}_{\text{in}}) - h^\mu(\mathbf{x}_{\text{out}})\right] + \delta \mathbf{1} \right)$$

$$= \mathbb{E}\left[\mathbf{W}^\top \left(h^\mu(\mathbf{x}_{\text{in}}) - h^\mu(\mathbf{x}_{\text{out}})\right)\right] + \delta \mathbf{W}^\top \mathbf{1}$$

$$\geq \mathbb{E}\left[f^\mu(\mathbf{x}_{\text{in}}) - f^\mu(\mathbf{x}_{\text{out}})\right] \qquad (\because \mathbf{W}^\top \mathbf{1} \geq 0;\ \text{Assumption 1})$$

Thus, our analysis demonstrates that the enhanced feature separation provided by both formulations of `DAVIS` directly propagates to the logit space, resulting in a more discriminative output for OOD detection. Note that based on Assumption 1 inspired by ReAct (Sun et al., 2021) the condition of $\mathbf{W}^\top \mathbf{1} \geq 0$ is sufficient but not necessary for this result to hold.

**Why `DAVIS` improves the OOD scoring functions?** Our analysis demonstrates that `DAVIS` improves OOD detection by amplifying the separation between the expected logit values of ID and OOD samples. For logit-based scoring functions, such as the energy score Liu et al. (2020), this increased logit separation directly translates to a wider gap between the ID and OOD score distributions. This enhanced separability improves the ability to distinguish between ID and OOD, leading to better OOD detection performance. This mechanism is empirically validated in Figure 2, which illustrates the clearer separation in score densities after applying `DAVIS`.

$\square$

## C  DETAILED OOD DETECTION PERFORMANCE

### C.1  CIFAR EVALUATION

Table 3 and Table 4 report detailed OOD performance across six test datasets for ResNet-18, ResNet-34, DenseNet-101, and MobileNet-v2 trained on CIFAR-10 and CIFAR-100. For MobileNet-v2, we modified the classification head so that the penultimate layer has 512 dimensions, comparable to other models (DenseNet-101: 342; ResNet-18/34: 512) used in our evaluation.

The foundational baselines (MSP, ODIN, Energy, ReAct, DICE, ASH) did not report results for ResNet-18, ResNet-34, or MobileNet-v2 in their original papers. To ensure a fair comparison, we re-evaluated these methods following the hyperparameter guidelines from their respective publications.

Table 3: Detailed results on six common OOD benchmark datasets: SVHN, Places365, iSUN, Textures, LSUN-crop, LSUN-resize. We used same model pre-trained on **CIFAR-10**. ↓ indicates lower values are better and ↑ indicates larger values are better.

| Model | Method | SVHN FPR95↓ | SVHN AUROC↑ | Places365 FPR95↓ | Places365 AUROC↑ | iSUN FPR95↓ | iSUN AUROC↑ | Textures FPR95↓ | Textures AUROC↑ | LSUN-c FPR95↓ | LSUN-c AUROC↑ | LSUN-r FPR95↓ | LSUN-r AUROC↑ | Average FPR95↓ | Average AUROC↑ |
|---|---|---|---|---|---|---|---|---|---|---|---|---|---|---|---|
| ResNet-18 | MSP | 60.39 | 92.40 | 63.49 | 88.38 | 56.59 | 91.18 | 62.71 | 90.10 | 51.87 | 93.64 | 55.53 | 91.69 | 58.43 | 91.23 |
| | ODIN | 35.96 | 94.70 | 41.11 | 92.06 | 23.36 | 96.56 | 46.74 | 91.97 | 6.66 | 98.71 | 20.04 | 96.93 | 28.98 | 95.16 |
| | Energy | 44.32 | 94.04 | 41.31 | 91.73 | 35.46 | 94.64 | 50.39 | 91.12 | 9.77 | 98.19 | 32.41 | 95.16 | 35.61 | 94.14 |
| | ReAct | 42.31 | 94.12 | 40.74 | 92.25 | 24.06 | 96.26 | 40.44 | 93.69 | 12.27 | 97.90 | 21.02 | 96.67 | 30.14 | 95.15 |
| | DICE | 17.60 | 97.09 | 46.16 | 90.66 | 38.68 | 94.32 | 44.50 | 91.81 | 1.90 | 99.57 | 36.66 | 94.67 | 30.92 | 94.69 |
| | ReAct+DICE | 12.25 | 97.85 | 45.77 | 91.27 | 18.12 | 96.92 | 26.90 | 95.60 | 1.35 | 99.65 | 16.93 | 97.09 | 20.22 | 96.40 |
| | ASH | 7.87 | 98.43 | 49.69 | 89.57 | 23.27 | 96.33 | 26.12 | 95.88 | 2.10 | 99.46 | 21.91 | 96.47 | 21.83 | 96.02 |
| | DAVIS(m) | 19.81 | 96.29 | 32.32 | 93.73 | 15.79 | 97.37 | 21.90 | 96.44 | 5.83 | 98.73 | 13.63 | 97.61 | 18.21 | 96.69 |
| | DAVIS(m) + +DICE | 7.95 | 98.50 | 30.55 | 93.97 | 6.70 | 98.59 | 9.66 | 98.28 | 1.24 | 99.73 | 6.63 | 98.57 | 10.46 | 97.94 |
| | DAVIS(μ,σ) | 19.83 | 96.47 | 36.11 | 93.16 | 20.48 | 96.90 | 26.77 | 95.89 | 5.62 | 98.79 | 17.75 | 97.21 | 21.09 | 96.40 |
| | DAVIS(μ,σ) + DICE | 7.85 | 98.57 | 35.27 | 93.18 | 11.96 | 97.96 | 14.04 | 97.71 | 1.14 | 99.76 | 10.71 | 98.03 | 13.49 | 97.54 |
| ResNet-34 | MSP | 62.20 | 91.10 | 62.76 | 88.95 | 52.52 | 92.81 | 57.93 | 90.75 | 42.06 | 95.17 | 51.72 | 92.98 | 54.86 | 91.96 |
| | ODIN | 40.33 | 92.49 | 37.63 | 92.05 | 10.29 | 98.08 | 39.06 | 92.94 | 2.58 | 99.31 | 8.44 | 98.32 | 23.06 | 95.53 |
| | Energy | 35.44 | 93.76 | 38.15 | 92.27 | 19.90 | 96.87 | 42.52 | 92.54 | 3.38 | 99.10 | 16.86 | 97.20 | 26.04 | 95.29 |
| | ReAct | 33.03 | 93.66 | 36.11 | 93.26 | 21.64 | 96.61 | 41.19 | 93.22 | 6.83 | 98.57 | 19.36 | 96.88 | 26.36 | 95.37 |
| | DICE | 26.78 | 95.46 | 39.75 | 91.83 | 16.00 | 97.41 | 40.05 | 92.91 | 0.71 | 99.82 | 14.73 | 97.58 | 23.00 | 95.84 |
| | ReAct+DICE | 19.63 | 96.31 | 40.69 | 92.65 | 17.98 | 97.05 | 33.32 | 94.18 | 0.93 | 99.77 | 17.98 | 97.07 | 21.75 | 96.17 |
| | ASH | 14.58 | 97.56 | 41.91 | 90.96 | 15.71 | 97.33 | 26.88 | 95.38 | 1.74 | 99.54 | 16.59 | 97.23 | 19.57 | 96.34 |
| | DAVIS(m) | 29.29 | 94.59 | 29.59 | 94.21 | 10.17 | 98.03 | 20.23 | 96.73 | 4.00 | 99.10 | 9.45 | 98.16 | 17.12 | 96.81 |
| | DAVIS(m) + +DICE | 14.79 | 97.29 | 27.51 | 94.59 | 5.09 | 98.98 | 10.07 | 98.26 | 1.22 | 99.75 | 5.36 | 98.95 | 10.67 | 97.97 |
| | DAVIS(μ,σ) | 28.96 | 94.90 | 32.66 | 93.69 | 11.31 | 97.88 | 23.07 | 96.39 | 3.47 | 99.22 | 10.35 | 98.04 | 18.30 | 96.69 |
| | DAVIS(μ,σ) + +DICE | 15.89 | 97.25 | 30.60 | 93.97 | 6.31 | 98.78 | 12.55 | 97.88 | 0.83 | 99.82 | 6.37 | 98.78 | 12.09 | 97.75 |
| DenseNet-101 | MSP | 64.76 | 88.33 | 60.19 | 88.56 | 33.34 | 95.41 | 56.60 | 90.17 | 23.41 | 96.75 | 33.88 | 95.39 | 45.36 | 92.43 |
| | ODIN | 33.09 | 94.41 | 36.68 | 92.34 | 3.22 | 99.20 | 38.49 | 91.61 | 1.84 | 99.53 | 2.89 | 99.28 | 19.37 | 96.06 |
| | Energy | 37.91 | 93.59 | 36.38 | 92.39 | 7.83 | 98.23 | 43.85 | 90.49 | 1.95 | 99.47 | 7.34 | 98.34 | 22.54 | 95.42 |
| | ReAct | 23.18 | 96.28 | 33.97 | 92.98 | 5.95 | 98.45 | 32.25 | 93.98 | 2.47 | 99.33 | 5.44 | 98.55 | 17.21 | 96.59 |
| | DICE | 16.68 | 96.96 | 37.46 | 92.06 | 2.25 | 99.41 | 28.05 | 92.70 | 0.16 | 99.94 | 2.44 | 99.35 | 14.51 | 96.74 |
| | ReAct+DICE | 4.63 | 99.02 | 36.10 | 92.93 | 1.80 | 99.51 | 17.32 | 96.76 | 0.12 | 99.95 | 1.93 | 99.47 | 10.32 | 97.94 |
| | ASH | 16.20 | 97.21 | 37.79 | 92.02 | 3.91 | 98.94 | 26.40 | 94.61 | 0.84 | 99.69 | 4.15 | 98.92 | 14.88 | 96.90 |
| | DAVIS(m) | 30.50 | 94.50 | 33.04 | 93.28 | 7.81 | 98.29 | 25.16 | 95.85 | 9.95 | 98.19 | 7.32 | 98.37 | 18.96 | 96.41 |
| | DAVIS(m) + DICE | 8.30 | 98.37 | 29.47 | 93.92 | 1.85 | 99.56 | 7.16 | 98.69 | 1.26 | 99.72 | 1.92 | 99.55 | 8.33 | 98.30 |
| | DAVIS(μ,σ) | 24.75 | 95.93 | 32.08 | 93.43 | 5.77 | 98.64 | 25.23 | 95.84 | 5.13 | 98.94 | 5.63 | 98.70 | 16.43 | 96.91 |
| | DAVIS(μ,σ) + DICE | 6.84 | 98.77 | 29.76 | 93.86 | 1.58 | 99.59 | 9.57 | 98.25 | 0.55 | 99.86 | 1.60 | 99.57 | 8.32 | 98.32 |
| MobileNet-v2 | MSP | 72.84 | 88.69 | 70.65 | 85.63 | 64.16 | 88.78 | 64.10 | 88.59 | 61.30 | 92.00 | 63.92 | 88.90 | 66.16 | 88.76 |
| | ODIN | 71.72 | 85.71 | 46.44 | 90.67 | 21.17 | 96.66 | 44.50 | 91.86 | 6.94 | 98.65 | 20.76 | 96.67 | 35.25 | 93.37 |
| | Energy | 75.83 | 85.85 | 44.98 | 90.62 | 29.68 | 95.03 | 48.67 | 91.19 | 9.54 | 98.12 | 29.80 | 95.10 | 39.75 | 92.65 |
| | ReAct | 73.06 | 85.65 | 45.30 | 90.21 | 28.00 | 95.12 | 44.08 | 92.15 | 11.31 | 97.84 | 27.31 | 95.31 | 38.18 | 92.71 |
| | DICE | 62.13 | 87.07 | 50.33 | 89.30 | 27.68 | 95.70 | 49.57 | 90.26 | 2.18 | 99.48 | 29.27 | 95.45 | 36.86 | 92.88 |
| | ReAct+DICE | 57.05 | 87.51 | 52.54 | 88.35 | 23.84 | 96.17 | 44.52 | 92.06 | 2.39 | 99.48 | 23.85 | 96.11 | 34.03 | 93.28 |
| | ASH | 46.23 | 91.48 | 61.75 | 84.40 | 41.46 | 93.36 | 43.87 | 91.50 | 7.28 | 98.68 | 40.97 | 93.46 | 40.26 | 92.15 |
| | DAVIS(m) | 69.61 | 87.56 | 44.33 | 91.00 | 23.33 | 96.26 | 37.87 | 94.11 | 17.62 | 97.02 | 22.48 | 96.40 | 35.87 | 93.73 |
| | DAVIS(m) + DICE | 38.71 | 92.48 | 50.33 | 89.34 | 14.89 | 97.42 | 25.05 | 95.84 | 4.39 | 99.21 | 15.28 | 97.40 | 24.78 | 95.28 |
| | DAVIS(μ,σ) | 62.07 | 89.18 | 43.87 | 90.86 | 22.38 | 96.40 | 33.24 | 94.71 | 14.21 | 97.50 | 21.32 | 96.57 | 32.85 | 94.20 |
| | DAVIS(μ,σ) + DICE | 37.14 | 92.79 | 50.38 | 89.05 | 15.39 | 97.34 | 24.45 | 96.00 | 3.56 | 99.32 | 15.38 | 97.34 | 24.38 | 95.31 |

Table 4: Detailed results on six common OOD benchmark datasets: SVHN, Places365, iSUN, Textures, LSUN-crop, LSUN-resize. We used same model pre-trained on CIFAR-100. ↓ indicates lower values are better and ↑ indicates larger values are better.

| Model | Method | SVHN FPR95↓ | SVHN AUROC↑ | Place365 FPR95↓ | Place365 AUROC↑ | iSUN FPR95↓ | iSUN AUROC↑ | Textures FPR95↓ | Textures AUROC↑ | LSUN-c FPR95↓ | LSUN-c AUROC↑ | LSUN-r FPR95↓ | LSUN-r AUROC↑ | Average FPR95↓ | Average AUROC↑ |
|---|---|---|---|---|---|---|---|---|---|---|---|---|---|---|---|
| ResNet-18 | MSP | 74.26 | 83.20 | 82.49 | 75.32 | 85.58 | 70.20 | 84.89 | 74.02 | 70.79 | 82.78 | 84.36 | 71.45 | 80.40 | 76.16 |
| | ODIN | 70.30 | 88.06 | 80.14 | 77.02 | 60.26 | 86.98 | 81.56 | 76.56 | 47.73 | 91.84 | 56.35 | 88.23 | 66.06 | 84.78 |
| | Energy | 66.64 | 89.53 | 81.23 | 76.84 | 73.67 | 82.01 | 85.30 | 75.68 | 48.01 | 91.63 | 70.30 | 83.38 | 70.86 | 83.18 |
| | ReAct | 55.03 | 91.95 | 79.78 | 77.48 | 58.66 | 87.78 | 60.90 | 87.94 | 47.42 | 91.22 | 54.78 | 88.77 | 59.43 | 87.52 |
| | DICE | 41.18 | 92.98 | 81.82 | 76.02 | 66.22 | 84.20 | 75.50 | 76.27 | 12.21 | 97.70 | 64.48 | 85.15 | 56.90 | 85.39 |
| | ReAct+DICE | 36.18 | 93.65 | 86.43 | 71.92 | 59.57 | 88.43 | 48.46 | 87.59 | 11.78 | 97.59 | 60.22 | 88.45 | 50.44 | 87.94 |
| | ASH | 29.10 | 95.46 | 82.56 | 75.38 | 67.09 | 85.01 | 56.49 | 87.80 | 27.06 | 95.55 | 64.72 | 85.62 | 54.50 | 87.47 |
| | DAVIS(m) | 38.40 | 94.25 | 77.62 | 78.98 | 56.43 | 89.99 | 55.73 | 89.14 | 35.90 | 94.17 | 55.87 | 89.95 | 53.32 | 89.41 |
| | DAVIS(m) + DICE | 10.77 | 97.91 | 80.06 | 77.59 | 33.36 | 94.13 | 31.52 | 93.52 | 7.31 | 98.53 | 37.27 | 93.39 | 33.38 | 92.51 |
| | DAVIS(μ,σ) | 42.01 | 93.75 | 78.99 | 77.58 | 61.13 | 88.60 | 59.89 | 87.89 | 38.56 | 93.48 | 59.76 | 88.86 | 56.72 | 88.36 |
| | DAVIS(μ,σ) + DICE | 12.40 | 97.59 | 80.46 | 76.21 | 38.77 | 92.69 | 36.74 | 92.09 | 7.78 | 98.42 | 40.98 | 92.27 | 36.19 | 91.54 |
| ResNet-34 | MSP | 69.72 | 83.11 | 82.28 | 75.84 | 83.04 | 76.64 | 84.95 | 74.24 | 76.57 | 81.44 | 81.54 | 77.19 | 79.68 | 78.08 |
| | ODIN | 70.15 | 86.07 | 80.97 | 77.27 | 57.90 | 88.77 | 83.23 | 76.93 | 57.77 | 89.70 | 54.97 | 89.54 | 67.50 | 84.71 |
| | Energy | 57.79 | 89.80 | 81.17 | 77.25 | 71.83 | 84.14 | 86.77 | 85.82 | 55.56 | 89.92 | 68.70 | 84.93 | 70.30 | 83.64 |
| | ReAct | 30.77 | 94.47 | 77.91 | 78.11 | 64.45 | 85.01 | 62.13 | 86.27 | 47.86 | 90.72 | 64.13 | 85.24 | 57.87 | 86.64 |
| | DICE | 25.88 | 95.11 | 80.75 | 77.18 | 65.76 | 85.44 | 74.73 | 78.11 | 18.31 | 96.55 | 65.59 | 85.30 | 55.17 | 86.28 |
| | ReAct+DICE | 20.70 | 95.78 | 89.29 | 69.02 | 81.03 | 78.47 | 49.77 | 85.31 | 26.09 | 94.34 | 86.08 | 76.00 | 58.83 | 83.16 |
| | ASH | 23.64 | 95.92 | 82.37 | 75.92 | 60.77 | 87.53 | 59.08 | 87.60 | 41.57 | 93.06 | 61.41 | 87.24 | 54.81 | 87.88 |
| | DAVIS(m) | 29.56 | 95.19 | 77.68 | 78.81 | 55.37 | 90.51 | 56.29 | 89.14 | 38.37 | 93.37 | 52.26 | 90.23 | 52.26 | 89.54 |
| | DAVIS(m) + DICE | 7.96 | 98.40 | 79.29 | 77.25 | 36.80 | 93.62 | 28.32 | 94.05 | 8.62 | 98.22 | 42.47 | 92.46 | 33.91 | 92.33 |
| | DAVIS(μ,σ) | 31.07 | 95.00 | 80.04 | 77.53 | 59.46 | 89.50 | 59.75 | 88.31 | 41.21 | 93.00 | 60.05 | 89.32 | 55.26 | 88.78 |
| | DAVIS(μ,σ) + DICE | 8.25 | 98.43 | 81.22 | 76.35 | 41.29 | 92.83 | 33.12 | 93.27 | 9.78 | 98.15 | 46.38 | 91.82 | 36.67 | 91.81 |
| DenseNet-101 | MSP | 81.38 | 75.71 | 82.62 | 74.04 | 84.12 | 68.22 | 86.95 | 68.37 | 81.82 | 87.93 | 78.04 | 69.51 | 82.49 | 73.96 |
| | ODIN | 85.94 | 80.35 | 75.59 | 77.62 | 48.03 | 89.12 | 83.37 | 67.83 | 12.78 | 97.70 | 57.67 | 91.35 | 60.56 | 84.00 |
| | Energy | 70.99 | 86.66 | 77.12 | 76.94 | 64.28 | 83.92 | 83.60 | 67.47 | 11.45 | 97.89 | 60.59 | 86.84 | 61.34 | 83.29 |
| | ReAct | 67.12 | 87.20 | 77.75 | 76.18 | 56.39 | 89.46 | 75.98 | 79.16 | 13.26 | 97.53 | 56.74 | 90.94 | 57.87 | 86.74 |
| | DICE | 33.87 | 93.97 | 79.95 | 76.75 | 47.76 | 89.61 | 63.42 | 73.33 | 0.79 | 99.76 | 44.91 | 91.00 | 45.12 | 87.40 |
| | ReAct+DICE | 28.01 | 95.38 | 83.56 | 74.73 | 37.02 | 93.92 | 46.93 | 86.09 | 0.68 | 99.79 | 38.90 | 93.93 | 39.18 | 90.64 |
| | ASH | 10.32 | 97.99 | 85.93 | 71.95 | 39.69 | 92.04 | 35.67 | 91.76 | 5.43 | 98.98 | 36.66 | 91.30 | 35.62 | 90.67 |
| | DAVIS(m) | 52.32 | 89.41 | 73.49 | 79.62 | 35.76 | 93.37 | 49.88 | 88.07 | 18.58 | 96.44 | 44.55 | 92.98 | 45.76 | 89.98 |
| | DAVIS(m) + DICE | 27.97 | 94.91 | 76.61 | 76.61 | 36.01 | 93.97 | 30.32 | 93.28 | 7.58 | 98.56 | 38.32 | 93.13 | 36.14 | 91.74 |
| | DAVIS(μ,σ) | 45.23 | 91.91 | 72.11 | 79.78 | 42.21 | 92.16 | 55.85 | 85.47 | 13.92 | 97.53 | 44.97 | 92.45 | 45.72 | 89.88 |
| | DAVIS(μ,σ) + DICE | 20.30 | 96.20 | 79.52 | 78.28 | 29.69 | 94.63 | 34.10 | 91.24 | 2.63 | 99.37 | 33.04 | 94.20 | 33.21 | 92.32 |
| MobileNet-v2 | MSP | 80.14 | 76.09 | 84.23 | 72.62 | 87.72 | 70.77 | 86.67 | 70.36 | 77.11 | 76.04 | 87.10 | 70.83 | 83.83 | 72.78 |
| | ODIN | 82.16 | 80.95 | 80.06 | 76.66 | 65.51 | 86.71 | 77.61 | 80.78 | 50.14 | 89.57 | 65.15 | 87.14 | 70.10 | 83.63 |
| | Energy | 69.65 | 85.98 | 81.26 | 75.21 | 78.17 | 83.33 | 80.02 | 78.63 | 50.19 | 89.42 | 76.59 | 84.06 | 72.65 | 82.77 |
| | ReAct | 28.41 | 95.15 | 79.46 | 74.05 | 62.45 | 87.65 | 47.70 | 90.19 | 41.25 | 92.31 | 62.16 | 88.06 | 53.57 | 87.90 |
| | DICE | 55.62 | 87.54 | 83.25 | 74.25 | 81.23 | 79.75 | 65.02 | 81.93 | 18.17 | 96.24 | 85.40 | 77.90 | 64.78 | 82.93 |
| | ReAct+DICE | 22.57 | 96.05 | 89.57 | 66.14 | 81.05 | 76.71 | 29.54 | 93.30 | 10.19 | 98.00 | 89.71 | 71.34 | 53.77 | 83.59 |
| | ASH | 21.90 | 96.46 | 85.12 | 69.51 | 70.46 | 82.84 | 34.80 | 92.65 | 24.14 | 95.56 | 73.46 | 81.22 | 51.65 | 86.37 |
| | DAVIS(m) | 41.94 | 92.00 | 81.49 | 73.31 | 64.52 | 86.40 | 44.22 | 90.68 | 34.22 | 93.72 | 65.30 | 86.10 | 55.28 | 87.04 |
| | DAVIS(m) + ASH | 11.62 | 97.72 | 87.30 | 66.62 | 65.66 | 83.89 | 22.04 | 95.40 | 20.60 | 96.20 | 70.72 | 81.92 | 46.32 | 86.96 |
| | DAVIS(μ,σ) | 40.00 | 92.73 | 81.64 | 73.23 | 64.99 | 86.36 | 42.70 | 91.25 | 34.67 | 93.75 | 66.30 | 86.04 | 55.05 | 87.23 |
| | DAVIS(μ,σ) + ASH | 12.22 | 97.74 | 86.54 | 67.52 | 65.85 | 84.13 | 22.46 | 95.43 | 20.60 | 96.28 | 70.41 | 82.28 | 46.35 | 87.23 |

## C.2 IMAGENET EVALUATION

Table 5 showcases detailed evaluation on ImageNet benchmark, using broad pre-trained model DenseNet-121, ResNet-50, MobileNet-v2, and EfficientNet-B0, for which we re-evaluated all baselines to ensure a fair comparison. Since results for DenseNet-121 and EfficientNet-b0 were not available in the original publications of foundational baselines (e.g., ReAct, DICE, ASH), we rigorously re-evaluated these methods ourselves. To ensure a fair and direct comparison, we carefully followed the hyperparameter selection protocols described in their respective papers.

Table 5: *OOD detection results on ImageNet benchmarks. All values are percentages and are averaged over four common OOD benchmark datasets. The foundational methods we compare against (e.g., MSP, Energy, ReAct, DICE) did not report results for a DenseNet-121 and EfficientNet-b0 model in their original publications. Therefore, we re-evaluated these baselines ourselves, carefully following the hyper-parameter selection guidelines from their respective papers to ensure a fair comparison. Here,* **ReDi** *represents Re-Act+DICE. The symbol ↓ indicates lower values are better; ↑ indicates larger values are better.*

| Model | Method | SUN | | Places365 | | Texture | | iNaturalist | | Average | |
|---|---|---|---|---|---|---|---|---|---|---|---|
| | | FPR95 ↓ | AUROC ↑ | FPR95 ↓ | AUROC ↑ | FPR95 ↓ | AUROC ↑ | FPR95 ↓ | AUROC ↑ | FPR95 ↓ | AUROC ↑ |
| DenseNet-121 | MSP | 67.49 | 81.41 | 69.53 | 80.95 | 67.23 | 79.18 | 49.58 | 89.05 | 63.46 | 82.65 |
| | ODIN | 54.13 | 86.33 | 60.39 | 84.14 | 50.82 | 85.81 | 32.47 | 93.66 | 49.45 | 87.48 |
| | Energy | 52.51 | 87.27 | 58.24 | 85.05 | 52.22 | 85.42 | 39.75 | 92.66 | 50.68 | 87.60 |
| | ReAct | 43.65 | 90.65 | 51.05 | 87.54 | 43.48 | 90.81 | 25.74 | 95.05 | 40.98 | 91.01 |
| | DICE | 38.75 | 89.91 | 49.29 | 86.24 | 40.85 | 88.09 | 25.78 | 94.37 | 38.67 | 89.65 |
| | ReAct+DICE | 48.98 | 86.74 | 61.50 | 80.88 | 37.61 | 91.11 | 34.73 | 91.67 | 45.70 | 87.60 |
| | ASH | 37.20 | 91.51 | 46.54 | 88.79 | 21.76 | 95.04 | 15.50 | 97.03 | 30.25 | 93.09 |
| | DAVIS($m$) | 48.55 | 88.03 | 59.36 | 83.53 | 27.54 | 93.62 | 33.74 | 92.20 | 42.30 | 89.34 |
| | DAVIS($m$) + ASH | 38.10 | 91.39 | 50.52 | 87.38 | 13.23 | 97.37 | 21.35 | 95.72 | 30.80 | 92.97 |
| | DAVIS($\mu,\sigma$) | 48.40 | 88.64 | 58.38 | 84.94 | 25.37 | 94.14 | 30.31 | 93.79 | 40.62 | 90.38 |
| | DAVIS($\mu,\sigma$) + ASH | 35.89 | 91.98 | 46.66 | 88.97 | 15.92 | 96.63 | 15.30 | 97.04 | 28.44 | 93.66 |
| ResNet-50 | MSP | 69.11 | 81.64 | 72.06 | 80.54 | 66.26 | 80.43 | 52.83 | 88.39 | 65.06 | 82.75 |
| | ODIN | 60.15 | 84.59 | 67.89 | 81.78 | 50.23 | 85.62 | 47.66 | 89.66 | 56.48 | 85.41 |
| | Energy | 59.26 | 85.89 | 64.92 | 82.86 | 53.72 | 85.99 | 55.72 | 89.95 | 58.41 | 86.17 |
| | ReAct | 24.20 | 94.20 | 33.85 | 91.58 | 47.30 | 89.80 | 20.38 | 96.22 | 31.43 | 92.95 |
| | DICE | 35.15 | 90.83 | 46.49 | 87.48 | 31.72 | 90.30 | 25.63 | 94.49 | 34.75 | 90.78 |
| | ReAct+DICE | 26.46 | 93.88 | 37.99 | 90.74 | 29.26 | 92.69 | 19.85 | 96.16 | 28.39 | 93.37 |
| | ASH | 27.98 | 94.02 | 39.78 | 90.98 | 11.93 | 97.60 | 11.49 | 97.87 | 22.80 | 95.12 |
| | DAVIS($m$) | 53.12 | 85.49 | 63.58 | 80.54 | 20.94 | 95.09 | 36.11 | 91.29 | 43.44 | 88.10 |
| | DAVIS($m$) + ASH | 29.89 | 93.61 | 43.08 | 89.89 | 9.22 | 98.26 | 15.25 | 97.21 | 24.36 | 94.74 |
| | DAVIS($\mu,\sigma$) | 54.71 | 86.47 | 64.06 | 82.55 | 21.13 | 95.32 | 34.50 | 92.95 | 43.60 | 89.33 |
| | DAVIS($\mu,\sigma$) + ASH | 27.67 | 94.02 | 40.01 | 90.73 | 9.91 | 98.07 | 11.69 | 97.81 | 22.32 | 95.16 |
| MobileNet-v2 | MSP | 74.28 | 78.81 | 76.65 | 78.10 | 71.05 | 78.93 | 59.70 | 86.72 | 70.42 | 80.64 |
| | ODIN | 54.07 | 85.88 | 57.36 | 84.71 | 49.96 | 85.03 | 55.39 | 87.62 | 54.20 | 85.81 |
| | Energy | 59.60 | 86.16 | 66.36 | 83.15 | 54.82 | 86.57 | 55.33 | 90.37 | 59.03 | 86.56 |
| | ReAct | 52.68 | 87.21 | 59.81 | 84.04 | 40.32 | 90.96 | 42.94 | 92.75 | 48.94 | 88.74 |
| | DICE | 38.81 | 90.46 | 52.95 | 85.82 | 33.00 | 91.27 | 42.95 | 90.87 | 41.93 | 89.60 |
| | ReAct+DICE | 30.66 | 92.97 | 46.06 | 88.30 | 15.99 | 96.32 | 31.32 | 93.80 | 31.01 | 92.85 |
| | ASH | 43.86 | 89.98 | 58.92 | 84.72 | 13.21 | 97.10 | 39.13 | 91.96 | 38.78 | 90.94 |
| | DAVIS($m$) | 53.64 | 87.26 | 63.61 | 82.93 | 25.87 | 94.03 | 40.25 | 92.22 | 45.84 | 89.11 |
| | DAVIS($m$) + ReDi | 33.82 | 89.43 | 47.57 | 84.27 | 18.85 | 93.97 | 26.22 | 93.68 | 31.61 | 90.34 |
| | DAVIS($\mu,\sigma$) | 55.47 | 87.15 | 64.82 | 83.18 | 25.78 | 94.33 | 38.94 | 92.98 | 46.25 | 89.41 |
| | DAVIS($\mu,\sigma$) + ReDi | 24.90 | 93.44 | 38.64 | 89.18 | 13.46 | 96.12 | 20.79 | 95.84 | 24.45 | 93.65 |
| EfficientNet-b0 | MSP | 72.56 | 80.06 | 74.07 | 79.17 | 66.83 | 81.19 | 57.42 | 87.28 | 67.72 | 81.92 |
| | ODIN | 72.57 | 75.74 | 77.87 | 73.20 | 64.96 | 79.15 | 59.01 | 83.99 | 68.60 | 78.02 |
| | Energy | 85.01 | 72.86 | 86.06 | 70.99 | 75.99 | 75.86 | 78.91 | 79.78 | 81.49 | 74.87 |
| | ReAct | 72.96 | 80.97 | 77.59 | 77.61 | 34.57 | 92.44 | 55.19 | 89.82 | 60.08 | 85.21 |
| | DICE | 98.15 | 44.65 | 99.17 | 39.44 | 93.83 | 59.29 | 99.66 | 39.80 | 97.70 | 45.79 |
| | ReAct+DICE | 93.93 | 57.49 | 97.00 | 49.68 | 61.86 | 81.18 | 95.55 | 56.92 | 87.09 | 61.32 |
| | ASH | 98.62 | 52.65 | 99.30 | 48.33 | 97.64 | 67.20 | 99.78 | 53.49 | 98.84 | 55.42 |
| | DAVIS($m$) | 69.42 | 80.74 | 79.09 | 73.67 | 15.82 | 96.38 | 63.68 | 80.63 | 57.00 | 82.85 |
| | DAVIS($m$) + ASH | 67.84 | 80.49 | 79.26 | 72.78 | 14.98 | 96.51 | 63.21 | 81.58 | 56.32 | 82.84 |
| | DAVIS($\mu,\sigma$) | 61.85 | 84.67 | 71.31 | 79.01 | 12.18 | 97.52 | 48.25 | 88.35 | 48.40 | 87.39 |
| | DAVIS($\mu,\sigma$) + ASH | 61.11 | 84.81 | 72.70 | 78.53 | 10.02 | 97.94 | 47.76 | 88.97 | 47.90 | 87.56 |

## C.3 Near-OOD Evaluation

We analyze `DAVIS` on the challenging near-OOD task of distinguishing CIFAR-10 from CIFAR-100 (Ghosal et al., 2024). As shown in Table 6, while the separation is inherently more difficult, our method still provides consistent improvements over all baselines, reducing the FPR95 by 7.74%, 6.29%, 1.70%, and 5.89% with ResNet-18, ResNet-34, DenseNet-101, and MobileNet-v2, respectively. Additionally, for DenseNet-101, `DAVIS`$(\mu, \sigma)$ + `DICE` achieves a further improvement of 4.50%.

Table 6: *Near-OOD detection evaluation. CIFAR-10 is ID dataset and CIFAR-100 is OOD dataset. The symbol ↓ indicates lower values are better; ↑ indicates higher values are better.*

| Method | ResNet-18 | | ResNet-34 | | DenseNet-101 | | MobileNet-v2 | |
|---|---|---|---|---|---|---|---|---|
| | FPR95 ↓ | AUROC ↑ | FPR95 ↓ | AUROC ↑ | FPR95 ↓ | AUROC ↑ | FPR95 ↓ | AUROC ↑ |
| MSP | 68.85 | 88.17 | 65.79 | 88.47 | 63.49 | 88.53 | 71.62 | 86.12 |
| ODIN | 51.79 | 90.26 | 48.92 | 90.31 | 47.73 | 90.36 | 55.82 | 88.63 |
| Energy | 52.32 | 90.14 | 50.85 | 90.39 | 48.86 | 90.29 | 57.35 | 88.49 |
| ReAct | 52.04 | 90.42 | 51.32 | 90.38 | 47.60 | 90.72 | 55.18 | 88.72 |
| DICE | 56.56 | 89.12 | 55.74 | 89.41 | 53.10 | 88.91 | 63.73 | 86.39 |
| ReAct+DICE | 57.43 | 88.92 | 56.98 | 89.31 | 53.20 | 89.20 | 63.02 | 86.46 |
| ASH | 51.54 | 90.08 | 50.25 | 89.82 | 47.16 | 90.34 | 57.08 | 88.22 |
| `DAVIS`$(m)$ | 47.82 | 90.21 | 47.56 | 90.29 | 50.06 | 89.37 | 56.05 | 88.48 |
| `DAVIS`$(m)$ + ASH | **47.08** | 90.31 | 47.64 | 90.19 | 48.17 | 89.73 | 54.24 | 88.37 |
| `DAVIS`$(\mu, \sigma)$ | 49.24 | 90.62 | 47.68 | **90.74** | 47.30 | 90.64 | 54.33 | 88.87 |
| `DAVIS`$(\mu, \sigma)$ + ASH | 47.71 | **90.69** | **47.09** | 90.65 | **46.36** | 90.73 | **53.72** | **88.59** |

# D Comparison with Other Baselines

While in the main paper we restrict our comparison to foundational representative techniques (i.e., MSP, ODIN, Energy, ReAct, DICE, and ASH), we provide a comparison of our method, `DAVIS`, with additional baselines fDBD (Liu & Qin, 2024), SCALE (Xu et al., 2024), and NCI (Liu & Qin, 2025) in this section. A comprehensive re-evaluation of fDBD and NCI across all architectures used in our study was determined to be beyond the scope of this work due to a fundamental difference in their design philosophy.

Methods like ReAct, DICE, and our own `DAVIS` are modular, post-hoc techniques that primarily modify the penultimate feature vector itself. In contrast, fDBD and NCI introduce entirely new scoring functions derived from the geometric relationship between features and the classifier's decision boundaries (fDBD) or class weight vectors (NCI). Integrating our feature-level modifications into these structurally different scoring frameworks would require significant, non-trivial engineering and could obscure a direct comparison. Therefore, for these two methods, we present a fair comparison limited to the overlapping architectures and datasets from their original publications.

## D.1 Neural Collapse Inspired (NCI) OOD Detector

As shown in Table 7 for CIFAR-10 and Table 8 for ImageNet, in a direct comparison against NCI's (Liu & Qin, 2025) reported results, our method `DAVIS` demonstrates a clear and significant advantage. On CIFAR-10 with a ResNet-18 backbone, `DAVIS`$(m)$ + `DICE` decisively outperforms NCI, reducing the average FPR95 by 43.90%. This strong performance is maintained on the large-scale ImageNet benchmark, where our method reduces the FPR95 by 43.30% on a ResNet-50.

Table 7: *A direct comparison of* DAVIS *against the NCI baseline, using their originally reported results for* **CIFAR-10** *with a* **ResNet-18** *backbone. The evaluation is restricted to the SVHN, Texture, and Places365 OOD datasets to ensure a fair comparison that matches the protocol from the original NCI paper.*

| Method | SVHN | | Places365 | | Texture | | Average | |
|---|---|---|---|---|---|---|---|---|
| | FPR95 ↓ | AUROC ↑ | FPR95 ↓ | AUROC ↑ | FPR95 ↓ | AUROC ↑ | FPR95 ↓ | AUROC ↑ |
| NCI | 28.92 | 90.81 | 34.01 | 90.74 | 26.53 | 92.18 | 29.82 | 91.24 |
| DAVIS$(m)$ | 19.81 | 96.29 | 32.32 | 93.73 | 21.90 | 96.44 | 24.68 | 95.49 |
| DAVIS$(m)$ + DICE | 7.95 | 98.50 | **30.55** | **93.97** | **9.66** | **98.28** | **16.72** | 96.25 |
| DAVIS$(\mu,\sigma)$ | 19.83 | 96.47 | 36.11 | 93.16 | 26.77 | 95.89 | 27.57 | 95.17 |
| DAVIS$(\mu,\sigma)$ + DICE | **7.85** | **98.57** | 35.27 | 93.18 | 14.04 | 97.71 | 19.72 | **96.49** |

Table 8: *A direct comparison of* DAVIS *against the NCI baseline, using their originally reported results for* **ImageNet** *with a* **ResNet-50** *backbone. The evaluation is restricted to the iNaturalist and Texture OOD datasets to ensure a fair comparison that matches the protocol from the original NCI paper.*

| Method | Texture | | iNaturalist | | Average | |
|---|---|---|---|---|---|---|
| | FPR95 ↓ | AUROC ↑ | FPR95 ↓ | AUROC ↑ | FPR95 ↓ | AUROC ↑ |
| NCI | 23.79 | 96.63 | 14.31 | 96.95 | 19.05 | 96.79 |
| DAVIS$(m)$ | 20.94 | 95.09 | 36.11 | 91.29 | 28.53 | 93.19 |
| DAVIS$(m)$ + ASH | **9.22** | **98.26** | 15.25 | 97.21 | 12.24 | 97.74 |
| DAVIS$(\mu,\sigma)$ | 21.13 | 95.32 | 34.50 | 92.95 | 27.82 | 94.14 |
| DAVIS$(\mu,\sigma)$ + ASH | 9.91 | 98.07 | **11.69** | **97.81** | **10.80** | **97.94** |

## D.2 FAST DECISION BOUNDARY BASED OOD DETECTOR

As shown in Tables 9 and 10, our method, DAVIS, demonstrates a decisive and substantial performance advantage over the fDBD's reported results in all comparable, overlapping settings. The strength of DAVIS is most apparent when it is composed with existing techniques, creating a powerful synergistic effect that dramatically improves OOD detection. On CIFAR-10, this combination is particularly effective. Using a ResNet-18, DAVIS$(m)$ + DICE slashes the average FPR95 by 55.9% relative to fDBD (from 31.09% down to 13.72%). The gains are even more pronounced on a DenseNet-101, where DAVIS$(\mu,\sigma)$ + DICE achieves an FPR95 reduction of 38.20%.

Table 9: *Direct comparison of* DAVIS *against the fDBD baseline on* **CIFAR-10**. *To ensure a fair comparison, the evaluation is restricted to the four OOD datasets reported in the original fDBD paper: SVHN, Places365, iSUN, and Texture.*

| Model | Method | SVHN | | Places365 | | iSUN | | Texture | | Average | |
|---|---|---|---|---|---|---|---|---|---|---|---|
| | | FPR95 ↓ | AUROC ↑ | FPR95 ↓ | AUROC ↑ | FPR95 ↓ | AUROC ↑ | FPR95 ↓ | AUROC ↑ | FPR95 ↓ | AUROC ↑ |
| **ResNet-18** | fDBD | 22.58 | 96.07 | 46.59 | 90.40 | 23.96 | 95.85 | 31.24 | 94.48 | 31.09 | 94.20 |
| | DAVIS$(m)$ | 19.81 | 96.29 | 32.32 | 93.73 | 15.79 | 97.37 | 21.90 | 96.44 | 22.46 | 95.96 |
| | DAVIS$(m)$ + DICE | 7.95 | 98.50 | 30.55 | 93.97 | 6.70 | 98.59 | 9.66 | 98.28 | 13.72 | 97.34 |
| | DAVIS$(\mu,\sigma)$ | 19.83 | 96.47 | 36.11 | 93.16 | 20.48 | 96.90 | 26.77 | 95.89 | 25.80 | 95.61 |
| | DAVIS$(\mu,\sigma)$ + DICE | **7.85** | **98.57** | 35.27 | 93.18 | 11.96 | 97.96 | 14.04 | 97.71 | 17.78 | 96.86 |
| **DenseNet-101** | fDBD | 5.89 | 98.67 | 39.52 | 91.53 | 5.90 | 98.75 | 22.75 | 95.81 | 18.52 | 96.19 |
| | DAVIS$(m)$ | 30.50 | 94.50 | 33.04 | 93.28 | 7.81 | 98.29 | 25.16 | 95.85 | 24.13 | 95.48 |
| | DAVIS$(m)$ + DICE | 8.30 | 98.37 | 29.47 | 93.92 | 1.85 | 99.56 | 7.16 | 98.69 | 11.70 | 97.64 |
| | DAVIS$(\mu,\sigma)$ | 24.75 | 95.93 | 32.08 | 93.43 | 5.77 | 98.64 | 25.23 | 95.84 | 24.46 | 95.96 |
| | DAVIS$(\mu,\sigma)$ + DICE | **6.84** | **98.77** | 29.76 | 93.86 | **1.58** | **99.59** | 9.57 | 98.25 | **11.44** | 97.62 |

As demonstrated in Table 10, this commanding performance extends to the large-scale ImageNet benchmark. While fDBD struggles with a high average FPR95 of 51.19%, our DAVIS$(\mu, \sigma)$+ASH achieves an FPR95 of just 22.32% – a massive 56.40% relative reduction. These results validate that by first enriching the feature representation with more discriminative statistics, DAVIS enables subsequent methods to operate far more effectively, establishing a new state-of-the-art over the fDBD baseline.

Table 10: *This table presents a direct comparison of* DAVIS *against the fDBD baseline on **ImageNet** using a **ResNet-50** backbone. To ensure a fair comparison, the evaluation is restricted to the iNaturalist and Texture OOD datasets, matching the protocol in the original fDBD paper.*

| Method | SUN | | Places365 | | Texture | | iNaturalist | | Average | |
|---|---|---|---|---|---|---|---|---|---|---|
| | FPR95 ↓ | AUROC ↑ | FPR95 ↓ | AUROC ↑ | FPR95 ↓ | AUROC ↑ | FPR95 ↓ | AUROC ↑ | FPR95 ↓ | AUROC ↑ |
| fDBD | 60.60 | 86.97 | 66.40 | 84.27 | 37.50 | 92.12 | 40.24 | 93.67 | 51.19 | 89.26 |
| DAVIS$(m)$ | 53.12 | 85.49 | 63.58 | 80.54 | 20.94 | 95.09 | 36.11 | 91.29 | 43.44 | 88.10 |
| DAVIS$(m)$ + ASH | 29.89 | 93.61 | 43.08 | 89.89 | **9.22** | **98.26** | 15.25 | 97.21 | 24.36 | 94.74 |
| DAVIS$(\mu, \sigma)$ | 54.71 | 86.47 | 64.06 | 82.55 | 21.13 | 95.32 | 34.50 | 92.95 | 43.60 | 89.33 |
| DAVIS$(\mu, \sigma)$ + ASH | **27.67** | **94.02** | **40.01** | **90.73** | 9.91 | 98.07 | **11.69** | **97.81** | **22.32** | **95.16** |

### D.3 SCALE OOD DETECTOR

The original evaluation of SCALE was limited to a single architecture for each benchmark (DenseNet-101 for CIFAR and ResNet-50 for ImageNet). To provide a more comprehensive comparison, we extended their evaluation across all architectures used in our study. Our expanded results in Table 11 show that composing DAVIS with SCALE yields consistent FPR95 reductions on ImageNet across standard backbones, including 7.44% on DenseNet-121, 9.96% on ResNet-50, and 8.26% on MobileNet-v2. This synergy is most critical on EfficientNet-b0, where the SCALE baseline completely fails (98.56% FPR95), while our combined method rescues its performance, achieving a functional 61.50% FPR95.

Table 11: *Demonstrating the synergy of* DAVIS *by showing its enhancement of **Scale** baseline on the **ImageNet** benchmark. Adding* DAVIS *provides consistent performance gains across all tested architectures, reducing the average FPR95 by up to 7.45% on DenseNet-121.*

| Model | Method | SUN | | Places365 | | Texture | | iNaturalist | | Average | |
|---|---|---|---|---|---|---|---|---|---|---|---|
| | | FPR95 ↓ | AUROC ↑ | FPR95 ↓ | AUROC ↑ | FPR95 ↓ | AUROC ↑ | FPR95 ↓ | AUROC ↑ | FPR95 ↓ | AUROC ↑ |
| DenseNet-121 | Scale | 33.85 | 92.16 | 42.92 | 89.62 | 22.27 | 94.63 | 13.21 | 97.40 | 28.06 | 93.45 |
| | DAVIS$(m)$ + Scale | 33.68 | 92.03 | 45.81 | 88.31 | **13.30** | **97.24** | 18.13 | 96.23 | 27.73 | 93.45 |
| | DAVIS$(\mu, \sigma)$ + Scale | **32.45** | **92.65** | **42.41** | **89.84** | 16.29 | 96.40 | **12.71** | **97.45** | **25.97** | **94.09** |
| ResNet-50 | Scale | 25.78 | 94.54 | 36.86 | 91.96 | 14.56 | 96.75 | 10.37 | 98.02 | 21.89 | 95.32 |
| | DAVIS$(m)$ + Scale | 25.38 | 94.55 | 37.51 | 91.09 | **10.48** | **97.95** | 11.94 | 97.75 | 21.33 | 95.34 |
| | DAVIS$(\mu, \sigma)$ + Scale | **23.42** | **94.93** | **34.61** | **91.98** | 11.37 | 97.74 | **9.46** | **98.15** | **19.71** | **95.70** |
| MobileNet-v2 | Scale | 38.74 | 91.64 | 53.49 | 87.34 | 14.79 | 96.65 | 30.09 | 94.46 | 34.28 | 92.52 |
| | DAVIS$(m)$ + Scale | **37.47** | **91.91** | 52.72 | 86.90 | **8.85** | **98.20** | 26.76 | 95.08 | **31.45** | **93.02** |
| | DAVIS$(\mu, \sigma)$ + Scale | 37.89 | 91.82 | **52.92** | **87.24** | 11.37 | 97.60 | 28.54 | 94.76 | 32.68 | 92.86 |
| EfficientNet-b0 | Scale | 98.48 | 53.34 | 99.27 | 47.85 | 96.81 | 71.19 | 99.67 | 55.44 | 98.56 | 56.95 |
| | DAVIS$(m)$ + Scale | **72.89** | **76.83** | **82.74** | **68.50** | **21.35** | **95.31** | **69.02** | **80.74** | **61.50** | **80.35** |
| | DAVIS$(\mu, \sigma)$ + Scale | 86.53 | 68.48 | 92.42 | 60.52 | 38.19 | 91.34 | 83.31 | 74.47 | 75.11 | 73.70 |

In our CIFAR evaluation, as illustrated in Table 12, our method, DAVIS, consistently improves upon the SCALE baseline across all tested architectures. For instance, using a ResNet-18, DAVIS reduces the average FPR95 by 36.84% on CIFAR-10 and 22.07% on CIFAR-100. Given that SCALE is a derivative of ASH, this strong synergistic effect is consistent with our findings throughout this paper, highlighting the role of DAVIS as a powerful, complementary module for these methods.

Table 12: *Demonstrating the synergy of DAVIS by showing its enhancement of* `Scale` *baseline on the* **CIFAR** *benchmark. Adding DAVIS provides consistent performance gains across all tested architectures. ↓ indicates lower values are better and ↑ indicates larger values are better.*

| Dataset | Model | Method | SVHN | | Place365 | | iSUN | | Textures | | LSUN-c | | LSUN-r | | Average | |
|---|---|---|---|---|---|---|---|---|---|---|---|---|---|---|---|---|
| | | | FPR95↓ | AUROC↑ | FPR95↓ | AUROC↑ | FPR95↓ | AUROC↑ | FPR95↓ | AUROC↑ | FPR95↓ | AUROC↑ | FPR95↓ | AUROC↑ | FPR95↓ | AUROC↑ |
| CIFAR-10 | ResNet-18 | Scale | 9.73 | 98.13 | 45.99 | 90.87 | 22.92 | 96.39 | 27.00 | 95.61 | 3.75 | 99.17 | 21.02 | 96.60 | 21.74 | 96.13 |
| | | DAVIS$(m)$ + Scale | 10.21 | 98.17 | 37.09 | 92.92 | 9.74 | 98.17 | 12.77 | 97.83 | 2.86 | 99.39 | 9.71 | 98.19 | 13.73 | 97.45 |
| | | DAVIS$(\mu,\sigma)$ + Scale | 8.88 | 98.43 | 39.84 | 92.38 | 12.36 | 97.93 | 14.11 | 97.65 | 2.71 | 99.43 | 11.62 | 97.99 | 14.92 | 97.30 |
| | ResNet-34 | Scale | 17.47 | 97.01 | 38.01 | 91.84 | 15.06 | 97.31 | 28.40 | 94.86 | 2.37 | 99.44 | 15.67 | 97.25 | 19.50 | 96.28 |
| | | DAVIS$(m)$ + Scale | 20.13 | 96.60 | 33.58 | 93.37 | 10.20 | 98.07 | 17.61 | 97.01 | 3.95 | 99.23 | 10.92 | 97.99 | 16.06 | 97.05 |
| | | DAVIS$(\mu,\sigma)$ + Scale | 17.83 | 97.00 | 34.47 | 92.99 | 10.35 | 98.01 | 18.49 | 96.87 | 3.10 | 99.35 | 11.08 | 97.92 | 15.89 | 97.02 |
| | DenseNet-101 | Scale | 23.06 | 96.13 | 36.53 | 92.24 | 4.54 | 98.76 | 30.53 | 93.62 | 1.23 | 99.61 | 4.59 | 98.75 | 16.75 | 96.52 |
| | | DAVIS$(m)$ + Scale | 20.38 | 96.25 | 33.11 | 93.27 | 5.86 | 98.63 | 15.80 | 97.27 | 6.57 | 98.72 | 6.07 | 98.61 | 14.63 | 97.13 |
| | | DAVIS$(\mu,\sigma)$ + Scale | 14.93 | 97.30 | 31.94 | 93.46 | 4.04 | 98.95 | 15.74 | 97.31 | 3.28 | 99.26 | 4.28 | 98.95 | 12.37 | 97.54 |
| | MobileNet-v2 | Scale | 56.48 | 89.83 | 59.43 | 85.26 | 34.81 | 94.33 | 42.36 | 91.91 | 9.82 | 98.23 | 34.46 | 94.50 | 39.56 | 92.34 |
| | | DAVIS$(m)$ + Scale | 47.67 | 91.98 | 55.06 | 86.26 | 22.92 | 95.87 | 29.06 | 95.24 | 16.04 | 97.31 | 21.92 | 96.21 | 32.11 | 93.81 |
| | | DAVIS$(\mu,\sigma)$ + Scale | 45.04 | 92.63 | 57.52 | 85.64 | 23.88 | 95.72 | 28.62 | 95.35 | 14.31 | 97.59 | 22.63 | 96.12 | 32.00 | 93.84 |
| CIFAR-100 | ResNet-18 | Scale | 22.12 | 96.38 | 81.96 | 74.95 | 61.62 | 86.65 | 44.50 | 90.72 | 18.62 | 96.78 | 59.76 | 86.74 | 48.10 | 88.70 |
| | | DAVIS$(m)$ + Scale | 10.40 | 98.10 | 82.86 | 74.60 | 45.08 | 91.15 | 24.10 | 95.04 | 14.09 | 97.49 | 48.37 | 89.96 | 37.48 | 91.06 |
| | | DAVIS$(\mu,\sigma)$ + Scale | 11.43 | 97.94 | 83.24 | 74.00 | 47.37 | 90.57 | 25.64 | 94.78 | 15.35 | 97.28 | 49.16 | 89.69 | 38.70 | 90.71 |
| | ResNet-34 | Scale | 13.68 | 97.51 | 80.72 | 75.87 | 57.87 | 87.14 | 46.35 | 90.27 | 28.17 | 95.07 | 61.31 | 86.01 | 48.02 | 88.64 |
| | | DAVIS$(m)$ + Scale | 6.55 | 98.75 | 82.68 | 73.88 | 49.14 | 89.47 | 24.08 | 94.64 | 16.16 | 97.05 | 57.11 | 87.44 | 39.29 | 90.21 |
| | | DAVIS$(\mu,\sigma)$ + Scale | 6.34 | 98.76 | 82.73 | 73.83 | 49.69 | 89.44 | 24.95 | 94.48 | 17.77 | 96.79 | 56.90 | 87.62 | 39.73 | 90.15 |
| | DenseNet-101 | Scale | 16.26 | 97.05 | 78.54 | 76.97 | 43.56 | 91.21 | 45.60 | 87.23 | 3.23 | 99.30 | 42.69 | 91.02 | 38.31 | 90.46 |
| | | DAVIS$(m)$ + Scale | 19.45 | 96.40 | 81.18 | 76.81 | 36.21 | 93.26 | 25.11 | 94.71 | 11.07 | 97.97 | 44.32 | 91.81 | 36.22 | 91.83 |
| | | DAVIS$(\mu,\sigma)$ + Scale | 13.46 | 97.53 | 78.75 | 77.82 | 32.68 | 93.77 | 24.29 | 95.06 | 6.09 | 98.85 | 38.36 | 92.74 | 32.27 | 92.63 |
| | MobileNet-v2 | Scale | 22.36 | 96.25 | 81.46 | 73.30 | 68.52 | 84.26 | 37.62 | 91.84 | 21.43 | 96.07 | 71.77 | 82.83 | 50.53 | 87.43 |
| | | DAVIS$(m)$ + Scale | 9.31 | 98.11 | 87.25 | 67.44 | 68.20 | 83.10 | 21.15 | 95.30 | 15.59 | 97.13 | 74.82 | 80.45 | 46.05 | 86.92 |
| | | DAVIS$(\mu,\sigma)$ + Scale | 9.65 | 98.09 | 86.41 | 68.38 | 67.73 | 83.51 | 21.44 | 95.45 | 15.58 | 97.26 | 73.79 | 81.01 | 45.77 | 87.28 |

## E   Reproducibility Statement

We are committed to ensuring the reproducibility of our research. To this end, we provide detailed information regarding our code, experimental setup, hyperparameter selection, and computational environment.

**Code and Data Availability.** The complete source code for our method, DAVIS, along with the scripts used to run all experiments and generate figures, will be made publicly available on GitHub[2]. We will also provide the model weights for our trained CIFAR models. All datasets used in this work (CIFAR-10, CIFAR-100, ImageNet-1k, and all OOD benchmarks) are publicly available and were used without modification, following the standard preprocessing steps described in their original publications and common benchmarks.

**Experimental Setup.**

- **CIFAR Benchmarks:** For fair comparison, our primary models include DenseNet-101, ResNet-18, ResNet-34, and MobileNet-v2. Since the standard MobileNet-v2 architecture is designed for the high-resolution ImageNet dataset, its classification head is ill-suited for the smaller 32x32 resolution of the CIFAR benchmarks. Therefore, following common practice in DICE (Sun & Li, 2022), we adapted its head for CIFAR classification by setting the penultimate layer dimension to 512. This dimension was chosen to be comparable to other models in our evaluation, such as the ResNet variants (512) and DenseNet-101 (342). Following established protocols (Sun et al., 2021; Sun & Li, 2022; Djurisic et al., 2023), all models were trained from scratch for 100 epochs using SGD with a momentum of 0.9, a weight decay of 0.0001, and a batch size of 64. The learning rate was initialized at 0.1 and decayed by a factor of 10 at epochs 50, 75, and 90.

- **ImageNet Benchmark:** For our large-scale experiments, we used the official pre-trained models provided by PyTorch for DenseNet-121, ResNet-50, MobileNet-v2, and EfficientNet-b0. No fine-tuning was performed.

**Hyperparameter Details.** Our method, DAVIS, introduces a single key hyperparameter, $\gamma$, for the mean-variance shift formulation (DAVIS($\mu, \sigma$)). As detailed in the main paper, this was selected using a proxy OOD validation set generated with Gaussian noise. This process yielded $\gamma = 3.0$ for CIFAR models and $\gamma = 0.5$ for most ImageNet models (with the exception of $\gamma = 4.0$ for EfficientNet-b0 due to its SiLU activations).

For all baseline methods (ODIN, ReAct, DICE, ASH, SCALE), we strictly followed the hyperparameter selection protocols described in their respective papers. When re-evaluating these baselines on new architectures not present in their original work, we performed a hyperparameter search using the same validation procedure they described. Key hyperparameters for these methods are summarized below:

- **ODIN:** We adopted the optimal hyperparameter values reported in the original publication. Accordingly, we set the temperature to $T = 1000$, with a noise magnitude $\epsilon$ of 0.004 for CIFAR and 0.0015 for ImageNet.

- **ReAct:** The clipping percentile $p$ was selected from $\{85, 90, 95\}$. While we found $p = 90$ to be optimal for the standalone ReAct baseline, consistent with the original paper, the optimal value shifted to $p = 95$ when ReAct was combined with our DAVIS.

- **DICE:** We selected the sparsity ratio $p$ from $\{70, 75, 80, 85, 90, 95\}$. Our validation process consistently identified $p = 70\%$ as the optimal value.

- **ASH:** The pruning percentile $p$ was selected from $\{80, 85, 90\}$. The optimal value was found to be dependent on the dataset and architecture. We report the specific optimal value for each major setting to ensure the strongest and fairest possible comparison.

---

[2]https://github.com/epsilon-2007/DAVIS

- For **ImageNet**, the optimal value was consistently $p = 90$ for most architectures, with the exception of EfficientNet-b0, which required a less aggressive pruning of $p = 50$.
- For **CIFAR-10**, the optimal values were $p = 80$ for both ResNet models, $p = 90$ for DenseNet, and $p = 70$ for MobileNet-v2. These values held for both the standalone baseline and when combined with DAVIS.
- For **CIFAR-100**, the optimal value for the ResNet models was consistently $p = 80$. For other architectures, we observed an interaction effect: the optimal percentile for DenseNet shifted from $p = 90$ (baseline) to $p = 80$ (with DAVIS), and for MobileNet-v2, it shifted from $p = 90$ to $p = 85$.

- **SCALE:** For the SCALE baseline, the pruning percentile p was set to a fixed value of $p = 85$ across all experiments. We adopted this value directly from the original SCALE paper (Xu et al., 2024) to ensure our re-implementation was consistent with the authors' reported optimal setting, providing a fair comparison.

**Computational Environment.** All CIFAR model training and OOD detection experiments were conducted on an Apple M2 Max system with 96 GB of RAM. The experiments were implemented in Python using PyTorch (v2.1) and the Torchvision library.

# F   ANALYSIS OF ID ACCURACY

While using our modified features $h^{\texttt{DAVIS}}(\mathbf{x})$ directly for classification can degrade ID accuracy – an effect most pronounced with MobileNet-v2 on ImageNet and when combined with DICE methods on CIFAR-100 (Tables 13 and 14). As a post-hoc method, DAVIS is deployed in a standard two-branch pipeline where the original, unmodified features are always used for the final classification of any sample deemed ID, thus preserving the base model's accuracy at no cost.

Table 13: *ID classification accuracy (%) under* DAVIS *for CIFAR datasets using ResNet-18, ResNet-34, DenseNet-101, and MobileNet-v2 architectures.*

| Method | CIFAR-10 | | | | CIFAR-100 | | | |
|---|---|---|---|---|---|---|---|---|
| | ResNet-18 | ResNet-34 | DenseNet-101 | MobileNet-v2 | ResNet-18 | ResNet-34 | DenseNet-101 | MobileNet-v2 |
| Standard | 93.89 | 93.96 | 93.61 | 92.52 | 75.20 | 75.70 | 74.47 | 72.69 |
| DAVIS($m$) | 93.50 | 93.60 | 92.88 | 91.88 | 75.07 | 75.38 | 67.78 | 70.41 |
| DAVIS($m$) + DICE | 93.23 | 93.39 | 92.53 | 89.21 | 71.50 | 73.15 | 62.23 | 69.54 |
| DAVIS($\mu, \sigma$) | 93.80 | 93.88 | 93.63 | 92.17 | 75.48 | 75.67 | 72.87 | 71.15 |
| DAVIS($\mu, \sigma$) + DICE | 93.37 | 93.68 | 93.36 | 88.65 | 72.50 | 73.81 | 68.47 | 70.50 |

Table 14: *ID classification accuracy (%) under* DAVIS *for ImageNet datasets using DenseNet-121, ResNet-50, MobileNet-v2, and EfficientNet-b0 architectures.*

| Method | DenseNet-121 | ResNet-50 | MobileNet-v2[3] | EfficientNet-b0 |
|---|---|---|---|---|
| Standard | 74.44 | 76.15 | 71.87 | 77.67 |
| DAVIS($m$) | 71.95 | 73.47 | 69.19 | 74.87 |
| DAVIS($m$) + ASH | 70.46 | 72.72 | 35.03 | 74.93 |
| DAVIS($\mu, \sigma$) | 73.54 | 75.63 | 71.09 | 76.97 |
| DAVIS($\mu, \sigma$) + ASH | 73.03 | 75.13 | 52.90 | 76.78 |

---

[3]The combined method for computing accuracy in MobileNet-V2 is ReAct+DICE.

Computing OOD scores with `DAVIS` modifies the original $h(\mathbf{x})$ (as detailed in Section 3) and introduces negligible overhead – only a single matrix multiplication for the OOD head, which is trivial compared to the GFLOPs of the main CNN backbone. For instance, in ResNet-50 on ImageNet, the backbone requires 5.42 GFLOPs, while the OOD head adds only 0.004 GFLOPs, i.e., less than 0.1% of the total cost.

# G  SYNERGY WITH EXISTING METHODS

Our method `DAVIS` is designed not as a replacement for existing techniques, but as a complementary module that enhances them. It acts as a feature pre-processing step, enriching the standard penultimate layer representation with more discriminative statistics (e.g., maximum and variance) that subsequent methods like `MSP` (Hendrycks & Gimpel, 2017), `ODIN` (Liang et al., 2018), `Energy` (Liu et al., 2020), `ReAct` (Sun et al., 2021), `DICE` (Sun & Li, 2022), `ASH` (Djurisic et al., 2023), and `SCALE` (Xu et al., 2024) can then leverage more effectively. So, instead of replacing existing techniques, `DAVIS` introduces an additional degree of freedom to modify the penultimate layer in a way that works in tandem with them.

**ImageNet.** This synergistic effect extends to the large-scale ImageNet benchmark, where the `DAVIS`$(\mu, \sigma)$ variant consistently enhances the performance of all foundational methods. Our results, detailed in Table 16, show significant gains even for strong baselines like ASH and SCALE. This demonstrates the scalability of our approach and confirms that enriching feature representations with statistical variance is a robust strategy for improving OOD detection on complex, large-scale datasets.

**CIFAR.** To validate this synergy, we conduct a systematic evaluation, comparing the performance of each foundational baseline with and without the application of our `DAVIS` module. The results, presented for CIFAR-10 (Tables 17 and 18) and CIFAR-100 (Tables 19 and 20), demonstrate consistent and significant performance gains across all architectures. Notably, integrating `DAVIS` boosts the performance of strong baselines, confirming that our enriched feature representation provides a more robust foundation for OOD detection.

For brevity, we omit results for the ODIN baseline in our main CIFAR tables (Tables 17, 18, 19, and 20). This decision is twofold: first, similar to MSP, the performance of ODIN when combined with `DAVIS` is largely neutral. Second, the energy score baseline is consistently superior, and ODIN's reliance on a computationally expensive input perturbation step makes it less practical for many applications.

**Note on Compatibility with Scoring Functions.** While `DAVIS` is compatible with any scoring function, our evaluation primarily uses the energy score as it demonstrates the most consistent and significant synergistic gains. We attribute this to the fundamental difference between scoring functions: the energy score leverages the entire logit vector to distinguish between ID and OOD samples, whereas MSP and ODIN rely solely on the single maximum logit value (i.e., the softmax confidence) as shown below:

$$S_{\text{MSP}}(\mathbf{x}) = \max_i P(y = i | \mathbf{x}) \qquad (17) \qquad S_{\text{Energy}}(\mathbf{x}) = \log \left( \sum_{i=1}^{C} e^{f_i(\mathbf{x})} \right) \qquad (18)$$

Figure 6: Comparison of scoring functions. The MSP score depends only on the single maximum softmax probability, while the Energy score incorporates all logit values ($f_i(\mathbf{x})$).

Our method, `DAVIS`, transforms the feature space to create a richer, more separable distribution of logits that is highly beneficial for the holistic energy score. A side effect of this transformation, however, is that the magnitude of the single dominant logit for ID samples can be suppressed. This suppression is the primary reason for the largely unchanged performance observed when combining `DAVIS` with MSP and ODIN as

shown in Table 15. Furthermore, ODIN requires a costly input perturbation step (based on FGSM attacks) to amplify softmax scores, rendering it impractical for many real-world applications.

This finding is consistent with the ID accuracy drop analyzed in Appendix F, which is also governed by the dominant logit. Therefore, the full potential of our enriched feature representation is best realized by holistic scoring functions like the energy score, rather than those dependent on a single "winner-takes-all" logit.

Table 15: *Detailed results of post-hoc MSP and ODIN combined with* `DAVIS` *using DensetNet-121, ResNet-50, MobileNet-v2, and EfficientNet-b0 trained on* **ImageNet-1K**. $\uparrow$ *indicates higher is better;* $\downarrow$ *indicates lower is better. The symbols denote the statistic used:* $\mu$ *(mean),* $\sigma$ *(std. deviation),* $m$ *(maximum).*

| Model | Combined Method | SUN | | Place365 | | Textures | | iNaturalist | | Average | |
|---|---|---|---|---|---|---|---|---|---|---|---|
| | | FPR95 $\downarrow$ | AUROC $\uparrow$ | FPR95 $\downarrow$ | AUROC $\uparrow$ | FPR95 $\downarrow$ | AUROC $\uparrow$ | FPR95 $\downarrow$ | AUROC $\uparrow$ | FPR95 $\downarrow$ | AUROC $\uparrow$ |
| DenseNet-121 | MSP | 67.49 | 81.41 | 69.53 | 80.95 | 67.23 | 79.18 | 49.58 | 89.05 | 63.46 | 82.65 |
| | + DAVIS($m$) | 71.86 | 79.62 | 73.50 | 79.11 | 65.90 | 79.97 | 56.40 | 86.22 | 66.92 | 81.23 |
| | + DAVIS($\mu,\sigma$) | 81.36 | 75.95 | 82.16 | 75.15 | 74.63 | 78.59 | 75.56 | 80.08 | 78.43 | 77.45 |
| | ODIN | 54.13 | 86.33 | 60.39 | 84.14 | 50.82 | 85.81 | 32.47 | 93.66 | 49.45 | 87.48 |
| | + DAVIS($m$) | 54.57 | 86.95 | 60.24 | 84.90 | 51.61 | 86.32 | 36.11 | 93.36 | 50.63 | 87.88 |
| | + DAVIS($\mu,\sigma$) | 54.90 | 86.33 | 60.77 | 84.19 | 51.52 | 85.77 | 33.15 | 93.62 | 50.09 | 87.48 |
| ResNet-50 | MSP | 69.11 | 81.64 | 72.06 | 80.54 | 66.26 | 80.43 | 52.83 | 88.39 | 65.06 | 82.75 |
| | + DAVIS($m$) | 81.56 | 75.85 | 83.35 | 74.42 | 71.26 | 80.95 | 75.63 | 79.93 | 77.95 | 77.79 |
| | + DAVIS($\mu,\sigma$) | 72.96 | 79.77 | 75.22 | 78.59 | 64.49 | 81.44 | 58.92 | 85.81 | 67.90 | 81.40 |
| | ODIN | 60.15 | 84.59 | 67.89 | 81.78 | 50.23 | 85.62 | 47.66 | 89.66 | 56.48 | 85.41 |
| | + DAVIS($m$) | 58.10 | 86.94 | 65.19 | 84.48 | 49.88 | 87.66 | 45.37 | 91.92 | 54.63 | 87.75 |
| | + DAVIS($\mu,\sigma$) | 57.27 | 86.75 | 64.96 | 84.12 | 47.75 | 87.74 | 42.39 | 92.19 | 53.09 | 87.70 |
| MobileNet-v2 | MSP | 74.28 | 78.81 | 76.65 | 78.10 | 71.05 | 78.93 | 59.70 | 86.72 | 70.42 | 80.64 |
| | + DAVIS($m$) | 84.10 | 73.13 | 85.49 | 72.38 | 75.94 | 77.11 | 78.74 | 78.08 | 81.07 | 75.17 |
| | + DAVIS($\mu,\sigma$) | 76.77 | 76.90 | 79.03 | 76.18 | 69.82 | 79.23 | 63.89 | 84.14 | 72.38 | 79.11 |
| | ODIN | 54.07 | 85.88 | 57.36 | 84.71 | 49.96 | 85.03 | 55.39 | 87.62 | 54.20 | 85.81 |
| | + DAVIS($m$) | 60.90 | 86.06 | 66.65 | 83.44 | 55.51 | 86.89 | 49.83 | 91.44 | 58.22 | 86.96 |
| | + DAVIS($\mu,\sigma$) | 60.10 | 85.94 | 66.80 | 83.13 | 51.60 | 87.32 | 47.61 | 91.62 | 56.53 | 87.00 |
| EfficentNet-b0 | MSP | 72.56 | 80.06 | 74.07 | 79.17 | 66.83 | 81.19 | 57.42 | 87.28 | 67.72 | 81.92 |
| | + DAVIS($m$) | 82.82 | 76.17 | 84.85 | 73.86 | 71.01 | 84.59 | 79.14 | 78.40 | 79.46 | 78.26 |
| | + DAVIS($\mu,\sigma$) | 72.94 | 79.69 | 75.91 | 78.34 | 58.92 | 84.91 | 61.34 | 84.91 | 67.28 | 81.96 |
| | ODIN | 72.57 | 75.74 | 77.87 | 73.20 | 64.96 | 79.15 | 59.01 | 83.99 | 68.60 | 78.02 |
| | + DAVIS($m$) | 70.05 | 78.12 | 74.74 | 75.91 | 63.16 | 80.75 | 56.37 | 85.89 | 66.08 | 80.17 |
| | + DAVIS($\mu,\sigma$) | 71.93 | 75.93 | 76.92 | 73.49 | 64.93 | 79.11 | 58.14 | 84.37 | 67.98 | 78.23 |

Table 16: *Detailed results of post-hoc methods combined with* DAVIS *using DensetNet-121, ResNet-50, MobileNet-v2, and EfficientNet-b0 trained on* **ImageNet-1K**. ↑ *indicates higher is better;* ↓ *indicates lower is better. The symbols denote the statistic used:* μ *(mean),* σ *(std. deviation),* m *(maximum).*

| Model | Combined Method | SUN | | Place365 | | Textures | | iNaturalist | | Average | |
|---|---|---|---|---|---|---|---|---|---|---|---|
| | | FPR95↓ | AUROC↑ | FPR95↓ | AUROC↑ | FPR95↓ | AUROC↑ | FPR95↓ | AUROC↑ | FPR95↓ | AUROC↑ |
| DenseNet-121 | Energy | 52.51 | 87.27 | 58.24 | 85.05 | 52.22 | 85.42 | 39.75 | 92.66 | 50.68 | 87.6 |
| | + DAVIS(m) | 48.55 | 88.03 | 59.36 | 83.53 | 27.54 | 93.62 | 33.74 | 92.20 | 42.30 | 89.34 |
| | + DAVIS(μ,σ) | 48.40 | 88.64 | 58.38 | 84.94 | 25.37 | 94.14 | 30.31 | 93.79 | 40.62 | 90.38 |
| | ReAct | 43.65 | 90.65 | 51.05 | 87.54 | 43.48 | 90.81 | 25.74 | 95.05 | 40.98 | 91.01 |
| | + DAVIS(m) | 48.35 | 89.33 | 57.07 | 85.46 | 26.91 | 94.41 | 31.62 | 93.54 | 40.99 | 90.69 |
| | + DAVIS(μ,σ) | 44.99 | 90.53 | 52.91 | 87.41 | 28.37 | 94.23 | 24.92 | 95.45 | 37.80 | 91.91 |
| | DICE | 38.75 | 89.91 | 49.29 | 86.24 | 40.85 | 88.09 | 25.78 | 94.37 | 38.67 | 89.65 |
| | + DAVIS(m) | 45.37 | 86.54 | 59.09 | 79.97 | 20.78 | 94.98 | 31.02 | 91.46 | 39.07 | 88.24 |
| | + DAVIS(μ,σ) | 37.85 | 90.15 | 50.22 | 85.78 | 26.81 | 93.55 | 23.08 | 94.84 | 34.49 | 91.08 |
| | ASH | 37.20 | 91.51 | 46.54 | 88.79 | 21.76 | 95.04 | 15.50 | 97.03 | 30.25 | 93.09 |
| | + DAVIS(m) | 38.10 | 91.39 | 50.52 | 87.38 | 13.23 | 97.37 | 21.35 | 95.72 | 30.80 | 92.97 |
| | + DAVIS(μ,σ) | 35.89 | 91.98 | 46.66 | 88.97 | 15.92 | 96.63 | 15.30 | 97.04 | 28.44 | 93.66 |
| | Scale | 33.85 | 92.16 | 42.92 | 89.62 | 22.27 | 94.63 | 13.21 | 97.40 | 28.06 | 93.45 |
| | +DAVIS(m) | 33.68 | 92.03 | 45.81 | 88.31 | 13.30 | 97.24 | 18.13 | 96.23 | 27.73 | 93.45 |
| | +DAVIS(μ,σ) | 32.45 | 92.65 | 42.41 | 89.84 | 16.29 | 96.40 | 12.71 | 97.45 | 25.97 | 94.09 |
| ResNet-50 | Energy | 58.82 | 86.58 | 65.99 | 83.96 | 52.43 | 86.72 | 53.74 | 90.62 | 57.74 | 86.97 |
| | + DAVIS(m) | 54.71 | 86.47 | 64.06 | 82.55 | 21.13 | 95.32 | 34.50 | 92.95 | 43.60 | 89.33 |
| | + DAVIS(μ,σ) | 53.12 | 85.49 | 63.58 | 80.54 | 20.94 | 95.09 | 36.11 | 91.29 | 43.44 | 88.10 |
| | ReAct | 24.20 | 94.20 | 33.85 | 91.58 | 47.30 | 89.80 | 20.38 | 96.22 | 31.43 | 92.95 |
| | + DAVIS(m) | 34.14 | 92.45 | 44.47 | 89.64 | 34.11 | 92.88 | 21.69 | 95.91 | 33.60 | 92.72 |
| | + DAVIS(μ,σ) | 43.97 | 89.54 | 53.47 | 85.96 | 26.81 | 93.91 | 30.74 | 93.64 | 38.75 | 90.76 |
| | DICE | 35.15 | 90.83 | 46.49 | 87.48 | 31.72 | 90.30 | 25.63 | 94.49 | 34.75 | 90.78 |
| | + DAVIS(m) | 49.67 | 84.39 | 63.99 | 77.73 | 15.87 | 96.21 | 31.88 | 91.98 | 40.35 | 87.58 |
| | + DAVIS(μ,σ) | 37.38 | 90.32 | 50.34 | 86.42 | 19.75 | 94.92 | 21.76 | 95.33 | 32.31 | 91.74 |
| | ASH | 27.98 | 94.02 | 39.78 | 90.98 | 11.93 | 97.60 | 11.49 | 97.87 | 22.80 | 95.12 |
| | + DAVIS(m) | 29.89 | 93.61 | 43.08 | 89.89 | 9.22 | 98.26 | 15.25 | 97.21 | 24.36 | 94.74 |
| | + DAVIS(μ,σ) | 27.67 | 94.02 | 40.01 | 90.73 | 9.91 | 98.07 | 11.69 | 97.81 | 22.32 | 95.16 |
| | Scale | 25.78 | 94.54 | 36.86 | 91.96 | 14.56 | 96.75 | 10.37 | 98.02 | 21.89 | 95.32 |
| | +DAVIS(m) | 25.38 | 94.55 | 37.51 | 91.09 | 10.48 | 97.95 | 11.94 | 97.75 | 21.33 | 95.34 |
| | +DAVIS(μ,σ) | 23.42 | 94.93 | 34.61 | 91.98 | 11.37 | 97.74 | 9.46 | 98.15 | 19.71 | 95.70 |
| MobileNet-v2 | Energy | 59.60 | 86.16 | 66.36 | 83.15 | 54.82 | 86.57 | 55.33 | 90.37 | 59.03 | 86.56 |
| | + DAVIS(m) | 53.64 | 87.26 | 63.61 | 82.93 | 25.87 | 94.03 | 40.25 | 92.22 | 45.84 | 89.11 |
| | + DAVIS(μ,σ) | 55.47 | 87.15 | 64.82 | 83.18 | 25.78 | 94.33 | 38.94 | 92.98 | 46.25 | 89.41 |
| | ReAct | 52.68 | 87.21 | 59.81 | 84.04 | 40.32 | 90.96 | 42.94 | 92.75 | 48.94 | 88.74 |
| | + DAVIS(m) | 54.27 | 87.14 | 63.94 | 82.97 | 26.70 | 93.80 | 40.64 | 92.23 | 46.39 | 89.03 |
| | + DAVIS(μ,σ) | 55.09 | 87.45 | 62.88 | 84.17 | 36.26 | 91.82 | 42.68 | 92.80 | 49.23 | 89.06 |
| | DICE | 38.81 | 90.46 | 52.95 | 85.82 | 33.00 | 91.27 | 42.95 | 90.87 | 41.93 | 89.60 |
| | + DAVIS(m) | 45.42 | 87.19 | 60.92 | 79.71 | 12.68 | 97.43 | 41.95 | 90.21 | 40.24 | 88.63 |
| | + DAVIS(μ,σ) | 38.43 | 90.17 | 53.35 | 84.95 | 19.13 | 95.34 | 38.01 | 91.95 | 37.23 | 90.60 |
| | ASH | 43.86 | 89.98 | 58.92 | 84.72 | 13.21 | 97.10 | 39.13 | 91.96 | 38.78 | 90.94 |
| | + DAVIS(m) | 46.24 | 89.41 | 61.42 | 83.51 | 10.83 | 97.88 | 41.40 | 91.95 | 39.97 | 90.69 |
| | + DAVIS(μ,σ) | 44.59 | 89.74 | 60.08 | 84.22 | 11.35 | 97.62 | 39.31 | 91.87 | 38.83 | 90.86 |
| | Scale | 38.74 | 91.64 | 53.49 | 87.34 | 14.79 | 96.65 | 30.09 | 94.46 | 34.28 | 92.52 |
| | +DAVIS(m) | 37.47 | 91.91 | 52.72 | 86.90 | 8.85 | 98.20 | 26.76 | 95.08 | 31.45 | 93.02 |
| | +DAVIS(μ,σ) | 37.89 | 91.82 | 52.92 | 87.24 | 11.37 | 97.60 | 28.54 | 94.76 | 32.68 | 92.86 |
| EfficentNet-b0 | Energy | 85.01 | 72.86 | 86.06 | 70.99 | 75.99 | 75.86 | 78.91 | 79.78 | 81.49 | 74.87 |
| | + DAVIS(m) | 69.42 | 80.74 | 79.09 | 73.67 | 15.82 | 96.38 | 63.68 | 80.63 | 57.00 | 82.85 |
| | + DAVIS(μ,σ) | 61.85 | 84.67 | 71.31 | 79.01 | 12.18 | 97.52 | 48.25 | 88.35 | 48.40 | 87.39 |
| | ReAct | 72.96 | 80.97 | 77.59 | 77.61 | 34.57 | 92.44 | 55.19 | 89.82 | 60.08 | 85.21 |
| | + DAVIS(m) | 68.81 | 83.32 | 76.36 | 78.36 | 17.52 | 96.73 | 59.12 | 87.07 | 55.45 | 86.37 |
| | + DAVIS(μ,σ) | 62.89 | 85.32 | 69.97 | 81.16 | 15.89 | 97.19 | 45.80 | 90.96 | 48.64 | 88.66 |
| | DICE | 98.15 | 44.65 | 99.17 | 39.44 | 93.83 | 59.29 | 99.66 | 39.80 | 97.70 | 45.79 |
| | + DAVIS(m) | 86.98 | 64.76 | 92.59 | 54.67 | 37.32 | 90.08 | 86.54 | 63.50 | 75.86 | 68.25 |
| | + DAVIS(μ,σ) | 93.24 | 59.58 | 96.61 | 51.01 | 56.97 | 83.18 | 94.03 | 59.37 | 85.21 | 63.29 |
| | ASH | 98.62 | 52.65 | 99.30 | 48.33 | 97.64 | 67.20 | 99.78 | 53.49 | 98.84 | 55.42 |
| | + DAVIS(m) | 67.84 | 80.49 | 79.26 | 72.78 | 14.98 | 96.51 | 63.21 | 81.58 | 56.32 | 82.84 |
| | + DAVIS(μ,σ) | 61.11 | 84.81 | 72.70 | 78.53 | 10.02 | 97.94 | 47.76 | 88.97 | 47.90 | 87.56 |
| | Scale | 98.48 | 53.34 | 99.27 | 47.85 | 96.81 | 71.19 | 99.67 | 55.44 | 98.56 | 56.95 |
| | +DAVIS(m) | 72.89 | 76.83 | 82.74 | 68.50 | 21.35 | 95.31 | 69.02 | 80.74 | 61.50 | 80.35 |
| | +DAVIS(μ,σ) | 86.53 | 68.48 | 92.42 | 60.52 | 38.19 | 91.34 | 83.31 | 74.47 | 75.11 | 73.70 |

Table 17: Detailed results of post-hoc methods combined with DAVIS using **ResNet-18** and **ResNet-34** trained on **CIFAR-10**. ↑ indicates higher is better; ↓ indicates lower is better. The symbols denote the statistic used: μ (mean), σ (std. deviation), m (maximum)

| Model | Combined Method | SVHN FPR95 ↓ | SVHN AUROC ↑ | Place365 FPR95 ↓ | Place365 AUROC ↑ | iSUN FPR95 ↓ | iSUN AUROC ↑ | Textures FPR95 ↓ | Textures AUROC ↑ | LSUN-c FPR95 ↓ | LSUN-c AUROC ↑ | LSUN-r FPR95 ↓ | LSUN-r AUROC ↑ | Average FPR95 ↓ | Average AUROC ↑ |
|---|---|---|---|---|---|---|---|---|---|---|---|---|---|---|---|
| ResNet-18 | MSP | 60.39 | 92.40 | 63.49 | 88.38 | 56.59 | 91.18 | 62.71 | 90.10 | 51.87 | 93.64 | 55.53 | 91.69 | 58.43 | 91.23 |
| | +DAVIS(m) | 51.03 | 89.90 | 64.30 | 81.73 | 56.30 | 87.06 | 55.87 | 87.04 | 43.94 | 92.10 | 54.37 | 88.06 | 54.30 | 87.65 |
| | +DAVIS(μ,σ) | 51.90 | 89.00 | 64.90 | 78.91 | 57.94 | 83.35 | 57.61 | 83.90 | 46.20 | 90.69 | 56.29 | 84.64 | 55.81 | 85.08 |
| | Energy | 44.32 | 94.04 | 41.31 | 91.73 | 35.46 | 94.64 | 50.39 | 91.12 | 9.77 | 98.19 | 32.41 | 95.16 | 35.61 | 94.14 |
| | +DAVIS(m) | 19.81 | 96.29 | 32.32 | 93.73 | 15.79 | 97.37 | 21.90 | 96.44 | 5.83 | 98.73 | 13.63 | 97.61 | 18.21 | 96.69 |
| | +DAVIS(μ,σ) | 19.83 | 96.47 | 36.11 | 93.16 | 20.48 | 96.90 | 26.77 | 95.89 | 5.62 | 98.79 | 17.75 | 97.21 | 21.09 | 96.00 |
| | ReAct | 42.31 | 94.12 | 40.74 | 92.25 | 24.06 | 96.26 | 40.44 | 93.69 | 12.27 | 97.90 | 21.02 | 96.67 | 30.14 | 95.15 |
| | +DAVIS(m) | 21.54 | 95.93 | 32.84 | 93.65 | 16.26 | 97.27 | 23.07 | 96.21 | 6.67 | 98.61 | 14.07 | 97.54 | 19.07 | 96.53 |
| | +DAVIS(μ,σ) | 21.70 | 96.10 | 36.12 | 93.18 | 20.04 | 96.90 | 26.93 | 95.74 | 6.43 | 98.67 | 17.24 | 97.25 | 21.40 | 96.31 |
| | DICE | 17.60 | 97.09 | 46.16 | 90.66 | 38.68 | 94.32 | 44.50 | 91.81 | 1.90 | 99.57 | 36.66 | 94.67 | 30.92 | 94.69 |
| | +DAVIS(m) | 7.95 | 98.50 | 30.55 | 93.97 | 6.70 | 98.59 | 9.66 | 98.28 | 1.24 | 99.73 | 6.63 | 98.57 | 10.46 | 97.94 |
| | +DAVIS(μ,σ) | 7.85 | 98.57 | 35.27 | 93.18 | 11.96 | 97.96 | 14.04 | 97.71 | 1.14 | 99.76 | 10.71 | 98.03 | 13.49 | 97.54 |
| | ReAct+DICE | 12.25 | 97.85 | 45.77 | 91.27 | 18.12 | 96.92 | 26.90 | 95.60 | 1.35 | 99.65 | 16.93 | 97.09 | 20.22 | 96.40 |
| | +DAVIS(m) | 8.23 | 98.47 | 30.40 | 94.06 | 6.49 | 98.63 | 9.52 | 98.29 | 1.33 | 99.73 | 6.34 | 98.62 | 10.38 | 97.97 |
| | +DAVIS(μ,σ) | 8.31 | 98.49 | 34.88 | 93.34 | 10.29 | 98.14 | 13.01 | 97.82 | 1.30 | 99.75 | 9.49 | 98.22 | 12.88 | 97.62 |
| | ASH | 7.87 | 98.43 | 49.69 | 89.57 | 23.27 | 96.33 | 26.12 | 95.88 | 2.10 | 99.46 | 21.91 | 96.47 | 21.83 | 96.02 |
| | +DAVIS(m) | 9.25 | 98.37 | 41.26 | 91.73 | 10.87 | 98.01 | 11.60 | 97.95 | 1.94 | 99.54 | 10.46 | 98.02 | 14.23 | 97.27 |
| | +DAVIS(μ,σ) | 7.31 | 98.68 | 43.72 | 91.00 | 12.58 | 97.81 | 12.16 | 97.86 | 1.70 | 99.60 | 12.23 | 97.85 | 14.95 | 97.13 |
| | Scale | 9.73 | 98.13 | 45.99 | 90.87 | 22.92 | 96.39 | 27.00 | 95.61 | 3.75 | 99.17 | 21.02 | 96.60 | 21.74 | 96.13 |
| | +DAVIS(m) | 10.21 | 98.17 | 37.09 | 92.92 | 9.74 | 98.17 | 12.77 | 97.83 | 2.86 | 99.39 | 9.71 | 98.19 | 13.73 | 97.45 |
| | +DAVIS(μ,σ) | 8.88 | 98.43 | 39.84 | 92.38 | 12.36 | 97.93 | 14.11 | 97.65 | 2.71 | 99.43 | 11.62 | 97.99 | 14.92 | 97.30 |
| ResNet-34 | MSP | 62.20 | 91.10 | 62.76 | 88.95 | 52.52 | 92.81 | 57.93 | 90.75 | 42.06 | 95.17 | 51.72 | 92.98 | 54.86 | 91.96 |
| | +DAVIS(m) | 55.11 | 83.23 | 60.27 | 82.68 | 52.55 | 88.19 | 52 | 87.21 | 34.05 | 93.38 | 51.08 | 88.75 | 50.84 | 87.24 |
| | +DAVIS(μ,σ) | 56.69 | 81.65 | 62.59 | 79.22 | 54.59 | 85.37 | 54.91 | 84.35 | 36.42 | 92.29 | 53.99 | 85.84 | 53.2 | 84.79 |
| | Energy | 35.44 | 93.76 | 38.15 | 92.27 | 19.90 | 96.87 | 42.52 | 92.54 | 3.38 | 99.10 | 16.86 | 97.20 | 26.04 | 95.29 |
| | +DAVIS(m) | 29.29 | 94.59 | 29.59 | 94.21 | 10.17 | 98.03 | 20.23 | 96.73 | 4.00 | 99.10 | 9.45 | 98.16 | 17.12 | 96.81 |
| | +DAVIS(μ,σ) | 28.96 | 94.90 | 32.66 | 93.69 | 11.31 | 97.88 | 23.07 | 96.39 | 3.47 | 99.22 | 10.35 | 98.04 | 18.30 | 96.69 |
| | ReAct | 33.03 | 93.66 | 36.11 | 93.26 | 21.64 | 96.61 | 41.19 | 93.22 | 6.83 | 98.57 | 19.36 | 96.88 | 26.36 | 95.37 |
| | +DAVIS(m) | 35.09 | 92.92 | 31.91 | 93.98 | 15.14 | 97.49 | 26.24 | 95.8 | 6.22 | 98.8 | 13.96 | 97.65 | 21.43 | 96.11 |
| | +DAVIS(μ,σ) | 33.49 | 93.54 | 33.43 | 93.77 | 16.27 | 97.32 | 28.46 | 95.49 | 5.45 | 98.91 | 14.62 | 97.5 | 21.95 | 96.09 |
| | DICE | 26.78 | 95.46 | 39.75 | 91.83 | 16.00 | 97.41 | 40.05 | 92.91 | 0.71 | 99.82 | 14.73 | 97.58 | 23.00 | 95.84 |
| | +DAVIS(m) | 14.79 | 97.29 | 27.51 | 94.59 | 5.09 | 98.98 | 10.07 | 98.26 | 1.22 | 99.75 | 5.36 | 98.95 | 10.67 | 97.97 |
| | +DAVIS(μ,σ) | 15.89 | 97.25 | 30.60 | 93.97 | 6.31 | 98.78 | 12.55 | 97.88 | 0.83 | 99.82 | 6.37 | 98.78 | 12.09 | 97.75 |
| | ReAct+DICE | 19.63 | 96.31 | 40.69 | 92.65 | 17.98 | 97.05 | 33.32 | 94.18 | 0.93 | 99.77 | 17.98 | 97.07 | 21.75 | 96.17 |
| | +DAVIS(m) | 16.64 | 96.64 | 27.92 | 94.48 | 6.14 | 98.78 | 12.23 | 97.82 | 1.57 | 99.70 | 6.71 | 98.71 | 11.87 | 97.69 |
| | +DAVIS(μ,σ) | 16.74 | 96.72 | 29.46 | 94.24 | 7.31 | 98.53 | 14.93 | 97.41 | 1.10 | 99.77 | 7.84 | 98.49 | 12.90 | 97.53 |
| | ASH | 14.58 | 97.56 | 41.91 | 90.96 | 15.71 | 97.33 | 26.88 | 95.38 | 1.74 | 99.54 | 16.59 | 97.23 | 19.57 | 96.34 |
| | +DAVIS(m) | 18.48 | 96.80 | 31.35 | 93.83 | 8.93 | 98.23 | 14.80 | 97.48 | 3.05 | 99.35 | 9.23 | 98.22 | 14.31 | 97.32 |
| | +DAVIS(μ,σ) | 13.23 | 97.78 | 35.56 | 92.48 | 10.01 | 98.09 | 14.82 | 97.38 | 2.30 | 99.51 | 10.55 | 98.00 | 14.41 | 97.21 |
| | Scale | 17.47 | 97.01 | 38.01 | 91.84 | 15.06 | 97.31 | 28.40 | 94.86 | 2.37 | 99.44 | 15.67 | 97.25 | 19.50 | 96.28 |
| | +DAVIS(m) | 20.13 | 96.60 | 33.58 | 93.37 | 10.20 | 98.07 | 17.61 | 97.01 | 3.95 | 99.23 | 10.92 | 97.99 | 16.06 | 97.05 |
| | +DAVIS(μ,σ) | 17.83 | 97.00 | 34.47 | 92.99 | 10.35 | 98.01 | 18.49 | 96.87 | 3.10 | 99.35 | 11.08 | 97.92 | 15.89 | 97.02 |

Table 18: *Detailed results of post-hoc methods combined with DAVIS using DenseNet-101 and MobileNet-v2 trained on CIFAR-10.* ↑ indicates higher is better; ↓ indicates lower is better. The symbols denote the statistic used: μ (mean), σ (std. deviation), m (maximum)

| Model | Combined Method | SVHN FPR95↓ | SVHN AUROC↑ | Place365 FPR95↓ | Place365 AUROC↑ | iSUN FPR95↓ | iSUN AUROC↑ | Textures FPR95↓ | Textures AUROC↑ | LSUN-c FPR95↓ | LSUN-c AUROC↑ | LSUN-r FPR95↓ | LSUN-r AUROC↑ | Average FPR95↓ | Average AUROC↑ |
|---|---|---|---|---|---|---|---|---|---|---|---|---|---|---|---|
| DenseNet-101 | MSP | 64.76 | 88.33 | 60.19 | 88.56 | 33.34 | 95.41 | 56.60 | 90.17 | 23.41 | 96.75 | 33.88 | 95.39 | 45.36 | 92.43 |
| | + DAVIS(m) | 65.33 | 85.68 | 67.63 | 83.28 | 51.76 | 91.67 | 61.38 | 87.95 | 50.8 | 91.17 | 53.27 | 91.43 | 58.36 | 88.53 |
| | + DAVIS(μ,σ) | 63.11 | 85.23 | 65.59 | 82.43 | 45.08 | 92.18 | 58.1 | 87.51 | 42.91 | 92.61 | 46.61 | 91.93 | 53.57 | 88.65 |
| | Energy | 37.91 | 93.59 | 36.38 | 92.39 | 7.83 | 98.23 | 43.85 | 90.49 | 1.95 | 99.47 | 7.34 | 98.34 | 22.54 | 95.42 |
| | + DAVIS(m) | 30.50 | 94.50 | 33.04 | 93.28 | 7.81 | 98.29 | 25.16 | 95.85 | 9.95 | 98.19 | 7.32 | 98.37 | 18.96 | 96.41 |
| | + DAVIS(μ,σ) | 24.75 | 95.93 | 32.08 | 93.43 | 5.77 | 98.64 | 25.23 | 95.84 | 5.13 | 98.94 | 5.63 | 98.70 | 16.43 | 96.91 |
| | ReAct | 23.18 | 96.28 | 33.97 | 92.98 | 5.95 | 98.45 | 32.25 | 93.98 | 2.47 | 99.33 | 5.44 | 98.55 | 17.21 | 96.59 |
| | + DAVIS(m) | 32.67 | 93.43 | 35.41 | 92.58 | 9.48 | 98.00 | 27.84 | 95.18 | 12.60 | 97.66 | 9.34 | 98.07 | 21.22 | 95.82 |
| | + DAVIS(μ,σ) | 23.78 | 95.59 | 34.68 | 92.90 | 7.37 | 98.37 | 26.12 | 95.46 | 7.26 | 98.60 | 7.30 | 98.43 | 17.75 | 96.56 |
| | DICE | 16.68 | 96.96 | 37.46 | 92.06 | 2.25 | 99.41 | 28.05 | 92.70 | 0.16 | 99.94 | 2.44 | 99.35 | 14.51 | 96.74 |
| | + DAVIS(m) | 8.30 | 98.37 | 29.47 | 93.92 | 1.85 | 99.56 | 7.16 | 98.69 | 1.26 | 99.72 | 1.92 | 99.55 | 8.33 | 98.30 |
| | + DAVIS(μ,σ) | 6.84 | 98.77 | 29.76 | 93.86 | 1.58 | 99.59 | 9.57 | 98.25 | 0.55 | 99.86 | 1.60 | 99.57 | 8.32 | 98.32 |
| | ReAct+DICE | 4.63 | 99.02 | 36.10 | 92.93 | 1.80 | 99.51 | 17.32 | 96.76 | 0.12 | 99.95 | 1.93 | 99.47 | 10.32 | 97.94 |
| | + DAVIS(m) | 7.86 | 98.39 | 29.27 | 93.87 | 1.73 | 99.58 | 5.90 | 98.85 | 1.19 | 99.73 | 1.84 | 99.56 | 7.96 | 98.33 |
| | + DAVIS(μ,σ) | 5.90 | 98.87 | 29.45 | 93.92 | 1.52 | 99.59 | 7.41 | 98.62 | 0.55 | 99.86 | 1.85 | 99.57 | 7.78 | 98.41 |
| | ASH | 16.20 | 97.21 | 37.79 | 92.02 | 3.91 | 98.94 | 26.40 | 94.61 | 0.84 | 99.69 | 4.15 | 98.92 | 14.88 | 96.90 |
| | + DAVIS(m) | 12.48 | 97.63 | 34.90 | 93.03 | 4.63 | 98.90 | 11.76 | 97.94 | 4.59 | 99.08 | 5.07 | 98.84 | 12.24 | 97.57 |
| | + DAVIS(μ,σ) | 9.78 | 98.20 | 34.54 | 93.10 | 3.59 | 99.10 | 12.45 | 97.87 | 2.25 | 99.44 | 4.06 | 99.08 | 11.11 | 97.80 |
| | Scale | 23.06 | 96.13 | 36.53 | 92.24 | 4.54 | 98.76 | 30.53 | 93.62 | 1.23 | 99.61 | 4.59 | 98.75 | 16.75 | 96.52 |
| | + DAVIS(m) | 20.38 | 96.25 | 33.11 | 93.27 | 5.86 | 98.63 | 15.80 | 97.27 | 6.57 | 98.72 | 6.07 | 98.61 | 14.63 | 97.13 |
| | + DAVIS(μ,σ) | 14.93 | 97.30 | 31.94 | 93.46 | 4.04 | 98.95 | 15.74 | 97.31 | 3.28 | 99.26 | 4.28 | 98.95 | 12.37 | 97.54 |
| MobileNet-v2 | MSP | 72.84 | 88.69 | 70.65 | 85.63 | 64.16 | 88.78 | 64.10 | 88.59 | 61.30 | 92.00 | 63.92 | 88.90 | 66.16 | 88.76 |
| | + DAVIS(m) | 69.73 | 88.60 | 74.16 | 84.24 | 65.84 | 89.06 | 64.82 | 89.12 | 69.83 | 89.88 | 65.63 | 89.23 | 68.34 | 88.36 |
| | + DAVIS(μ,σ) | 68.41 | 88.03 | 74.47 | 81.77 | 66.95 | 87.10 | 64.24 | 87.86 | 69.30 | 89.58 | 66.52 | 87.54 | 68.31 | 86.98 |
| | Energy | 75.83 | 85.85 | 44.98 | 90.62 | 29.68 | 95.03 | 48.67 | 91.19 | 9.54 | 98.12 | 29.80 | 95.10 | 39.75 | 92.65 |
| | + DAVIS(m) | 69.61 | 87.56 | 44.33 | 91.00 | 23.33 | 96.26 | 37.87 | 94.11 | 17.62 | 97.02 | 22.48 | 96.40 | 35.87 | 93.73 |
| | + DAVIS(μ,σ) | 62.07 | 89.18 | 43.87 | 90.86 | 22.38 | 96.40 | 33.24 | 94.71 | 14.21 | 97.50 | 21.32 | 96.57 | 32.85 | 94.20 |
| | ReAct | 73.06 | 85.65 | 45.30 | 90.21 | 28.00 | 95.12 | 44.08 | 92.15 | 11.31 | 97.84 | 27.31 | 95.31 | 38.18 | 92.71 |
| | + DAVIS(m) | 70.00 | 87.25 | 45.21 | 90.63 | 24.01 | 96.07 | 37.06 | 94.10 | 19.21 | 96.77 | 23.00 | 96.26 | 36.41 | 93.51 |
| | + DAVIS(μ,σ) | 63.17 | 88.86 | 44.50 | 90.48 | 23.18 | 96.19 | 32.77 | 94.66 | 15.41 | 97.27 | 21.81 | 96.40 | 33.47 | 93.98 |
| | DICE | 62.13 | 87.07 | 50.33 | 89.30 | 27.68 | 95.70 | 49.57 | 90.26 | 2.18 | 99.48 | 29.27 | 95.45 | 36.86 | 92.88 |
| | + DAVIS(m) | 38.71 | 92.48 | 50.33 | 89.34 | 14.89 | 97.42 | 25.05 | 95.84 | 4.39 | 99.21 | 15.28 | 97.40 | 24.78 | 95.28 |
| | + DAVIS(μ,σ) | 37.14 | 92.79 | 50.38 | 89.05 | 15.39 | 97.34 | 24.45 | 96.00 | 3.56 | 99.32 | 15.38 | 97.34 | 24.38 | 95.31 |
| | ReAct+DICE | 57.05 | 87.51 | 52.54 | 88.35 | 23.84 | 96.17 | 44.52 | 92.06 | 2.39 | 99.48 | 23.85 | 96.11 | 34.03 | 93.28 |
| | + DAVIS(m) | 37.17 | 92.71 | 52.01 | 88.47 | 15.63 | 97.29 | 23.42 | 96.08 | 4.58 | 99.17 | 15.53 | 97.34 | 24.72 | 95.17 |
| | + DAVIS(μ,σ) | 36.32 | 92.87 | 52.18 | 88.15 | 16.12 | 97.18 | 23.32 | 96.17 | 3.85 | 99.28 | 15.62 | 97.26 | 24.57 | 95.15 |
| | ASH | 46.23 | 91.48 | 61.75 | 84.40 | 41.46 | 93.36 | 43.87 | 91.50 | 7.28 | 98.68 | 40.97 | 93.46 | 40.26 | 92.15 |
| | + DAVIS(m) | 31.33 | 94.41 | 59.92 | 84.29 | 31.64 | 94.39 | 31.81 | 94.50 | 11.90 | 97.94 | 32.38 | 94.67 | 33.16 | 93.37 |
| | + DAVIS(μ,σ) | 30.55 | 94.66 | 61.29 | 83.62 | 31.36 | 94.39 | 30.71 | 94.73 | 10.62 | 98.15 | 31.60 | 94.80 | 32.69 | 93.39 |
| | Scale | 56.48 | 89.83 | 59.43 | 85.26 | 34.81 | 94.33 | 42.36 | 91.91 | 9.82 | 98.23 | 34.46 | 94.50 | 39.56 | 92.34 |
| | + DAVIS(m) | 47.67 | 91.98 | 55.06 | 86.26 | 22.92 | 95.87 | 29.06 | 95.24 | 16.04 | 97.31 | 21.92 | 96.21 | 32.11 | 93.81 |
| | + DAVIS(μ,σ) | 45.04 | 92.63 | 57.52 | 85.64 | 23.88 | 95.72 | 28.62 | 95.35 | 14.31 | 97.59 | 22.63 | 96.12 | 32.00 | 93.84 |

Table 19: *Detailed results of post-hoc methods combined with DAVIS using **ResNet-18** and **ResNet-34** trained on **CIFAR-10**. ↑ indicates higher is better; ↓ indicates lower is better. The symbols denote the statistic used: μ (mean), σ (std. deviation), m (maximum)*

| Model | Combined Method | SVHN FPR95↓ | SVHN AUROC↑ | Place365 FPR95↓ | Place365 AUROC↑ | iSUN FPR95↓ | iSUN AUROC↑ | Textures FPR95↓ | Textures AUROC↑ | LSUN-c FPR95↓ | LSUN-c AUROC↑ | LSUN-r FPR95↓ | LSUN-r AUROC↑ | Average FPR95↓ | Average AUROC↑ |
|---|---|---|---|---|---|---|---|---|---|---|---|---|---|---|---|
| ResNet-18 | MSP | 74.26 | 83.20 | 82.49 | 75.32 | 85.58 | 70.20 | 84.89 | 74.02 | 70.79 | 82.78 | 84.36 | 71.45 | 80.40 | 76.16 |
| | + DAVIS(m) | 79.01 | 83.56 | 86.57 | 73.79 | 87.79 | 72.86 | 83.39 | 77.95 | 79.77 | 81.23 | 87.53 | 73.03 | 84.01 | 77.07 |
| | + DAVIS(μ,σ) | 78.90 | 82.88 | 86.06 | 72.47 | 87.68 | 70.35 | 83.07 | 76.64 | 78.68 | 80.12 | 87.38 | 70.78 | 83.63 | 75.54 |
| | Energy | 66.64 | 89.53 | 81.23 | 76.84 | 73.67 | 82.01 | 85.30 | 75.68 | 48.01 | 91.63 | 70.30 | 83.38 | 70.86 | 83.18 |
| | + DAVIS(m) | 38.40 | 94.25 | 77.62 | 78.98 | 56.43 | 89.99 | 55.73 | 89.14 | 35.90 | 94.17 | 55.87 | 83.95 | 53.32 | 89.41 |
| | + DAVIS(μ,σ) | 42.01 | 93.75 | 78.99 | 77.58 | 61.13 | 88.60 | 59.89 | 87.89 | 38.56 | 93.48 | 59.76 | 88.86 | 56.72 | 88.36 |
| | ReAct | 55.03 | 91.95 | 79.78 | 77.48 | 58.66 | 87.78 | 60.90 | 87.94 | 47.42 | 91.22 | 54.78 | 88.77 | 59.43 | 87.52 |
| | + DAVIS(m) | 42.98 | 93.55 | 77.00 | 78.94 | 55.73 | 89.57 | 54.18 | 89.53 | 40.68 | 92.67 | 54.12 | 89.84 | 54.12 | 89.02 |
| | + DAVIS(μ,σ) | 44.43 | 93.37 | 78.24 | 78.18 | 57.83 | 88.94 | 56.26 | 89.06 | 42.53 | 92.19 | 55.52 | 89.5 | 55.8 | 88.54 |
| | DICE | 41.18 | 92.98 | 81.82 | 76.02 | 66.22 | 84.20 | 75.50 | 76.27 | 12.21 | 97.70 | 64.48 | 85.15 | 56.90 | 85.39 |
| | + DAVIS(m) | 10.77 | 97.91 | 80.06 | 77.59 | 33.36 | 94.13 | 31.52 | 93.52 | 7.31 | 98.53 | 37.27 | 93.39 | 33.38 | 92.51 |
| | + DAVIS(μ,σ) | 12.40 | 97.59 | 80.46 | 76.21 | 38.77 | 92.69 | 36.74 | 92.09 | 7.78 | 98.42 | 40.98 | 92.27 | 36.19 | 91.54 |
| | ReAct+DICE | 36.18 | 93.65 | 86.43 | 71.92 | 59.57 | 88.43 | 48.46 | 87.59 | 11.78 | 97.59 | 60.22 | 88.45 | 50.44 | 87.94 |
| | + DAVIS(m) | 11.37 | 97.76 | 82.96 | 75.31 | 35.09 | 93.96 | 29.26 | 93.39 | 8.53 | 98.24 | 38.86 | 93.25 | 34.34 | 91.99 |
| | + DAVIS(μ,σ) | 13.03 | 97.49 | 82.67 | 74.91 | 38.04 | 93.08 | 31.99 | 92.61 | 8.91 | 98.13 | 40.32 | 92.68 | 35.83 | 91.48 |
| | ASH | 29.10 | 95.46 | 82.56 | 75.38 | 67.09 | 85.01 | 56.49 | 87.80 | 27.06 | 95.55 | 64.72 | 85.62 | 54.50 | 87.47 |
| | + DAVIS(m) | 12.41 | 97.87 | 81.30 | 75.93 | 46.38 | 91.18 | 26.58 | 94.94 | 18.44 | 96.99 | 48.47 | 90.45 | 38.93 | 91.23 |
| | + DAVIS(μ,σ) | 13.44 | 97.70 | 81.85 | 75.09 | 49.06 | 90.36 | 28.55 | 94.52 | 19.93 | 96.69 | 50.46 | 89.97 | 40.55 | 90.72 |
| | Scale | 22.12 | 96.38 | 81.96 | 74.95 | 61.62 | 86.65 | 44.50 | 90.72 | 18.62 | 96.78 | 59.76 | 86.74 | 48.10 | 88.70 |
| | + DAVIS(m) | 10.40 | 98.10 | 82.86 | 74.60 | 45.08 | 91.15 | 24.10 | 95.04 | 14.09 | 97.49 | 48.37 | 89.96 | 37.48 | 91.06 |
| | + DAVIS(μ,σ) | 11.43 | 97.94 | 83.24 | 74.00 | 47.37 | 90.57 | 25.64 | 94.78 | 15.35 | 97.28 | 49.16 | 89.69 | 38.70 | 90.71 |
| ResNet-34 | MSP | 69.72 | 83.11 | 82.28 | 75.84 | 83.04 | 76.64 | 84.95 | 74.24 | 76.57 | 81.44 | 81.54 | 77.19 | 79.68 | 78.08 |
| | + DAVIS(m) | 74.60 | 84.67 | 85.45 | 74.67 | 84.62 | 76.94 | 82.96 | 77.55 | 80.34 | 81.10 | 85.69 | 76.53 | 82.28 | 78.54 |
| | + DAVIS(μ,σ) | 75.75 | 83.75 | 85.37 | 73.07 | 85.37 | 74.93 | 83.32 | 76.04 | 80.85 | 79.93 | 85.70 | 74.53 | 82.73 | 77.04 |
| | Energy | 57.79 | 89.80 | 81.17 | 77.25 | 71.83 | 84.14 | 86.77 | 75.82 | 55.56 | 89.92 | 68.70 | 84.93 | 70.30 | 83.64 |
| | + DAVIS(m) | 29.56 | 95.19 | 77.68 | 78.81 | 55.37 | 90.51 | 56.29 | 89.14 | 38.37 | 93.37 | 56.27 | 90.23 | 52.26 | 89.54 |
| | + DAVIS(μ,σ) | 31.07 | 95.00 | 80.04 | 77.53 | 59.46 | 89.50 | 59.75 | 88.31 | 41.21 | 93.00 | 60.05 | 89.32 | 55.26 | 88.78 |
| | ReAct | 30.77 | 94.47 | 77.91 | 78.11 | 64.45 | 85.01 | 62.13 | 86.27 | 47.86 | 90.72 | 64.13 | 85.24 | 57.87 | 86.64 |
| | + DAVIS(m) | 28.40 | 95.20 | 77.07 | 78.92 | 61.06 | 88.14 | 57.04 | 88.54 | 43.89 | 91.94 | 61.26 | 87.90 | 54.79 | 88.44 |
| | + DAVIS(μ,σ) | 29.26 | 95.18 | 78.97 | 78.32 | 64.08 | 87.49 | 60.05 | 88.15 | 46.02 | 91.80 | 63.78 | 87.36 | 57.03 | 88.05 |
| | DICE | 25.88 | 95.11 | 80.75 | 77.18 | 65.76 | 85.44 | 74.73 | 78.11 | 18.31 | 96.55 | 65.59 | 85.30 | 55.17 | 86.28 |
| | + DAVIS(m) | 7.96 | 98.40 | 79.29 | 77.25 | 36.80 | 93.62 | 28.32 | 94.05 | 8.62 | 98.22 | 42.47 | 92.46 | 33.91 | 92.33 |
| | + DAVIS(μ,σ) | 8.25 | 98.43 | 81.22 | 76.35 | 41.29 | 92.83 | 33.12 | 93.27 | 9.78 | 98.15 | 46.38 | 91.82 | 36.67 | 91.81 |
| | ReAct+DICE | 20.70 | 95.78 | 89.29 | 69.02 | 81.03 | 78.47 | 49.77 | 85.31 | 26.09 | 94.34 | 86.08 | 76.00 | 58.83 | 83.16 |
| | + DAVIS(m) | 9.22 | 98.14 | 84.00 | 74.77 | 49.02 | 90.86 | 29.34 | 92.95 | 12.21 | 97.68 | 56.40 | 89.27 | 40.03 | 90.61 |
| | + DAVIS(μ,σ) | 8.42 | 98.29 | 83.28 | 74.99 | 50.44 | 90.44 | 31.65 | 92.64 | 11.99 | 97.70 | 56.76 | 89.02 | 40.42 | 90.51 |
| | ASH | 23.64 | 95.92 | 82.37 | 75.92 | 60.77 | 87.53 | 59.08 | 87.60 | 41.57 | 93.06 | 61.41 | 87.24 | 54.81 | 87.88 |
| | + DAVIS(m) | 7.78 | 98.52 | 81.29 | 75.24 | 41.99 | 91.95 | 25.76 | 94.92 | 21.30 | 96.37 | 47.56 | 90.87 | 37.61 | 91.31 |
| | + DAVIS(μ,σ) | 8.20 | 98.45 | 82.54 | 74.72 | 44.91 | 91.39 | 28.35 | 94.52 | 23.96 | 95.95 | 49.24 | 90.44 | 39.53 | 90.91 |
| | Scale | 13.68 | 97.51 | 80.72 | 75.87 | 57.87 | 87.14 | 46.35 | 90.27 | 28.17 | 95.07 | 61.31 | 86.01 | 48.02 | 88.64 |
| | + DAVIS(m) | 6.55 | 98.75 | 82.68 | 73.88 | 49.14 | 89.47 | 24.08 | 94.64 | 16.16 | 97.05 | 57.11 | 87.44 | 39.29 | 90.21 |
| | + DAVIS(μ,σ) | 6.34 | 98.76 | 82.73 | 73.83 | 49.69 | 89.44 | 24.95 | 94.48 | 17.77 | 96.79 | 56.90 | 87.62 | 39.73 | 90.15 |

Table 20: Detailed results of post-hoc methods combined with *DAVIS DenseNet-101* and *MobileNet-v2* trained on *CIFAR-100*. ↑ indicates higher is better; ↓ indicates lower is better. The symbols denote the statistic used: $\mu$ (mean), $\sigma$ (std. deviation), $m$ (maximum)

| Model | Combined Method | SVHN | | Place365 | | iSUN | | Textures | | LSUN-c | | LSUN-r | | Average | |
|---|---|---|---|---|---|---|---|---|---|---|---|---|---|---|---|
| | | FPR95↓ | AUROC↑ | FPR95↓ | AUROC↑ | FPR95↓ | AUROC↑ | FPR95↓ | AUROC↑ | FPR95↓ | AUROC↑ | FPR95↓ | AUROC↑ | FPR95↓ | AUROC↑ |
| DenseNet-101 | MSP | 81.38 | 75.71 | 82.62 | 74.04 | 84.12 | 68.22 | 86.95 | 68.37 | 51.82 | 87.93 | 81.34 | 69.51 | 78.04 | 73.96 |
| | + DAVIS($m$) | 84.27 | 77.18 | 88.63 | 69.53 | 86.77 | 72.39 | 86.15 | 72.76 | 80.46 | 79.07 | 86.99 | 71.73 | 85.54 | 73.78 |
| | + DAVIS($\mu,\sigma$) | 83.27 | 78.70 | 87.90 | 70.83 | 87.57 | 69.56 | 85.85 | 72.53 | 77.17 | 80.93 | 88.06 | 69.25 | 84.97 | 73.63 |
| | Energy | 70.99 | 86.66 | 77.12 | 76.94 | 64.28 | 83.92 | 83.60 | 67.47 | 11.45 | 97.89 | 56.08 | 86.84 | 60.59 | 83.29 |
| | + DAVIS($m$) | 52.32 | 89.41 | 73.49 | 79.62 | 35.76 | 93.37 | 49.88 | 88.07 | 18.58 | 96.44 | 37.27 | 92.98 | 44.55 | 89.98 |
| | + DAVIS($\mu,\sigma$) | 45.23 | 91.91 | 72.11 | 79.78 | 42.21 | 92.16 | 55.85 | 85.47 | 13.92 | 97.53 | 40.52 | 92.45 | 44.97 | 89.88 |
| | ReAct | 67.12 | 87.20 | 77.75 | 76.18 | 56.39 | 89.46 | 75.98 | 79.16 | 13.26 | 97.53 | 49.92 | 90.94 | 56.74 | 86.74 |
| | + DAVIS($m$) | 52.11 | 89.87 | 75.65 | 78.41 | 37.90 | 93.24 | 51.42 | 87.49 | 22.32 | 95.63 | 39.01 | 93.03 | 46.40 | 89.61 |
| | + DAVIS($\mu,\sigma$) | 44.85 | 92.25 | 74.50 | 78.63 | 43.84 | 92.30 | 55.18 | 86.60 | 17.47 | 96.83 | 41.97 | 92.74 | 46.30 | 89.89 |
| | DICE | 33.87 | 93.97 | 79.95 | 76.75 | 47.76 | 89.61 | 63.42 | 73.33 | 0.79 | 99.76 | 43.65 | 91.00 | 44.91 | 87.40 |
| | + DAVIS($m$) | 27.97 | 94.91 | 86.13 | 76.61 | 36.01 | 93.97 | 30.32 | 93.28 | 7.58 | 98.56 | 41.90 | 93.13 | 38.32 | 91.74 |
| | + DAVIS($\mu,\sigma$) | 20.30 | 96.20 | 79.52 | 78.28 | 29.69 | 94.63 | 34.10 | 91.24 | 2.63 | 99.37 | 32.01 | 94.20 | 33.04 | 92.32 |
| | ReAct+DICE | 28.01 | 95.38 | 83.56 | 74.73 | 37.02 | 93.92 | 46.93 | 86.09 | 0.68 | 99.79 | 37.20 | 93.93 | 38.90 | 90.64 |
| | + DAVIS($m$) | 23.75 | 95.30 | 86.53 | 74.82 | 36.01 | 93.77 | 28.30 | 93.55 | 7.59 | 98.53 | 41.77 | 92.98 | 37.32 | 91.49 |
| | + DAVIS($\mu,\sigma$) | 17.65 | 96.60 | 82.54 | 76.47 | 29.84 | 94.80 | 30.57 | 92.97 | 2.73 | 99.36 | 32.66 | 94.38 | 32.66 | 92.43 |
| | ASH | 10.32 | 97.99 | 85.93 | 71.95 | 39.69 | 92.04 | 35.67 | 91.76 | 5.43 | 98.98 | 42.89 | 91.30 | 36.66 | 90.67 |
| | + DAVIS($m$) | 17.78 | 96.87 | 81.38 | 76.97 | 34.35 | 93.71 | 24.22 | 95.11 | 10.20 | 98.18 | 38.96 | 92.90 | 34.48 | 92.29 |
| | + DAVIS($\mu,\sigma$) | 12.12 | 97.72 | 79.01 | 77.28 | 33.61 | 93.56 | 26.33 | 94.75 | 6.26 | 98.81 | 37.11 | 92.96 | 32.41 | 92.51 |
| | Scale | 16.26 | 97.05 | 78.54 | 76.97 | 43.56 | 91.21 | 45.60 | 87.23 | 3.23 | 99.30 | 42.69 | 91.02 | 38.31 | 90.46 |
| | + DAVIS($m$) | 19.45 | 96.40 | 81.18 | 76.81 | 36.21 | 93.26 | 25.11 | 94.71 | 11.07 | 97.97 | 44.32 | 91.81 | 36.22 | 91.83 |
| | + DAVIS($\mu,\sigma$) | 13.46 | 97.53 | 78.75 | 77.82 | 32.68 | 93.77 | 24.29 | 95.06 | 6.09 | 98.85 | 38.36 | 92.74 | 32.27 | 92.63 |
| MobileNet-v2 | MSP | 80.14 | 76.09 | 84.23 | 72.62 | 87.72 | 70.77 | 86.67 | 70.36 | 77.11 | 76.04 | 87.10 | 70.83 | 83.83 | 72.78 |
| | + DAVIS($m$) | 83.57 | 75.75 | 88.75 | 68.85 | 89.31 | 69.72 | 85.27 | 72.89 | 81.94 | 75.71 | 88.63 | 70.04 | 86.24 | 72.16 |
| | + DAVIS($\mu,\sigma$) | 84.66 | 76.01 | 89.52 | 68.86 | 90.10 | 69.46 | 86.29 | 73.12 | 83.81 | 75.20 | 89.55 | 69.55 | 87.32 | 72.03 |
| | Energy | 69.65 | 85.98 | 81.26 | 75.21 | 78.17 | 83.33 | 80.02 | 78.63 | 50.19 | 89.42 | 76.59 | 84.06 | 72.65 | 82.77 |
| | + DAVIS($m$) | 41.94 | 92.00 | 81.49 | 73.31 | 64.52 | 86.40 | 44.22 | 90.68 | 34.22 | 93.72 | 65.30 | 86.10 | 55.28 | 87.04 |
| | + DAVIS($\mu,\sigma$) | 40.00 | 92.73 | 81.64 | 73.23 | 64.99 | 86.36 | 42.70 | 91.25 | 34.67 | 93.75 | 66.30 | 86.04 | 55.05 | 87.23 |
| | ReAct | 28.41 | 95.15 | 79.46 | 74.05 | 62.45 | 87.65 | 47.70 | 90.19 | 41.25 | 92.31 | 62.16 | 88.06 | 53.57 | 87.90 |
| | + DAVIS($m$) | 34.94 | 93.57 | 81.11 | 73.91 | 64.49 | 86.36 | 44.15 | 91.08 | 39.10 | 92.48 | 64.48 | 86.37 | 54.71 | 87.29 |
| | + DAVIS($\mu,\sigma$) | 33.27 | 94.07 | 80.85 | 73.99 | 65.01 | 86.31 | 43.14 | 91.49 | 38.80 | 92.53 | 65.63 | 86.26 | 54.45 | 87.44 |
| | DICE | 55.62 | 87.54 | 83.25 | 74.25 | 81.23 | 79.75 | 65.02 | 81.93 | 18.17 | 96.24 | 85.40 | 77.90 | 64.78 | 82.93 |
| | + DAVIS($m$) | 27.03 | 95.00 | 87.44 | 68.73 | 73.54 | 82.05 | 27.22 | 94.08 | 11.28 | 97.90 | 82.10 | 78.79 | 51.43 | 86.09 |
| | + DAVIS($\mu,\sigma$) | 27.03 | 95.08 | 86.73 | 69.45 | 73.04 | 82.35 | 27.62 | 94.01 | 12.08 | 97.84 | 80.59 | 79.29 | 51.18 | 86.34 |
| | ReAct+DICE | 22.57 | 96.05 | 89.57 | 66.14 | 81.05 | 76.71 | 29.54 | 93.30 | 10.19 | 98.00 | 89.71 | 71.34 | 53.77 | 83.59 |
| | + DAVIS($m$) | 19.70 | 96.39 | 88.30 | 67.95 | 74.80 | 81.54 | 24.10 | 94.83 | 11.39 | 97.90 | 83.86 | 77.92 | 50.36 | 86.09 |
| | + DAVIS($\mu,\sigma$) | 19.52 | 96.53 | 87.24 | 69.36 | 74.05 | 82.11 | 24.95 | 94.89 | 11.53 | 97.97 | 82.37 | 78.79 | 49.94 | 86.61 |
| | ASH | 21.90 | 96.46 | 85.12 | 69.51 | 70.46 | 82.84 | 34.80 | 92.65 | 24.14 | 95.56 | 73.46 | 81.22 | 51.65 | 86.37 |
| | + DAVIS($m$) | 11.62 | 97.72 | 87.30 | 66.62 | 65.66 | 83.89 | 22.04 | 95.40 | 20.60 | 96.20 | 70.72 | 81.92 | 46.32 | 86.96 |
| | + DAVIS($\mu,\sigma$) | 12.22 | 97.74 | 86.54 | 67.52 | 65.85 | 84.13 | 22.46 | 95.43 | 20.60 | 96.28 | 70.41 | 82.28 | 46.35 | 87.23 |
| | Scale | 22.36 | 96.25 | 81.46 | 73.30 | 68.52 | 84.26 | 37.62 | 91.84 | 21.43 | 96.07 | 71.77 | 82.83 | 50.53 | 87.43 |
| | + DAVIS($m$) | 9.31 | 98.11 | 87.25 | 67.44 | 68.20 | 83.10 | 21.15 | 95.30 | 15.59 | 97.13 | 74.82 | 80.45 | 46.05 | 86.92 |
| | + DAVIS($\mu,\sigma$) | 9.65 | 98.09 | 86.41 | 68.38 | 67.73 | 83.51 | 21.44 | 95.45 | 15.58 | 97.26 | 73.79 | 81.01 | 45.77 | 87.28 |

## H  ALTERNATE STATISTICS: MEDIAN AND SHANNON ENTROPY

To justify our choice of mean, variance, and maximum statistics, we conducted an ablation study exploring two alternatives: the median and Shannon entropy. While an activation map encodes many statistical properties, for our framework to be effective, the chosen statistic must produce a distinctive signature for ID versus OOD samples.

**Median.** We begin by extracting the *median* from each activation map of $g(\mathbf{x}) \in \mathbb{R}^{n \times k \times k}$, transforming it into an $n$-dimensional feature vector $h(\mathbf{x}) \in \mathbb{R}^n$ using global median pooling, as defined in Equation 19:

$$h(\mathbf{x}) = \texttt{median}\,(g(\mathbf{x})) \tag{19}$$

Here, $\texttt{median}$ denotes a global median pooling operation applied independently to each of the $n$ activation maps in $g(\mathbf{x})$.

**Shannon Entropy**. In addition to the median, we compute the *Shannon entropy* for each activation map. For the $i$-th channel activation $g_i(\mathbf{x}) \in \mathbb{R}^{k \times k}$, the entropy is computed as shown in Equation 21. To do so, we first flatten $g_i(\mathbf{x})$ into a vector of length $k^2$, and normalize it to define a discrete probability distribution $p_{ij}$, as described in Equation 20. By collecting the entropy values across all channels, we obtain the final feature representation $h(\mathbf{x}) \in \mathbb{R}^n$, as defined in Equation 22 .

$$p_{ij} = \frac{g_i(\mathbf{x})_j}{\sum_{l=1}^{k^2} g_i(\mathbf{x})_l}, \quad j = 1, \ldots, k^2 \tag{20}$$

$$\texttt{entropy}_i(\mathbf{x}) = -\sum_{j=1}^{k^2} p_{ij} \log p_{ij} \tag{21}$$

$$h(\mathbf{x}) = \texttt{entropy}\,(g(\mathbf{x})) = [\texttt{entropy}_1(\mathbf{x}), \ldots, \texttt{entropy}_n(\mathbf{x})]^\top \tag{22}$$

The performance of these alternate statistics is presented in Table 21 (ImageNet) and Table 22 (CIFAR). The results are unambiguous: representations built from the maximum and mean/variance statistics are overwhelmingly superior to those from the median and entropy. The entropy-based features, in particular, perform poorly, with FPR95 scores often approaching 100%, rendering them ineffective for OOD detection. The median-based features also struggle significantly, with FPR95 scores consistently above 80% on ImageNet.

The reason for this poor performance is illustrated in the distributions shown in Figure 7. Both median and entropy produce feature representations where OOD samples consistently yield higher-magnitude values than ID samples. This creates an "inverted separation" that is fundamentally at odds with standard scoring functions (like energy), which assume higher scores correspond to ID samples. This confirms our central hypothesis: effective OOD detection requires statistics that specifically amplify the characteristic signals of ID samples relative to OOD samples, a property that the maximum and variance possess, but the median and entropy do not in this context.

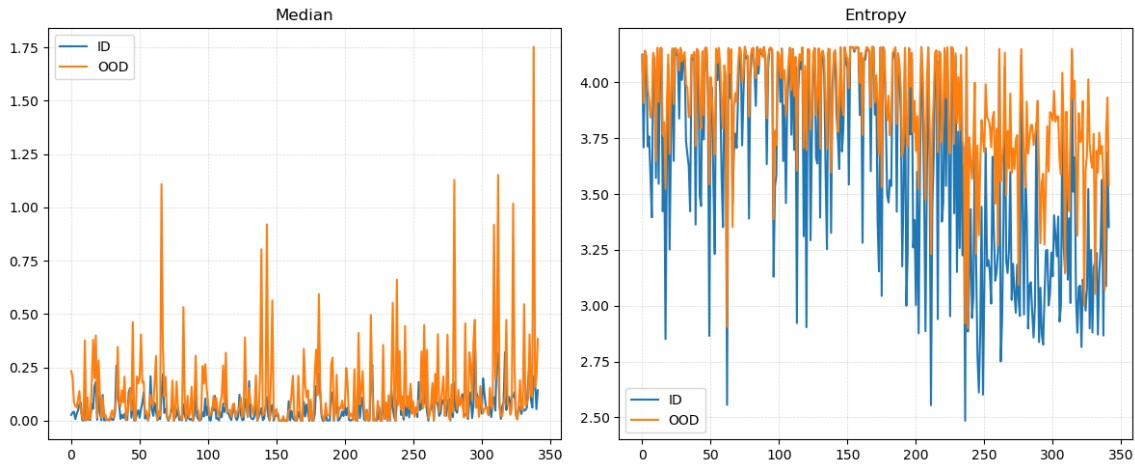

Figure 7: *Illustration of penultimate layer features derived from median (**left**) and entropy (**right**) statistics. For both measures, out-of-distribution (OOD) samples (Texture) exhibit consistently higher values than in-distribution (ID) samples (CIFAR-100), creating an "inverted separation" that is challenging for standard OOD scoring. (Model: DenseNet-101).*

Table 21: *Performance of various pre-pooling statistics on the ImageNet-1K benchmark, evaluated across DenseNet-121, ResNet-50, MobileNet-v2, and EfficientNet-b0 architectures. To isolate the effect of each statistic, results are based solely on the energy score without any additional post-hoc methods applied. ↓ indicates that lower values are better, while ↑ indicates that higher values are better.*

| Model | Statistics | SUN | | Place | | Texture | | iNaturalist | | Average | |
|---|---|---|---|---|---|---|---|---|---|---|---|
| | | FPR95 ↓ | AUROC ↑ | FPR95 ↓ | AUROC ↑ | FPR95 ↓ | AUROC ↑ | FPR95 ↓ | AUROC ↑ | FPR95 ↓ | AUROC ↑ |
| DenseNet-121 | $\mu(\mathbf{x})$ | 52.51 | 87.27 | 58.24 | 85.05 | 52.22 | 85.42 | 39.75 | 92.66 | 50.68 | 87.60 |
| | $\mu(\mathbf{x}) + 2.0\sigma(\mathbf{x})$ | 48.40 | 88.64 | 58.38 | 84.94 | 25.37 | 94.14 | 30.31 | 93.79 | 40.62 | 90.38 |
| | $m(\mathbf{x})$ | 48.55 | 88.03 | 59.36 | 83.53 | 27.54 | 93.62 | 33.74 | 92.20 | 42.30 | 89.34 |
| | $median(\mathbf{x})$ | 91.02 | 73.85 | 88.85 | 73.68 | 87.43 | 67.32 | 90.3 | 78.44 | 89.4 | 73.32 |
| | $entropy(\mathbf{x})$ | 99.84 | 22.68 | 99.82 | 25.15 | 99.65 | 39.69 | 99.94 | 20.19 | 99.81 | 26.93 |
| ResNet-50 | $\mu(\mathbf{x})$ | 58.82 | 86.58 | 65.99 | 83.96 | 52.43 | 86.72 | 53.74 | 90.62 | 57.74 | 86.97 |
| | $\mu(\mathbf{x}) + 2.0\sigma(\mathbf{x})$ | 54.71 | 86.47 | 64.06 | 82.55 | 21.13 | 95.32 | 34.50 | 92.95 | 43.60 | 89.33 |
| | $m(\mathbf{x})$ | 53.12 | 85.49 | 63.58 | 80.54 | 20.94 | 95.09 | 36.11 | 91.29 | 43.44 | 88.10 |
| | $median(\mathbf{x})$ | 82.74 | 79.96 | 83.14 | 78.55 | 82.66 | 76.52 | 86.80 | 82.26 | 83.83 | 79.32 |
| | $entropy(\mathbf{x})$ | 99.84 | 27.69 | 99.65 | 30.39 | 99.36 | 43.69 | 99.69 | 36.11 | 99.64 | 34.47 |
| MobileNet-v2 | $\mu(\mathbf{x})$ | 59.60 | 86.16 | 66.36 | 83.15 | 54.82 | 86.57 | 55.33 | 90.37 | 59.03 | 86.56 |
| | $\mu(\mathbf{x}) + 2.0\sigma(\mathbf{x})$ | 55.47 | 87.15 | 64.82 | 83.18 | 25.78 | 94.33 | 38.94 | 92.98 | 46.25 | 89.41 |
| | $m(\mathbf{x})$ | 53.64 | 87.26 | 63.61 | 82.93 | 25.87 | 94.03 | 40.25 | 92.22 | 45.84 | 89.11 |
| | $median(\mathbf{x})$ | 85.12 | 78.08 | 85.02 | 76.46 | 83.88 | 74.62 | 86.35 | 80.98 | 85.09 | 77.53 |
| | $entropy(\mathbf{x})$ | 99.61 | 19.59 | 99.12 | 25.53 | 99.77 | 18.35 | 99.58 | 27.55 | 99.52 | 22.75 |
| EfficientNet-b0 | $\mu(\mathbf{x})$ | 85.01 | 72.86 | 86.06 | 70.99 | 75.99 | 75.86 | 78.91 | 79.78 | 81.49 | 74.87 |
| | $\mu(\mathbf{x}) + 2.0\sigma(\mathbf{x})$ | 61.85 | 84.67 | 71.31 | 79.01 | 12.18 | 97.52 | 48.25 | 88.35 | 48.40 | 87.39 |
| | $m(\mathbf{x})$ | 69.42 | 80.74 | 79.09 | 73.67 | 15.82 | 96.38 | 63.68 | 80.63 | 57.00 | 82.85 |
| | $median(\mathbf{x})$ | 99.92 | 17.72 | 99.71 | 22.15 | 100.00 | 3 .80 | 100.00 | 10.63 | 99.91 | 13.57 |
| | $entropy(\mathbf{x})$ | 97.25 | 47.42 | 97.96 | 41.37 | 81.35 | 76.51 | 98.14 | 38.82 | 93.67 | 51.03 |

Table 22: *Performance of various pre-pooling statistics on the CIFAR-10 and CIFAR-100 benchmarks, evaluated across ResNet-18, DenseNet-101, and MobileNet-v2 architectures. To isolate the effect of each statistic, results are based solely on the energy score without any additional post-hoc methods applied.* ↓ *indicates that lower values are better, while* ↑ *indicates that higher values are better.*

| Dataset | Model | Statistics | SVHN FPR95↓ | SVHN AUROC↑ | Place365 FPR95↓ | Place365 AUROC↑ | iSUN FPR95↓ | iSUN AUROC↑ | Textures FPR95↓ | Textures AUROC↑ | LSUN-c FPR95↓ | LSUN-c AUROC↑ | LSUN-r FPR95↓ | LSUN-r AUROC↑ | Average FPR95↓ | Average AUROC↑ |
|---|---|---|---|---|---|---|---|---|---|---|---|---|---|---|---|---|
| CIFAR-10 | ResNet-18 | $\mu(x)$ | 44.32 | 94.04 | 41.31 | 91.73 | 35.46 | 94.64 | 50.39 | 91.12 | 9.77 | 98.19 | 32.41 | 95.16 | 35.61 | 94.14 |
| | | $\mu(x) + 3.0\sigma(x)$ | 19.83 | 96.47 | 36.11 | 93.16 | 20.48 | 96.90 | 26.77 | 95.89 | 5.62 | 98.79 | 17.75 | 97.21 | 21.09 | 96.40 |
| | | $m(x)$ | 19.81 | 96.29 | 32.32 | 93.73 | 15.79 | 97.37 | 21.90 | 96.44 | 5.83 | 98.73 | 13.63 | 97.61 | 18.21 | 96.69 |
| | | $median(x)$ | 99.57 | 31.77 | 88.60 | 63.39 | 96.20 | 50.98 | 99.22 | 31.82 | 95.09 | 65.21 | 95.68 | 51.78 | 95.73 | 49.16 |
| | | $entropy(x)$ | 98.01 | 78.03 | 98.99 | 52.54 | 99.99 | 43.91 | 99.52 | 65.07 | 99.90 | 62.83 | 99.98 | 41.74 | 99.40 | 57.35 |
| | ResNet-34 | $\mu(x)$ | 35.44 | 93.76 | 38.15 | 92.27 | 19.90 | 96.87 | 42.52 | 92.54 | 3.38 | 99.10 | 16.86 | 97.20 | 26.04 | 95.29 |
| | | $\mu(x) + 3.0\sigma(x)$ | 28.96 | 94.90 | 32.66 | 93.69 | 11.31 | 97.88 | 23.07 | 96.39 | 3.47 | 99.22 | 10.35 | 98.04 | 18.30 | 96.69 |
| | | $m(x)$ | 29.29 | 94.59 | 29.59 | 94.21 | 10.17 | 98.03 | 20.23 | 96.73 | 4.00 | 99.10 | 9.45 | 98.16 | 17.12 | 96.81 |
| | | $median(x)$ | 99.80 | 24.42 | 96.12 | 55.41 | 99.24 | 48.47 | 99.86 | 27.82 | 97.35 | 58.12 | 98.91 | 51.98 | 98.55 | 44.37 |
| | | $entropy(x)$ | 80.63 | 89.37 | 97.05 | 62.50 | 99.73 | 61.00 | 96.52 | 79.21 | 98.92 | 78.69 | 99.66 | 60.31 | 95.42 | 71.85 |
| | DenseNet-101 | $\mu(x)$ | 37.91 | 93.59 | 36.38 | 92.39 | 7.83 | 98.23 | 43.85 | 90.49 | 1.95 | 99.47 | 7.34 | 98.34 | 22.54 | 95.42 |
| | | $\mu(x) + 3.0\sigma(x)$ | 24.75 | 95.93 | 32.08 | 93.43 | 5.77 | 98.64 | 25.23 | 95.84 | 5.13 | 98.94 | 5.63 | 98.70 | 16.43 | 96.91 |
| | | $m(x)$ | 30.50 | 94.50 | 33.04 | 93.28 | 7.81 | 98.29 | 25.16 | 95.85 | 9.95 | 98.19 | 7.32 | 98.37 | 18.96 | 96.41 |
| | | $median(x)$ | 96.80 | 78.31 | 98.70 | 55.92 | 99.78 | 72.66 | 98.55 | 70.95 | 99.49 | 67.91 | 99.76 | 72.17 | 98.85 | 69.65 |
| | | $entropy(x)$ | 92.28 | 69.08 | 83.00 | 75.15 | 87.27 | 79.73 | 95.92 | 58.62 | 17.72 | 96.86 | 87.34 | 79.75 | 77.26 | 76.53 |
| | MobileNet-v2 | $\mu(x)$ | 75.83 | 85.85 | 44.98 | 90.62 | 29.68 | 95.03 | 48.67 | 91.19 | 9.54 | 98.12 | 29.80 | 95.10 | 39.75 | 92.65 |
| | | $\mu(x) + 3.0\sigma(x)$ | 62.07 | 89.18 | 43.87 | 90.86 | 22.38 | 96.40 | 33.24 | 94.71 | 14.21 | 97.50 | 21.32 | 96.57 | 32.85 | 94.20 |
| | | $m(x)$ | 69.61 | 87.56 | 44.33 | 91.00 | 23.33 | 96.26 | 37.87 | 94.11 | 17.62 | 97.02 | 22.48 | 96.40 | 35.87 | 93.73 |
| | | $median(x)$ | 89.92 | 80.06 | 60.84 | 85.74 | 54.69 | 89.94 | 66.37 | 84.49 | 12.42 | 97.68 | 54.48 | 89.74 | 56.45 | 87.94 |
| | | $entropy(x)$ | 99.75 | 80.37 | 99.68 | 55.79 | 99.94 | 63.55 | 99.75 | 71.94 | 99.56 | 72.44 | 99.95 | 62.49 | 99.77 | 67.76 |
| CIFAR-100 | ResNet-18 | $\mu(x)$ | 66.64 | 89.53 | 81.23 | 76.84 | 73.67 | 82.01 | 85.30 | 75.68 | 48.01 | 91.63 | 70.30 | 83.38 | 70.86 | 83.18 |
| | | $\mu(x) + 3.0\sigma(x)$ | 42.01 | 93.75 | 78.99 | 77.58 | 61.13 | 88.60 | 59.89 | 87.89 | 38.56 | 93.48 | 59.76 | 88.86 | 56.72 | 88.36 |
| | | $\mu(x) + 4.0\sigma(x)$ | 40.43 | 93.92 | 78.88 | 77.54 | 59.96 | 88.98 | 57.50 | 88.55 | 38.04 | 93.55 | 58.75 | 89.18 | 55.59 | 88.62 |
| | | $median(x)$ | 92.21 | 76.58 | 86.71 | 72.47 | 88.20 | 68.61 | 96.58 | 56.13 | 74.53 | 84.11 | 85.97 | 71.06 | 87.37 | 71.50 |
| | | $entropy(x)$ | 99.68 | 22.01 | 97.29 | 50.70 | 99.46 | 49.21 | 98.37 | 38.77 | 99.57 | 32.48 | 99.34 | 50.35 | 98.95 | 40.59 |
| | ResNet-34 | $\mu(x)$ | 57.79 | 89.80 | 81.17 | 77.25 | 71.83 | 84.14 | 86.77 | 75.82 | 55.56 | 89.92 | 68.70 | 84.93 | 70.30 | 83.64 |
| | | $\mu(x) + 3.0\sigma(x)$ | 31.07 | 95.00 | 80.04 | 77.53 | 59.46 | 89.50 | 59.75 | 88.31 | 41.21 | 93.00 | 60.05 | 89.32 | 55.26 | 88.78 |
| | | $m(x)$ | 29.56 | 95.19 | 77.68 | 78.81 | 55.37 | 90.51 | 56.29 | 89.14 | 38.37 | 93.37 | 56.27 | 90.23 | 52.26 | 89.54 |
| | | $median(x)$ | 99.80 | 24.42 | 96.12 | 55.41 | 99.24 | 48.47 | 99.86 | 27.82 | 97.35 | 58.12 | 98.91 | 51.98 | 98.55 | 44.37 |
| | | $entropy(x)$ | 80.63 | 89.37 | 97.05 | 62.50 | 99.73 | 61.00 | 96.52 | 79.21 | 98.92 | 78.69 | 99.66 | 60.31 | 95.42 | 71.85 |
| | DenseNet-101 | $\mu(x)$ | 70.99 | 86.66 | 77.12 | 76.94 | 64.28 | 83.92 | 83.60 | 67.47 | 11.45 | 97.89 | 56.08 | 86.84 | 60.59 | 83.29 |
| | | $\mu(x) + 3.0\sigma(x)$ | 45.23 | 91.91 | 72.11 | 79.78 | 42.21 | 92.16 | 55.85 | 85.47 | 13.92 | 97.53 | 40.52 | 92.45 | 44.97 | 89.88 |
| | | $m(x)$ | 52.32 | 89.41 | 73.49 | 79.62 | 35.76 | 93.37 | 49.88 | 88.07 | 18.58 | 96.44 | 37.27 | 92.98 | 44.55 | 89.98 |
| | | $median(x)$ | 99.42 | 41.70 | 94.57 | 57.14 | 98.23 | 48.13 | 99.66 | 28.11 | 63.70 | 85.71 | 96.48 | 55.05 | 92.01 | 52.64 |
| | | $entropy(x)$ | 98.87 | 48.54 | 97.38 | 47.45 | 97.54 | 56.47 | 96.56 | 56.33 | 99.15 | 48.05 | 96.62 | 62.24 | 97.69 | 53.18 |
| | MobileNet-v2 | $\mu(x)$ | 69.65 | 85.98 | 81.26 | 75.21 | 78.17 | 83.33 | 80.02 | 78.63 | 50.19 | 89.42 | 76.59 | 84.06 | 72.65 | 82.77 |
| | | $\mu(x) + 3.0\sigma(x)$ | 40.00 | 92.73 | 81.64 | 73.23 | 64.99 | 86.36 | 42.70 | 91.25 | 34.67 | 93.75 | 66.30 | 86.04 | 55.05 | 87.23 |
| | | $m(x)$ | 41.94 | 92.00 | 81.49 | 73.31 | 64.52 | 86.40 | 44.22 | 90.68 | 34.22 | 93.72 | 65.30 | 86.10 | 55.28 | 87.04 |
| | | $median(x)$ | 96.12 | 73.46 | 84.46 | 73.88 | 91.78 | 75.12 | 94.52 | 62.93 | 71.15 | 81.93 | 90.61 | 76.70 | 88.11 | 74.00 |
| | | $entropy(x)$ | 99.81 | 30.47 | 99.13 | 33.71 | 99.94 | 33.94 | 99.73 | 36.10 | 99.86 | 40.63 | 99.95 | 32.66 | 99.74 | 34.59 |

# I ANALYSIS OF THE HYPERPARAMETER $\gamma$

In this section, we conduct two ablation studies. We begin by evaluating the OOD detection performance using different statistical feature representations, including the maximum $m(\mathbf{x})$, the mean $\mu(\mathbf{x})$, and a combination of $\mu(\mathbf{x})$ and the standard deviation $\sigma(\mathbf{x})$. This is followed by an analysis using the median value within each activation map, as well as the entropy computed over individual activation maps.

We evaluate OOD detection performance using different formulations of `DAVIS` on the CIFAR and ImageNet benchmark, as formulated in Equation 23. In particular for CIFAR benchmark, we vary the parameter $\gamma$ over the set $\{1.0, 2.0, 3.0, 4.0\}$ and compare the results using the energy score, as summarized in Table 24. Similarly for ImageNet benchmark, we vary the parameter $\gamma$ over the set $\{0.5, 1.0, 1.5, 2.0\}$ and compare the results using the energy score, as summarized in Table 23. Notably, we do not incorporate any existing techniques in this experiment; instead, we directly report the FPR95 and AUROC scores using the standard energy-based score.

$$h(\mathbf{x}) = \mu(\mathbf{x}) \tag{23a}$$
$$h(\mathbf{x}) = \mu(\mathbf{x}) + \gamma\sigma(\mathbf{x}) \tag{23b}$$
$$h(\mathbf{x}) = m(\mathbf{x}) \tag{23c}$$

From Table 23 and Table 24, we observe that as $\gamma$ increases, the improvement in OOD detection performance gradually diminishes. When $\gamma$ becomes very large, the performance saturates, leading to marginal gains or stagnation. Additionally, as $\gamma$ increases, the behavior of the score function tends to converge to that of using $m(\mathbf{x})$ alone.

Table 23: *Ablation study of applying* `DAVIS` *under different formulations on ImageNet-1K, using DenseNet-121, ResNet-50, MobileNet-v2 and EfficientNet-b0. The results are based solely on energy scores, with* `DAVIS` *not combined with any other post-hoc methods.* ↓ *indicates lower values are better and* ↑ *indicates larger values are better.*

| Model | DAVIS | SUN | | Place | | Texture | | iNaturalist | | Average | |
|---|---|---|---|---|---|---|---|---|---|---|---|
| | | FPR95 ↓ | AUROC ↑ | FPR95 ↓ | AUROC ↑ | FPR95 ↓ | AUROC ↑ | FPR95 ↓ | AUROC ↑ | FPR95 ↓ | AUROC ↑ |
| DenseNet-121 | $\mu(\mathbf{x})$ | 52.51 | 87.27 | 58.24 | 85.05 | 52.22 | 85.42 | 39.75 | 92.66 | 50.68 | 87.60 |
| | $\mu(\mathbf{x}) + 0.5\sigma(\mathbf{x})$ | 48.33 | 88.64 | 56.95 | 85.80 | 37.50 | 90.73 | 31.69 | 94.14 | 43.62 | 89.83 |
| | $\mu(\mathbf{x}) + 1.0\sigma(\mathbf{x})$ | 48.35 | 88.81 | 57.28 | 85.58 | 31.05 | 92.67 | 30.51 | 94.17 | 41.80 | 90.30 |
| | $\mu(\mathbf{x}) + 1.5\sigma(\mathbf{x})$ | 48.31 | 88.75 | 57.99 | 85.24 | 27.38 | 93.61 | 30.40 | 93.99 | 41.02 | 90.40 |
| | $\mu(\mathbf{x}) + 2.0\sigma(\mathbf{x})$ | 48.40 | 88.64 | 58.38 | 84.94 | 25.37 | 94.14 | 30.31 | 93.79 | 40.62 | 90.38 |
| | $m(\mathbf{x})$ | 48.55 | 88.03 | 59.36 | 83.53 | 27.54 | 93.62 | 33.74 | 92.20 | 42.30 | 89.34 |
| ResNet-50 | $\mu(\mathbf{x})$ | 58.82 | 86.58 | 65.99 | 83.96 | 52.43 | 86.72 | 53.74 | 90.62 | 57.74 | 86.97 |
| | $\mu(\mathbf{x}) + 0.5\sigma(\mathbf{x})$ | 54.55 | 87.36 | 63.27 | 84.26 | 34.40 | 91.81 | 39.60 | 92.80 | 47.95 | 89.06 |
| | $\mu(\mathbf{x}) + 1.0\sigma(\mathbf{x})$ | 53.82 | 87.18 | 63.12 | 83.72 | 27.36 | 93.80 | 35.92 | 93.14 | 45.05 | 89.46 |
| | $\mu(\mathbf{x}) + 1.5\sigma(\mathbf{x})$ | 54.41 | 86.83 | 63.69 | 83.10 | 23.39 | 94.77 | 35.01 | 93.10 | 44.12 | 89.45 |
| | $\mu(\mathbf{x}) + 2.0\sigma(\mathbf{x})$ | 54.71 | 86.47 | 64.06 | 82.55 | 21.13 | 95.32 | 34.50 | 92.95 | 43.60 | 89.33 |
| | $m(\mathbf{x})$ | 53.12 | 85.49 | 63.58 | 80.54 | 20.94 | 95.09 | 36.11 | 91.29 | 43.44 | 88.10 |
| MobileNet-v2 | $\mu(\mathbf{x})$ | 59.60 | 86.16 | 66.36 | 83.15 | 54.82 | 86.57 | 55.33 | 90.37 | 59.03 | 86.56 |
| | $\mu(\mathbf{x}) + 0.5\sigma(\mathbf{x})$ | 56.37 | 87.07 | 64.55 | 83.69 | 38.76 | 90.99 | 44.87 | 92.34 | 51.14 | 88.53 |
| | $\mu(\mathbf{x}) + 1.0\sigma(\mathbf{x})$ | 55.44 | 87.23 | 64.27 | 83.59 | 31.40 | 92.82 | 41.02 | 92.84 | 48.03 | 89.12 |
| | $\mu(\mathbf{x}) + 1.5\sigma(\mathbf{x})$ | 55.23 | 87.22 | 64.45 | 83.39 | 27.70 | 93.77 | 39.68 | 92.97 | 46.76 | 89.33 |
| | $\mu(\mathbf{x}) + 2.0\sigma(\mathbf{x})$ | 55.47 | 87.15 | 64.82 | 83.18 | 25.78 | 94.33 | 38.94 | 92.98 | 46.25 | 89.41 |
| | $m(\mathbf{x})$ | 53.64 | 87.26 | 63.61 | 82.93 | 25.87 | 94.03 | 40.25 | 92.22 | 45.84 | 89.11 |
| EfficientNet-b0 | $\mu(\mathbf{x})$ | 85.01 | 72.86 | 86.06 | 70.99 | 75.99 | 75.86 | 78.91 | 79.78 | 81.49 | 74.87 |
| | $\mu(\mathbf{x}) + 0.5\sigma(\mathbf{x})$ | 62.63 | 83.98 | 70.06 | 79.49 | 24.01 | 95.43 | 47.51 | 89.36 | 51.05 | 87.06 |
| | $\mu(\mathbf{x}) + 1.0\sigma(\mathbf{x})$ | 61.07 | 84.67 | 70.09 | 79.51 | 16.01 | 96.97 | 46.69 | 89.09 | 48.47 | 87.56 |
| | $\mu(\mathbf{x}) + 1.5\sigma(\mathbf{x})$ | 61.56 | 84.72 | 70.87 | 79.24 | 13.55 | 97.37 | 47.65 | 88.67 | 48.41 | 87.50 |
| | $\mu(\mathbf{x}) + 2.0\sigma(\mathbf{x})$ | 61.85 | 84.67 | 71.31 | 79.01 | 12.18 | 97.52 | 48.25 | 88.35 | 48.40 | 87.39 |
| | $m(\mathbf{x})$ | 69.42 | 80.74 | 79.09 | 73.67 | 15.82 | 96.38 | 63.68 | 80.63 | 57.00 | 82.85 |

Table 24: Ablation study of applying *DAVIS* under different formulations on *CIFAR-10* and *CIFAR-100*, using *ResNet-18, ResNet-34, DenseNet-101* and *MobileNet-v2* architectures. *The results are based solely on energy scores, with DAVIS not combined with any other post-hoc methods.* ↓ *indicates lower values are better and* ↑ *indicates larger values are better.*

| Dataset | DAVIS | Model | SVHN FPR95↓ | SVHN AUROC↑ | Place365 FPR95↓ | Place365 AUROC↑ | iSUN FPR95↓ | iSUN AUROC↑ | Textures FPR95↓ | Textures AUROC↑ | LSUN-c FPR95↓ | LSUN-c AUROC↑ | LSUN-r FPR95↓ | LSUN-r AUROC↑ | Average FPR95↓ | Average AUROC↑ |
|---|---|---|---|---|---|---|---|---|---|---|---|---|---|---|---|---|
| CIFAR-10 | ResNet-18 | $\mu(x)$ | 44.32 | 94.04 | 41.31 | 91.73 | 35.46 | 94.64 | 50.39 | 91.12 | 9.77 | 98.19 | 32.41 | 95.16 | 35.61 | 94.14 |
| | | $\mu(x) + 1.0\sigma(x)$ | 23.17 | 96.07 | 37.39 | 92.89 | 24.13 | 96.45 | 33.60 | 94.96 | 6.35 | 98.69 | 21.25 | 96.81 | 24.32 | 95.98 |
| | | $\mu(x) + 2.0\sigma(x)$ | 20.51 | 96.36 | 36.60 | 93.08 | 21.74 | 96.77 | 28.74 | 95.62 | 5.80 | 98.77 | 18.78 | 97.09 | 22.03 | 96.28 |
| | | $\mu(x) + 3.0\sigma(x)$ | 19.83 | 96.47 | 36.11 | 93.16 | 20.48 | 96.90 | 26.77 | 95.89 | 5.62 | 98.79 | 17.75 | 97.21 | 21.09 | 96.40 |
| | | $\mu(x) + 4.0\sigma(x)$ | 19.23 | 96.52 | 35.59 | 93.20 | 19.82 | 96.97 | 25.60 | 96.03 | 5.53 | 98.81 | 17.08 | 97.27 | 20.48 | 96.47 |
| | | $m(x)$ | 19.81 | 96.29 | 32.32 | 93.73 | 15.79 | 97.37 | 21.90 | 96.44 | 5.83 | 98.73 | 13.63 | 97.61 | 18.21 | 96.69 |
| | ResNet-34 | $\mu(x)$ | 35.44 | 93.76 | 38.15 | 92.27 | 19.90 | 96.87 | 42.52 | 92.54 | 3.38 | 99.10 | 16.86 | 97.20 | 26.04 | 95.29 |
| | | $\mu(x) + 1.0\sigma(x)$ | 29.12 | 94.81 | 33.66 | 93.45 | 12.81 | 97.70 | 27.71 | 95.66 | 3.28 | 99.22 | 11.35 | 97.89 | 19.65 | 96.45 |
| | | $\mu(x) + 2.0\sigma(x)$ | 28.70 | 94.88 | 32.81 | 93.62 | 11.50 | 97.83 | 24.41 | 96.18 | 3.39 | 99.22 | 10.55 | 97.99 | 18.56 | 96.62 |
| | | $\mu(x) + 3.0\sigma(x)$ | 28.96 | 94.90 | 32.66 | 93.69 | 11.31 | 97.88 | 23.07 | 96.39 | 3.47 | 99.22 | 10.35 | 98.04 | 18.30 | 96.69 |
| | | $\mu(x) + 4.0\sigma(x)$ | 28.88 | 94.91 | 32.29 | 93.72 | 10.96 | 97.91 | 22.34 | 96.50 | 3.47 | 99.21 | 10.15 | 98.06 | 18.01 | 96.72 |
| | | $m(x)$ | 29.29 | 94.59 | 29.59 | 94.21 | 10.17 | 98.03 | 20.23 | 96.73 | 4.00 | 99.10 | 9.45 | 98.16 | 17.12 | 96.81 |
| | DenseNet-101 | $\mu(x)$ | 37.91 | 93.59 | 36.38 | 92.39 | 7.83 | 98.23 | 43.85 | 90.49 | 1.95 | 99.47 | 7.34 | 98.34 | 22.54 | 95.42 |
| | | $\mu(x) + 1.0\sigma(x)$ | 27.94 | 95.62 | 32.66 | 93.32 | 5.52 | 98.63 | 29.96 | 94.64 | 3.68 | 99.17 | 5.43 | 98.70 | 17.53 | 96.68 |
| | | $\mu(x) + 2.0\sigma(x)$ | 25.73 | 95.87 | 32.50 | 93.42 | 5.65 | 98.65 | 26.79 | 95.49 | 4.66 | 99.02 | 5.44 | 98.71 | 16.79 | 96.86 |
| | | $\mu(x) + 3.0\sigma(x)$ | 24.75 | 95.93 | 32.08 | 93.43 | 5.77 | 98.64 | 25.23 | 95.84 | 5.13 | 98.94 | 5.63 | 98.70 | 16.43 | 96.91 |
| | | $\mu(x) + 4.0\sigma(x)$ | 24.20 | 95.96 | 31.95 | 93.44 | 5.80 | 98.63 | 24.04 | 96.03 | 5.47 | 98.88 | 5.64 | 98.69 | 16.18 | 96.94 |
| | | $m(x)$ | 30.50 | 94.50 | 33.04 | 93.28 | 7.81 | 98.29 | 25.16 | 95.85 | 9.95 | 98.19 | 7.32 | 98.37 | 18.96 | 96.41 |
| | MobileNet-v2 | $\mu(x)$ | 75.83 | 85.85 | 44.98 | 90.62 | 29.68 | 95.03 | 48.67 | 91.19 | 9.54 | 98.12 | 29.80 | 95.10 | 39.75 | 92.65 |
| | | $\mu(x) + 1.0\sigma(x)$ | 66.48 | 88.29 | 43.56 | 90.97 | 24.53 | 96.08 | 38.67 | 93.72 | 11.88 | 97.81 | 23.22 | 96.22 | 34.72 | 93.85 |
| | | $\mu(x) + 2.0\sigma(x)$ | 63.81 | 88.91 | 43.94 | 90.92 | 23.16 | 96.31 | 35.46 | 94.40 | 13.38 | 97.62 | 21.98 | 96.47 | 33.62 | 94.11 |
| | | $\mu(x) + 3.0\sigma(x)$ | 62.07 | 89.18 | 43.87 | 90.86 | 22.38 | 96.40 | 33.24 | 94.71 | 14.21 | 97.50 | 21.32 | 96.57 | 32.85 | 94.20 |
| | | $\mu(x) + 4.0\sigma(x)$ | 60.98 | 89.32 | 43.59 | 90.81 | 21.82 | 96.44 | 31.79 | 94.88 | 14.55 | 97.42 | 20.56 | 96.62 | 32.21 | 94.25 |
| | | $m(x)$ | 69.61 | 87.56 | 44.33 | 91.00 | 23.33 | 96.26 | 37.87 | 94.11 | 17.62 | 97.02 | 22.48 | 96.40 | 35.87 | 93.73 |
| CIFAR-100 | ResNet-18 | $\mu(x)$ | 66.64 | 89.53 | 81.23 | 76.84 | 73.67 | 82.01 | 85.30 | 75.68 | 48.01 | 91.63 | 70.30 | 83.38 | 70.86 | 83.18 |
| | | $\mu(x) + 1.0\sigma(x)$ | 50.36 | 92.68 | 79.76 | 77.60 | 66.90 | 86.55 | 70.83 | 84.19 | 41.42 | 93.01 | 64.24 | 87.16 | 62.25 | 86.86 |
| | | $\mu(x) + 2.0\sigma(x)$ | 44.41 | 93.44 | 79.20 | 77.62 | 62.91 | 87.94 | 63.87 | 86.71 | 39.10 | 93.34 | 61.13 | 88.32 | 58.44 | 87.89 |
| | | $\mu(x) + 3.0\sigma(x)$ | 42.01 | 93.75 | 78.99 | 77.58 | 61.13 | 88.60 | 59.89 | 87.89 | 38.56 | 93.48 | 59.76 | 88.86 | 56.72 | 88.36 |
| | | $\mu(x) + 4.0\sigma(x)$ | 40.43 | 93.92 | 78.88 | 77.54 | 59.96 | 88.98 | 57.50 | 88.55 | 38.04 | 93.55 | 58.75 | 89.18 | 55.59 | 88.62 |
| | | $m(x)$ | 38.40 | 94.25 | 77.62 | 78.98 | 56.43 | 89.99 | 55.73 | 89.14 | 35.90 | 94.17 | 55.87 | 89.95 | 53.32 | 89.41 |
| | ResNet-34 | $\mu(x)$ | 57.79 | 89.80 | 81.17 | 77.25 | 71.83 | 84.14 | 86.77 | 75.82 | 55.56 | 89.92 | 68.70 | 84.93 | 70.30 | 83.64 |
| | | $\mu(x) + 1.0\sigma(x)$ | 38.34 | 93.68 | 80.15 | 77.72 | 63.63 | 87.91 | 71.26 | 84.56 | 45.92 | 92.16 | 62.71 | 88.03 | 60.33 | 87.34 |
| | | $\mu(x) + 2.0\sigma(x)$ | 33.40 | 94.61 | 79.93 | 77.63 | 61.00 | 88.99 | 64.01 | 87.13 | 42.73 | 92.74 | 60.97 | 88.91 | 57.01 | 88.34 |
| | | $\mu(x) + 3.0\sigma(x)$ | 31.07 | 95.00 | 80.04 | 77.53 | 59.46 | 89.50 | 59.75 | 88.31 | 41.21 | 93.00 | 60.05 | 89.32 | 55.26 | 88.78 |
| | | $\mu(x) + 4.0\sigma(x)$ | 30.17 | 95.21 | 80.44 | 77.45 | 58.88 | 89.78 | 57.50 | 88.99 | 40.66 | 93.14 | 59.67 | 89.56 | 54.55 | 89.02 |
| | | $m(x)$ | 29.56 | 95.19 | 77.68 | 78.81 | 55.37 | 90.51 | 56.29 | 89.14 | 38.37 | 93.37 | 56.27 | 90.23 | 52.26 | 89.54 |
| | DenseNet-101 | $\mu(x)$ | 70.99 | 86.66 | 77.12 | 76.94 | 64.28 | 83.92 | 83.60 | 67.47 | 11.45 | 97.89 | 56.08 | 86.84 | 60.59 | 83.29 |
| | | $\mu(x) + 1.0\sigma(x)$ | 48.54 | 91.64 | 72.99 | 79.57 | 48.89 | 90.36 | 65.82 | 80.52 | 12.65 | 97.80 | 44.65 | 91.29 | 48.92 | 88.53 |
| | | $\mu(x) + 2.0\sigma(x)$ | 45.38 | 91.93 | 71.75 | 79.77 | 43.78 | 91.65 | 58.81 | 83.94 | 13.18 | 97.63 | 41.41 | 92.13 | 45.72 | 89.51 |
| | | $\mu(x) + 3.0\sigma(x)$ | 45.23 | 91.91 | 72.11 | 79.78 | 42.21 | 92.16 | 55.85 | 85.47 | 13.92 | 97.53 | 40.52 | 92.45 | 44.97 | 89.88 |
| | | $\mu(x) + 4.0\sigma(x)$ | 45.67 | 91.86 | 72.40 | 79.76 | 41.30 | 92.43 | 54.45 | 86.33 | 14.39 | 97.46 | 40.31 | 92.61 | 44.75 | 90.07 |
| | | $m(x)$ | 52.32 | 89.41 | 73.49 | 79.62 | 35.76 | 93.37 | 49.88 | 88.07 | 18.58 | 96.44 | 37.27 | 92.98 | 44.55 | 89.98 |
| | MobileNet-v2 | $\mu(x)$ | 69.65 | 85.98 | 81.26 | 75.21 | 78.17 | 83.33 | 80.02 | 78.63 | 50.19 | 89.42 | 76.59 | 84.06 | 72.65 | 82.77 |
| | | $\mu(x) + 1.0\sigma(x)$ | 47.36 | 91.05 | 80.49 | 74.45 | 68.34 | 85.74 | 56.91 | 87.66 | 38.63 | 92.57 | 68.52 | 85.76 | 60.04 | 86.20 |
| | | $\mu(x) + 2.0\sigma(x)$ | 42.01 | 92.25 | 81.14 | 73.72 | 66.13 | 86.22 | 47.16 | 90.17 | 35.84 | 93.39 | 67.00 | 86.00 | 56.55 | 86.96 |
| | | $\mu(x) + 3.0\sigma(x)$ | 40.00 | 92.73 | 81.64 | 73.23 | 64.99 | 86.36 | 42.70 | 91.25 | 34.67 | 93.75 | 66.30 | 86.04 | 55.05 | 87.23 |
| | | $\mu(x) + 4.0\sigma(x)$ | 38.28 | 92.99 | 81.77 | 72.90 | 64.03 | 86.42 | 40.00 | 91.85 | 33.51 | 93.93 | 65.60 | 86.02 | 53.87 | 87.35 |
| | | $m(x)$ | 41.94 | 92.00 | 81.49 | 73.31 | 64.52 | 86.40 | 44.22 | 90.68 | 34.22 | 93.72 | 65.30 | 86.10 | 55.28 | 87.04 |

## J  SiLU Activation on OOD Detection

This section details our investigation into the performance degradation of standard OOD baselines on SiLU-based architectures like EfficientNet. We attribute this failure to the fundamental difference between the sparse feature maps produced by the ReLU activation function versus the dense maps produced by SiLU.

The ReLU function, $\text{ReLU}(\mathbf{x}) = \max(0, \mathbf{x})$, creates sparse activations by forcing all negative inputs to zero. This sparsity is an implicit assumption for pruning-based methods like ASH. In contrast, the SiLU function, $\text{SiLU}(\mathbf{x}) = \mathbf{x} \cdot \sigma(\mathbf{x})$ where $\sigma(\mathbf{x})$ is the sigmoid, is non-sparsifying and produces dense feature maps, altering the statistical landscape on which these methods were designed to operate.

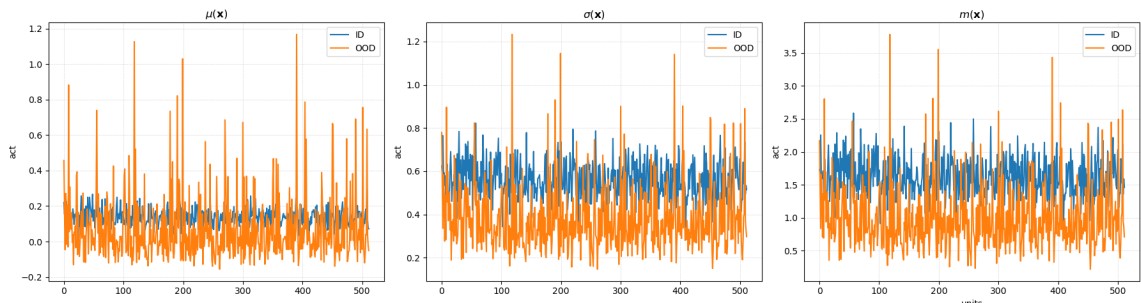

Figure 8: *Feature statistics for ID (CIFAR-100) vs. OOD (Texture) samples on an ResNet18-SiLU backbone. While mean $\mu(\mathbf{x})$ show poor separation, both the standard deviation $\sigma(\mathbf{x})$ and maximum $m(\mathbf{x})$ statistics maintain a clear separation between ID and OOD activations.*

To isolate this effect, we evaluated ResNet-18 with SiLU instead of ReLU on the CIFAR benchmarks. (termed ResNet18-SiLU) on CIFAR benchmarks. This experiment revealed two key findings as shown in Figure 8

- High Overlap of Mean Features: The standard mean features extracted using GAP from the ResNet18-SiLU model show a high degree of overlap between ID and OOD samples. This poor separability is the primary reason for the failure of baselines that rely solely on these features.

- Enhanced Separation with `DAVIS`: In contrast, the feature representations from our method, `DAVIS`$(m)$ and `DAVIS`$(\mu, \sigma)$, successfully establish a clear and discriminative boundary between the ID and OOD distributions. This demonstrates that the maximum and standard deviation statistics are robust cues even in a dense activation landscape.

These qualitative findings are confirmed by the quantitative results in Table 25. There, we show that standard baselines on ResNet18-SiLU perform poorly compared to standard ResNet18 in Table 3, but are critically rescued when combined with `DAVIS`. This analysis strongly supports our hypothesis that the dense nature of SiLU challenges conventional OOD methods, and it highlights the robustness and generality of the statistical cues leveraged by our approach.

| Dataset | Combined Method | SVHN FPR95 ↓ | SVHN AUROC ↑ | Place365 FPR95 ↓ | Place365 AUROC ↑ | iSUN FPR95 ↓ | iSUN AUROC ↑ | Textures FPR95 ↓ | Textures AUROC ↑ | LSUN-c FPR95 ↓ | LSUN-c AUROC ↑ | LSUN-r FPR95 ↓ | LSUN-r AUROC ↑ | Average FPR95 ↓ | Average AUROC ↑ |
|---|---|---|---|---|---|---|---|---|---|---|---|---|---|---|---|
| CIFAR-10 | MSP | 56.15 | 92.93 | 62.45 | 88.72 | 41.70 | 94.49 | 58.03 | 91.02 | 26.15 | 96.60 | 39.12 | 94.79 | 47.27 | 93.10 |
| | +DAVIS(m) | 54.29 | 85.57 | 65.92 | 79.62 | 49.12 | 89.53 | 52.64 | 86.94 | 31.24 | 93.75 | 47.55 | 89.76 | 50.13 | 87.53 |
| | +DAVIS(μ,σ) | 53.47 | 85.33 | 66.64 | 77.71 | 48.62 | 88.38 | 52.71 | 85.59 | 29.52 | 93.75 | 47.41 | 88.87 | 49.73 | 86.61 |
| | ODIN | 26.52 | 95.89 | 39.60 | 91.87 | 6.64 | 98.59 | 35.55 | 93.89 | 1.41 | 99.53 | 6.33 | 98.65 | 19.34 | 96.40 |
| | +DAVIS(m) | 25.47 | 95.83 | 40.46 | 91.74 | 7.76 | 98.25 | 35.60 | 93.83 | 1.80 | 99.31 | 7.01 | 98.34 | 19.68 | 96.22 |
| | +DAVIS(μ,σ) | 25.56 | 95.91 | 41.01 | 91.72 | 7.50 | 98.35 | 35.85 | 93.84 | 1.74 | 99.39 | 6.82 | 98.43 | 19.75 | 96.27 |
| | Energy | 32.13 | 95.20 | 40.20 | 91.53 | 10.39 | 97.91 | 39.57 | 92.97 | 1.93 | 99.34 | 9.45 | 98.03 | 22.28 | 95.83 |
| | +DAVIS(m) | 23.88 | 96.03 | 35.55 | 93.30 | 7.20 | 98.50 | 14.79 | 97.50 | 3.15 | 99.27 | 7.88 | 98.43 | 15.41 | 97.17 |
| | +DAVIS(μ,σ) | 23.90 | 96.14 | 38.24 | 92.82 | 8.25 | 98.39 | 16.76 | 97.23 | 2.49 | 99.39 | 8.57 | 98.36 | 16.37 | 97.05 |
| | ReAct | 31.97 | 95.25 | 39.12 | 91.96 | 10.03 | 97.95 | 37.18 | 93.68 | 1.71 | 99.36 | 9.08 | 98.08 | 21.51 | 96.05 |
| | +DAVIS(m) | 24.58 | 95.93 | 36.02 | 93.13 | 7.37 | 98.48 | 15.32 | 97.41 | 3.38 | 99.25 | 8.18 | 98.41 | 15.81 | 97.10 |
| | +DAVIS(μ,σ) | 23.66 | 96.09 | 38.01 | 92.72 | 8.09 | 98.38 | 17.00 | 97.16 | 2.59 | 99.38 | 8.51 | 98.36 | 16.31 | 97.01 |
| | DICE | 23.87 | 96.29 | 43.98 | 90.89 | 9.19 | 98.22 | 29.06 | 94.70 | 0.20 | 99.92 | 10.04 | 98.12 | 19.39 | 96.36 |
| | +DAVIS(m) | 14.99 | 97.42 | 30.73 | 93.87 | 4.48 | 99.05 | 8.03 | 98.53 | 0.54 | 99.84 | 5.40 | 98.91 | 10.70 | 97.94 |
| | +DAVIS(μ,σ) | 15.10 | 97.46 | 33.85 | 93.40 | 5.30 | 98.94 | 9.86 | 98.35 | 0.35 | 99.89 | 5.80 | 98.82 | 11.71 | 97.81 |
| | ReAct+DICE | 22.79 | 96.43 | 42.96 | 91.39 | 8.87 | 98.27 | 26.47 | 95.57 | 0.15 | 99.93 | 9.50 | 98.18 | 18.46 | 96.63 |
| | +DAVIS(m) | 15.00 | 97.43 | 30.92 | 93.79 | 4.45 | 99.06 | 8.23 | 98.52 | 0.49 | 99.84 | 5.16 | 98.93 | 10.71 | 97.93 |
| | +DAVIS(μ,σ) | 14.72 | 97.49 | 33.72 | 93.37 | 5.23 | 98.95 | 9.88 | 98.33 | 0.34 | 99.89 | 5.64 | 98.85 | 11.59 | 97.81 |
| | ASH | 24.10 | 96.08 | 46.30 | 90.63 | 12.24 | 97.76 | 29.88 | 94.88 | 0.13 | 99.94 | 13.39 | 97.66 | 21.01 | 96.16 |
| | +DAVIS(m) | 19.73 | 96.50 | 33.94 | 93.39 | 7.64 | 98.45 | 11.10 | 98.00 | 1.26 | 99.63 | 8.73 | 98.32 | 13.73 | 97.38 |
| | +DAVIS(μ,σ) | 19.15 | 96.68 | 36.41 | 92.97 | 8.92 | 98.32 | 12.22 | 97.88 | 0.83 | 99.71 | 9.42 | 98.22 | 14.49 | 97.30 |
| | Scale | 21.51 | 96.45 | 44.14 | 91.24 | 13.64 | 97.59 | 32.93 | 94.35 | 0.08 | 99.94 | 13.88 | 97.56 | 21.03 | 96.19 |
| | +DAVIS(m) | 19.04 | 96.60 | 33.45 | 93.45 | 7.20 | 98.49 | 11.72 | 97.90 | 1.01 | 99.66 | 8.04 | 98.38 | 13.41 | 97.41 |
| | +DAVIS(μ,σ) | 19.04 | 96.66 | 36.78 | 92.99 | 8.80 | 98.31 | 13.48 | 97.74 | 0.70 | 99.72 | 9.28 | 98.21 | 14.68 | 97.27 |
| CIFAR-100 | MSP | 77.86 | 79.19 | 83.80 | 74.05 | 91.19 | 63.67 | 85.76 | 73.19 | 59.92 | 87.22 | 89.41 | 65.42 | 81.32 | 73.79 |
| | +DAVIS(m) | 80.46 | 82.22 | 88.45 | 70.49 | 90.27 | 66.73 | 83.95 | 76.98 | 76.11 | 83.41 | 90.84 | 67.03 | 85.01 | 74.48 |
| | +DAVIS(μ,σ) | 79.12 | 82.26 | 87.15 | 69.95 | 91.24 | 63.77 | 83.92 | 75.93 | 74.55 | 83.71 | 90.77 | 64.41 | 84.46 | 73.34 |
| | ODIN | 68.44 | 88.58 | 81.12 | 75.81 | 80.93 | 76.74 | 83.46 | 76.82 | 17.55 | 97.00 | 75.89 | 80.15 | 67.90 | 82.52 |
| | +DAVIS(m) | 63.88 | 89.67 | 80.31 | 76.13 | 83.00 | 76.10 | 81.31 | 77.75 | 20.49 | 96.51 | 78.92 | 79.34 | 67.99 | 82.58 |
| | +DAVIS(μ,σ) | 63.93 | 89.69 | 80.60 | 76.02 | 82.22 | 76.39 | 81.65 | 77.62 | 19.72 | 96.70 | 78.11 | 79.63 | 67.70 | 82.68 |
| | Energy | 70.16 | 88.22 | 81.62 | 75.39 | 87.89 | 71.90 | 83.84 | 75.59 | 18.58 | 96.74 | 84.22 | 75.60 | 71.07 | 80.57 |
| | +DAVIS(m) | 25.26 | 95.92 | 77.90 | 78.86 | 69.87 | 86.94 | 38.67 | 92.50 | 11.95 | 97.97 | 69.65 | 87.19 | 48.88 | 89.90 |
| | +DAVIS(μ,σ) | 25.38 | 95.97 | 78.64 | 77.80 | 76.25 | 83.98 | 45.50 | 91.10 | 10.54 | 98.19 | 75.38 | 84.73 | 51.95 | 88.63 |
| | ReAct | 54.12 | 90.84 | 80.64 | 76.61 | 66.98 | 88.00 | 66.72 | 86.45 | 27.90 | 94.96 | 63.06 | 88.84 | 59.90 | 87.62 |
| | +DAVIS(m) | 23.81 | 96.03 | 77.76 | 78.79 | 68.00 | 87.57 | 38.30 | 92.61 | 12.51 | 97.86 | 67.60 | 87.52 | 48.00 | 90.06 |
| | +DAVIS(μ,σ) | 22.80 | 96.29 | 79.00 | 77.97 | 72.74 | 86.03 | 43.69 | 91.76 | 11.38 | 98.08 | 72.16 | 86.20 | 50.29 | 89.39 |
| | DICE | 19.78 | 96.53 | 84.56 | 72.27 | 84.66 | 71.12 | 59.17 | 82.42 | 1.98 | 99.58 | 82.31 | 74.30 | 55.41 | 82.70 |
| | +DAVIS(m) | 12.52 | 97.44 | 82.47 | 76.44 | 50.05 | 91.72 | 23.28 | 94.97 | 3.46 | 99.16 | 52.83 | 91.29 | 37.44 | 91.84 |
| | +DAVIS(μ,σ) | 10.68 | 97.87 | 81.90 | 75.76 | 59.23 | 89.12 | 25.99 | 94.39 | 2.21 | 99.45 | 59.88 | 89.16 | 39.98 | 90.96 |
| | ReAct+DICE | 10.77 | 97.83 | 89.79 | 66.94 | 65.33 | 87.49 | 30.30 | 93.20 | 2.91 | 99.36 | 70.11 | 86.15 | 44.87 | 88.50 |
| | +DAVIS(m) | 12.98 | 97.43 | 83.72 | 75.80 | 49.29 | 92.14 | 23.88 | 94.99 | 3.67 | 99.13 | 53.48 | 91.52 | 37.84 | 91.83 |
| | +DAVIS(μ,σ) | 10.84 | 97.85 | 83.03 | 75.13 | 54.04 | 90.80 | 25.02 | 94.71 | 2.43 | 99.40 | 56.13 | 90.34 | 38.58 | 91.37 |
| | ASH | 13.60 | 97.39 | 85.86 | 70.56 | 86.41 | 73.43 | 53.72 | 86.43 | 1.96 | 99.56 | 85.89 | 74.29 | 54.57 | 83.61 |
| | +DAVIS(m) | 8.18 | 98.41 | 81.23 | 75.79 | 49.45 | 91.44 | 20.20 | 95.87 | 5.58 | 98.90 | 54.73 | 90.31 | 36.56 | 91.79 |
| | +DAVIS(μ,σ) | 7.14 | 98.64 | 81.11 | 75.19 | 54.30 | 90.30 | 21.05 | 95.80 | 5.02 | 99.02 | 58.78 | 89.22 | 37.90 | 91.36 |
| | Scale | 14.28 | 97.36 | 85.50 | 71.18 | 83.92 | 76.62 | 52.38 | 87.33 | 1.86 | 99.57 | 82.56 | 77.55 | 53.42 | 84.94 |
| | +DAVIS(m) | 9.47 | 98.16 | 83.17 | 75.36 | 56.46 | 90.05 | 22.59 | 95.39 | 6.14 | 98.78 | 63.04 | 88.62 | 40.14 | 91.06 |
| | +DAVIS(μ,σ) | 7.94 | 98.44 | 82.81 | 75.10 | 60.86 | 88.86 | 22.54 | 95.43 | 5.36 | 98.96 | 65.95 | 87.44 | 40.91 | 90.70 |

Table 25: *Detailed results of post-hoc methods combined with DAVIS using **ResNet-18 with SiLU** activation function instead of **ReLU** trained on CIFAR dataset. ↑ indicates higher is better; ↓ indicates lower is better. The symbols denote the statistic used: μ (mean), σ (std. deviation), m (maximum).*

