# OpenReview forum: "DAVIS: Out-of-Distribution Detection via Dominant Activations and Variance for Increased Separation"
_ICLR.cc/2026/Conference — Submitted to ICLR 2026_

### Official Review · Reviewer_3MkL · 2025-10-17

**Soundness:** 2
**Presentation:** 1
**Contribution:** 2
**Rating:** 2
**Confidence:** 5

**Summary:**

This paper hypothesizes that the penultimate feature is information lossy due to the global average pooling (GAP) operation, resulting in an inferior OOD performance.  Therefore,  they propose to modify the feature before the GAP via 1)  utilizing channel-wise variance and 2) dominant (maximum) activations  and Energy score is adopted as the final OOD score. However, the proposed method only works for the architectures that use a spatial aggregation operations as pointed in the section "Limitations".

**Strengths:**

Strengths:

- The hypothesis that feature after GAP operation is information lossy sounds reasonable for OOD detection.
- Comprehensive experiments are done for architectures with global average pooling operations.
- The Discussion section is well-written and comprehensive.

**Weaknesses:**

- I have to say that this paper miss a lot of highly relevant related works including but not limited to Mahalanobis [1], GradNorm[2], ViM [3], GEN [4]  and NN-guide [5]. I list some of important and relevant ones, but please read  Generalized Out-of-Distribution Detection: A Survey  for reference.
- The proposed method is limited to the architectures with global average pooling, which constrains its applicability. Yet the authors have done experiments extensively on different architectures,  there are several strong baseline methods are missing to compare.  I highly recommend the authors follow the standard benchmark openOOD (https://zjysteven.github.io/OpenOOD/) for a fair and comprehensive experiments as other papers do!
-  Another concern is that the performance of the proposed two alternatives $\text{DAVIS}(m)$ and $\text{DAVIS}(\mu, \sigma)$ is unstable and it is really difficult to tell from the table 1 and 2. It seems that the best performance is highlighted and it looks like  the variant of $\text{DAVIS}(m)$ and $\text{DAVIS}(\mu, \sigma)$ equipped with DICE and DICE* achieve the SOTA performance. However, there is no one method is consistently better than or equivalently good as other baselines.  For readability, please report the averaged results across architectures,  $i.e.$, adding two columns at the end of each table.

[1] A simple unified framework for detecting out-of-distribution samples and adversarial attacks, NeurIPS, 2018.
[2] On the importance of gradients for detecting distributional shifts in the wild, NeurIPS, 2021.
[3] Vim: Out-of-distribution with virtual-logit matching, CVPR, 2022.
[4] Gen: Pushing the limits of softmax-based out-of-distribution detection, CVPR, 2023.
[5] Nearest neighbor guidance for out-of-distribution detection, ICCV, 2023.


Minor weaknesses:
- Figures 1 and 2 should be arranged in one row for better readability.

**Questions:**

See weaknesses.

---

> ### Author Response · Authors · 2025-11-26
>
> $\textbf{Weaknesses}$
>
> $\textbf{W1.}$ We clarify that our selection of primary baselines (Energy, ReAct, DICE, ASH, SCALE) was guided by their consistently strong performance under our rigorous evaluation protocol, ensuring comparison against the most competitive and widely adopted post-hoc OOD detectors. Methods such as Mahalanobis, GradNorm, ViM, GEN, and NN-Guide were omitted as primary baselines due to their limited performance (Section 4.2, lines 288–290). For example, Mahalanobis baseline yields an FPR95 of 87% on ImageNet (ResNet-50), compared to 22.83% from ASH baseline, while also being significantly more computationally expensive. We are happy to include a broader comparison to these additional methods, covered in our literature survey -- in the final version, where space permits.
>
> $\textbf{W2.}$ We agree that DAVIS is specifically designed for architectures employing Global Average Pooling (GAP). This is the limitation of existing version of DAVIS and technical focus on solving a fundamental feature bottleneck present in the vast majority of modern CNNs (ResNet, DenseNet, MobileNet, EfficientNet). Our core contribution is precisely based on recovering critical statistical information (variance and dominant activations) that the GAP operation intrinsically discards. We explicitly discuss this architectural scope and the extension to non-GAP models (such as ViTs, which use [CLS] tokens) in Section 5: Limitation and Future Work, where we frame it as a necessary avenue for future research.
>
> We agree that standardized benchmarks such as OpenOOD are valuable for reproducibility. For completeness, we also evaluated on the OpenImage-O dataset and found DAVIS improves on the reported primary baselines. For instance, using pre-trained DenseNet-121, $DAVIS(\mu,\sigma)$ improves FPR95 on OpenImage-O by 29.13% compared to Energy baselines and $DAVIS(\mu,\sigma)+ASH$ improves ASH by 7.68\%. We argue, with OpenImage-O our protocol subsumes and is stricter than OpenOOD v1.5 in two key respects:
>
> $\textbf{1. Use of full, unaltered datasets:}$ We evaluate on the same canonical OOD benchmarks used in Energy, ReAct, ASH, DICE, and SCALE—specifically SUN, Places, iSUN, LSUN, LSUN-resize, iNaturalist, and the full Texture dataset. In contrast, the OpenOOD[2] ImageNet protocol excludes several of these widely-used and challenging testbeds (e.g., SUN, Places, and multiple high-complexity Texture categories such as bubbly, honeycombed, cobwebbed, and spiralled)[1] , and its CIFAR protocol omits iSUN, LSUN, and LSUN-resize. With the addition of evaluation of  ``OpenImage-O’’ our evaluation provides a more comprehensive assessment of OOD behavior.
>
> $\textbf{2. Strict post-hoc tuning (no OOD exposure):}$ OpenOOD[2] uses held-out OOD validation data for hyper-parameter tuning. We follow the stricter and widely adopted OOD-free protocol from ReAct and DICE, tuning only on ID data with Gaussian noise. This avoids any form of OOD leakage and aligns with the assumptions of post-hoc detectors.
>
>
> $\textbf{W3.}$ We would like to clarify that DAVIS is not proposed as a new scoring function, but as a feature-enrichment module that restores statistical information (dominant activations, variance) discarded by global average pooling. Its primary purpose is to improve the separability of the feature representation on which existing OOD scoring functions operate, rather than competing with them as a standalone detector.
>
> Just as no single existing post-hoc method consistently dominates across all architectures and datasets, we agree that standalone DAVIS(m) or DAVIS(μ,σ) does not always exceed strong standalone baselines such as ASH or DICE (Tables 1-5). The central result of our work is the synergistic and systematic improvement achieved when DAVIS is paired with these baselines. For example, DAVIS+DICE and DAVIS+ASH yield consistent and substantial performance boosts across all architectures and datasets in our evaluation, demonstrating that DAVIS provides a stronger feature foundation for post-hoc scoring.
>
> Moreover, DAVIS exhibits markedly lower performance variability across architectures and datasets compared to standalone baselines, particularly in challenging activation regimes (e.g., SiLU-based EfficientNet-B0) where many post-hoc methods collapse. This stability underscores the robustness of the enriched statistical representation introduced by DAVIS.
>
> To directly address the reviewer’s concern, we will add averaged results across architectures to Tables 1 and 2 in the revised version. This will make the overall consistency and contribution of DAVIS clearer at a glance.
>
> $\textbf{W4.}$ To address the reviewer’s concern, we will place Figures 1 and 2 side-by-side in the final version, enabling clearer visual comparison and improved readability of the feature distributions.
>
> [1] ViM: Out-of-distribution with virtual-logit matching.
> [2]penood v1.5: Enhanced benchmark for out-of-distribution detection.

---

> > ### Author Response · Authors · 2025-12-02
> > **Additional Details: OpenImage-o Evaluations**
> >
> > For completeness, we also evaluated DAVIS on the OpenImage-O dataset and found that it consistently improves over the primary baselines reported in OpenOOD. Using pre-trained DenseNet-121, $DAVIS(\mu,\sigma)$ improves FPR95 by 29.13 % relative to Energy, and  $DAVIS(\mu,\sigma)+ASH$ further improves ASH by 7.68%. With pre-trained ResNet-50, $DAVIS(\mu,\sigma)$ improves the Energy baseline by 27.09%, and combining DAVIS with SCALE yields a 7.92% improvement. We observe similar trends for MobileNet-v2, and notably, DAVIS mitigates the catastrophic failure of existing methods on pre-trained EfficientNet-B0 by more than 50%. Taken together, these results -- along with our use of full, unaltered OOD datasets and strict OOD-free tuning -- indicate that our evaluation protocol both subsumes and is stricter than the OpenOOD v1.5 setting.

---

### Official Review · Reviewer_zDTq · 2025-10-19

**Soundness:** 3
**Presentation:** 3
**Contribution:** 1
**Rating:** 2
**Confidence:** 4

**Summary:**

This paper proposes DAVIS, a post-hoc enhancement framework for OOD detection. The authors argue that GAP discards discriminative statistical information and propose incorporating channel-wise maximum activations and variances into the features to improve the separability between ID and OOD data. Experiments show that this method significantly reduces the FPR95 on standard benchmarks like CIFAR and ImageNet and works synergistically with several leading post-hoc methods

**Strengths:**

- The authors have conducted exhaustive evaluations across a wide range of architectures and datasets.
- The paper provides an open-source implementation and details its hyperparameter search strategy.

**Weaknesses:**

I believe the main issue with this paper is its novelty. The motivation and the proposed method primarily stem from the distribution of activation values, particularly the distribution before the GAP layer. However, this has actually been proposed and explored in both [r1] and [r2]. I believe the core of this paper almost completely overlaps with these two 2024 papers. Additionally, there are in fact many other similar studies centered around activations; I have only listed the two most relevant ones here.

[r1] Tang, Keke, et al. "Cores: Convolutional response-based score for out-of-distribution detection." Proceedings of the IEEE/CVF Conference on Computer Vision and Pattern Recognition. 2024.
[r2] Wan, Weilin, et al. "Out-of-distribution detection using neural activation prior." arXiv preprint arXiv:2402.18162 (2024).

**Questions:**

please see weaknesses.

---

> ### Author Response · Authors · 2025-11-26
> **$\text{DAVIS}$ is a feature enhancement module that recovers channel variance and maximum activations lost by Global Average Pooling, providing an enriched feature vector ($\mathbf{f}'$) to existing OOD scores. This differs fundamentally from $\text{CORES}$ (a multi-layer, kernel-relevance-based scoring function) and $\text{NAP}$ (a multiplicative, Max/Avg-ratio scoring function).**
>
> $\textbf{Response to Weakness}$
>
> DAVIS fundamentally differs from both CORES and NAP in objective, representation, and mechanism. CORES introduces a new relevance-propagation–based scoring function, while NAP defines a new ratio-based OOD score. In contrast, DAVIS does not define a scoring rule at all, it is a lightweight, forward-only feature-enrichment module that restores statistical information discarded by GAP and is designed to be plugged into existing post-hoc detectors. Thus, DAVIS operates at a different stage of the pipeline and does not overlap with either method. We provide the detailed distinction below:
>
>
> $\textbf{CORES: }$  CORES and DAVIS differ fundamentally in motivation, representation, and mechanism. CORES does not analyze pre-pooled activation statistics, nor does it address information loss induced by global average pooling. Instead, CORES constructs a complex kernel-relevance -- based scoring function: it propagates the confidence score backward through the network to compute per-kernel relevance, selects the top-K and bottom-K kernels per layer, and aggregates these using layer-specific metrics such as Response Magnitudes (RM$^+$, RM$^-$) and Response Frequencies (RF$^+$, RF$^-$). The final CORES score depends on multiple engineered hyper-parameters (K, magnitude/frequency thresholds, and layer-fusion weights). Thus, CORES is a multi-stage relevance-propagation + kernel-selection + scoring pipeline.
>
> DAVIS is categorically different. It introduces no kernel selection, no backward propagation, and no new OOD scoring rule. DAVIS is a lightweight, forward-only feature enrichment module that recovers channel-level statistics -- maximum and variance -- that are discarded by GAP. These statistics augment the pre-logit feature vector directly and can be consumed by any post-hoc detector (Energy, ReAct, ASH, DICE, SCALE) without modifying classifier parameters or scoring mechanisms.
>
> Therefore, CORES and DAVIS operate on different objects (kernel relevance vs. channel statistics), rely on different computation paradigms (backward relevance propagation vs. forward statistical extraction), and serve different roles in the OOD pipeline (new scoring function vs. representation enhancement). CORES and DAVIS do not overlap; DAVIS is orthogonal, significantly simpler, and broadly compatible with existing post-hoc methods.
>
>
>
> $\textbf{NAP: }$ NAP proposes a standalone OOD score $S_{\text{NAP}}$ derived from a per-channel Max/Avg ratio, inspired by a signal-to-noise (SNR) analogy. This scalar is then combined multiplicatively (via geometric mean) with existing post-hoc detectors, resulting in a new final OOD scoring function that requires tuning of coupling hyper-parameters. Importantly, although NAP refers to this ratio as an activation prior,  the quantity is fully deterministic: for a fixed model and input, the activation map - and therefore its maximum and mean - are fixed values. As also noted by reviewers on the NAP OpenReview discussion, this ``prior” usage reflects a heuristic analogy rather than a probabilistic prior in the standard sense. In contrast, DAVIS does not rely on any prior analogy or ratio; it treats activation statistics according to their direct statistical meaning.
>
>
> DAVIS introduces no new scoring function. Instead, it provides a feature-level correction that restores discriminative information discarded by GAP. DAVIS extracts maximum ($m$), mean ($\mu$), and std($\sigma$) (with ablations on median and entropy) from each channel’s activation distribution and uses them to form an enriched penultimate feature vector $f'(\mathbf{x})$. This enhanced representation is then fed unchanged into existing post-hoc methods (Energy, ReAct, ASH, DICE, SCALE), preserving all classifier weights and score formulations. DAVIS is purely forward-only, requires no auxiliary priors, and performs no multiplicative score coupling.
>
> Methodologically, NAP relies solely on a Max/Avg ratio, which captures activation intensity but not dispersion or reliability within a channel. DAVIS explicitly models variance, which our experiments show to be essential for ID–OOD separation once GAP has collapsed spatial information. NAP does not incorporate variance or other distributional descriptors, and therefore cannot target the specific failure mode—information loss induced by pooling - that DAVIS directly addresses.
>
>
> In summary, NAP defines a ratio-based scoring function, whereas DAVIS provides a statistical feature enhancement module that augments the representation used by existing post-hoc detectors. They differ in objective (scoring vs. feature correction), representation (SNR-style ratio vs. descriptive statistics), and mechanics (multiplicative fusion vs. forward-only concatenation). DAVIS is orthogonal to NAP and does not overlap with it.

---

> > ### Comment · Reviewer_zDTq · 2025-11-28
> >
> > Thanks for the rebuttal.
> > While I appreciate the detailed differences highlighted between CORES, NAP, and DAVIS, I still believe that the assertion of "fundamental differences" made by the authors at the beginning of their response does not hold as strongly as they suggest. I would like to elaborate on this point by point:
> >
> > - CORES vs. DAVIS: The authors mention that CORES introduces no kernel selection, no backward propagation, and no new OOD scoring rule. In my perspective, this distinction may not be as significant. Both CORES and DAVIS seek to capture the response magnitudes and frequencies of activation values, albeit from different angles. CORES aims to extend this analysis across the entire network, which inherently includes aspects of GAP that DAVIS concentrates on. Thus, I view CORES as an "engineering-extended version" of DAVIS, incorporating additional design elements such as kernel selection and layer weight adjustments. Consequently, I believe it may not be entirely appropriate to criticize CORES for its complexity; rather, both methodologies fundamentally focus on the distribution of activation values. While it's true that complexity is not always beneficial and involves trade-offs, this further solidifies my view that CORES and DAVIS should not be regarded as fundamentally different.
> >
> > - Scoring Rules and OOD Pipeline: The authors argue that DAVIS plays a different role in the OOD pipeline by not establishing a new scoring rule. However, this distinction does not create significant barriers. It's entirely feasible for DAVIS to utilize its statistical values as an OOD score, similar to NAP. Moreover, CORES could potentially integrate its scores into activation values as a form of representation enhancement. Such adaptations are quite common in current OOD approaches and often involve only minor code changes, which does not necessarily indicate a fundamental difference between the methods. I suggest that you can explore these potential synergies, like integrating DAVIS scores into CORES or enhancing CORES with DAVIS scores, to demonstrate the compatibility and potential benefits of combining the two methodologies.
> >
> > - Orthogonality Claims: The authors assert that CORES and DAVIS are orthogonal based on their differing roles in the pipeline. I contend that orthogonality is determined more by the underlying features leveraged by the methods than by the introduction of a new scoring rule. For instance, if one method captures a feature \(a\) for OOD detection and another method integrates this feature into a scoring function, the performance enhancement will depend on the relevance of the scoring function selected. Given that the features utilized by DAVIS have been explored in existing works, including discussions surrounding CORES and NAP, it’s challenging to definitively claim that DAVIS is orthogonal to these works solely based on its lack of a new scoring function.

---

> > ### Comment · Reviewer_zDTq · 2025-11-28
> >
> > - Heuristic Analogy vs. Feature Utilization: The authors make a distinction where they assert that NAP relies on a heuristic analogy while DAVIS does not. However, I see this as a stylistic difference in writing rather than a substantive one.
> >
> > - New Scoring Function: The authors noted that DAVIS does not introduce a new scoring function but provides feature-level correction. As I mentioned in my discussion on CORES, this is actually just a different usage of the same features. It is evident that both DAVIS and NAP capture similar activation features, with one being directly used for scoring, and the other being integrated into activation values.
> >
> > - Focus on GAP: The authors emphasize that DAVIS’s contribution lies in recovering discriminative information that was discarded by GAP. However, this point is also highlighted in the abstract of the NAP. Here is a direct excerpt from its abstract:
> > > (arXiv:2402.18162v4) Notice that previous methods primarily rely on post-global-pooling features of the neural networks, while the within-channel distribution information we leverage would be discarded by the global pooling operator.
> > I believe this sufficiently demonstrates that the contribution point emphasized in DAVIS has actually been acknowledged in NAP. Additionally, CORES has implicitly focused on this as well, though they did not emphasize it in their paper.
> >
> > - Modeling Variance: The authors claim that NAP does not account for variance, while DAVIS does. However, NAP discusses variance in its work, as shown in Table 6 of the withdrawn version of ICLR from last year. Consequently, both DAVIS and NAP converge on similar goals regarding GAP, using consistent statistical measures (including maximum, mean, and variance). The differences between them largely lie in engineering choices, which may not be sufficient to establish the novelty of DAVIS.
> >
> > In summary, DAVIS overlaps with CORES and NAP in various important aspects, even being covered by them; Unfortunately, despite the authors' thorough rebuttal, based on the points mentioned above, I still do not consider DAVIS to be a novel contribution in 2025. However, I acknowledge that the authors have made contributions in areas of unexpected innovation, and therefore, I will raise my rating to 4.
> >
> > I suggest that the authors could consider submitting their work to a journal like TMLR, which emphasizes solid contributions, rather than to ICLR, which tends to focus more on novelty.

---

> > > ### Author Response · Authors · 2025-12-02
> > >
> > > We thank the reviewer for their thoughtful follow-up and for engaging deeply with the distinctions we outlined. We agree that many OOD methods operate on activation patterns at some level, the novelty of DAVIS lies not only in what information it accesses, but also how it operates. DAVIS is intentionally designed as a simple forward-only feature-enrichment module that restores statistical information (maximum, variance) that is explicitly lost due to global average pooling. This simplicity is central to its contribution: without introducing a new scoring rule, relevance propagation, or layer-wise kernel selection, DAVIS improves the underlying representation on which existing scoring functions operate, and consistently strengthens strong post-hoc detectors (ASH, DICE, ReAct, SCALE) across diverse architectures (ResNet, DenseNet, MobileNet, EfficientNet) and activation functions (ReLU, SiLU, GELU, Mish). We therefore view DAVIS as complementary -- rather than overlapping -- with scoring-based approaches such as CORES and NAP, which define new decision functions and rely on multi-stage pipelines.
> > >
> > > Regarding the reviewer’s suggestion that CORES may be viewed as an “engineering extension” of DAVIS, we respectfully clarify that the two differ fundamentally in mechanism: CORES performs relevance tracing via a backward pass with kernel-level selection, whereas DAVIS relies solely on forward statistical extraction at the final activation layer. We attempted to explore integrating DAVIS with CORES during the rebuttal, but no publicly available code or sufficient implementation detail exists to enable a faithful reproduction (CORES involves multiple relevance-propagation components).

---

### Official Review · Reviewer_FzsQ · 2025-10-29

**Soundness:** 3
**Presentation:** 3
**Contribution:** 2
**Rating:** 6
**Confidence:** 5

**Summary:**

This work identifies a key limitation in standard post-hoc OOD detection: the information loss from relying solely on Global Average Pooling (GAP), which discards valuable statistical cues like channel-wise variance and maximum activations. The authors propose DAVIS, a simple and effective method that augments the standard feature vector with these discriminative statistics, leading to a more powerful representation for separating in-distribution from out-of-distribution data. Extensive experiments demonstrate that DAVIS establishes a new state-of-the-art across diverse model architectures, significantly reducing the FPR95, and provides a principled basis for moving beyond mean-based features.

**Strengths:**

1. The paper compellingly identifies a critical, previously overlooked flaw in standard post-hoc OOD detection: the significant information loss inherent in relying solely on Global Average Pooling (GAP), which discards valuable spatial distribution statistics like variance and maximum activations. The proposed DAVIS method directly and effectively addresses this core limitation by enriching features with these discriminative statistics.
2. DAVIS is presented as a remarkably simple, plug-and-play technique that seamlessly integrates with existing pipelines and diverse architectures (ResNet, DenseNet, MobileNet-V2, EfficientNet). Its strength lies in its synergistic ability to significantly boost the performance of standard baselines without requiring architectural changes or retraining.
3. The paper provides extensive and convincing experimental evidence, demonstrating that DAVIS establishes a new state-of-the-art across multiple challenging benchmarks (CIFAR-10, CIFAR-100, ImageNet-1k).  The analysis offers valuable insights into the mechanism behind the improvement.

**Weaknesses:**

1. Although the reported results demonstrate the effectiveness of the method, comparisons with some highly relevant approaches are missing, such as BATS [1] and LAPS [2].
2. The method is predicated on models that utilize Global Average Pooling (GAP). It may not be applicable to models that do not employ GAP.
3. According to Table 1, the results of DAVIS(m) alone are not as good as ASH, and it is only when combined with DICE that DAVIS(m) surpasses ASH. This suggests that the performance improvement of DAVIS(m) by itself is limited.

[1] Zhu, Yao, et al. "Boosting out-of-distribution detection with typical features." Advances in Neural Information Processing Systems 35 (2022): 20758-20769.
[2] He, Rundong, et al. "Exploring channel-aware typical features for out-of-distribution detection." Proceedings of the AAAI conference on artificial intelligence. Vol. 38. No. 11. 2024.

**Questions:**

The paper only reports the combinations of the proposed DAVIS with ASH and DICE. How about the combination effects of DAVIS with NCI and fdbd?

---

> ### Author Response · Authors · 2025-11-26
>
> $\textbf{Questions}$
>
> $\textbf{Q1.}$ The paper only reports the combinations of the proposed DAVIS with ASH and DICE. How about the combination effects of DAVIS with NCI and fdbd?
>
> We demonstrate complementary synergy of DAVIS with 8 major post-hoc baselines (MSP, ODIN, Energy, ReAct, DICE, ReAct+DICE, ASH, SCALE), as detailed in Table 11-12 (Appendix D) and Table 15-20. Notably, compared to most existing methods, we spans a broader set of models: DenseNet-121 and EfficientNet-B0 on ImageNet, and ResNet-18/34 and MobileNet-v2 on CIFAR, providing rigorous assessment of cross-architecture behavior.
>
> DAVIS is, by design, compatible with any method that consumes the penultimate features, and this includes NCI and fDBD. However, both methods involve multi-stage geometric pipelines (e.g., class-center deviations, weight-space regularization, and score normalization), and integrating their code base into DAVIS is a substantial engineering effort. Performing this integration across all architectures and datasets in our study was outside the scope of this work. For fairness, we therefore report baseline comparisons to NCI and fDBD using the architectures and datasets from their original publications (Section 4.3 and Appendix D). Extending DAVIS to these geometric scoring functions is a promising direction, and we will consider this in future work.
>
> $\textbf{Weaknesses}$
>
> $\textbf{W1.}$  We appreciate the reviewer highlighting BATS and LAPS. Both are indeed relevant post-hoc OOD detectors. Our initial selection of primary baselines -- Energy, ReAct, DICE, ASH, and SCALE -- was guided by their superior and competitive performance, ensuring we benchmark against the most challenging competitors. Due to space and computational constraints, we prioritized these widely-adopted detectors for inclusion in the main paper.
>
> Importantly, DAVIS is a feature-enrichment module that augments the penultimate representation by recovering channel-wise statistics (max and variance) discarded by GAP. In contrast, BATS and LAPS are feature-rectification methods that operate directly on the existing features. These methods are therefore complementary to DAVIS rather than competing alternatives. Because DAVIS strengthens the feature vector itself, it can be applied before BATS or LAPS, and we expect these methods to benefit from the richer feature representation. We acknowledge the oversight of not including BATS and LAPS in the original submission and will add comparisons under our evaluation setting in the camera-ready version, space permitting.
>
> $\textbf{W2.}$  As discussed in Section 5, we agree that DAVIS is specifically designed for architectures employing Global Average Pooling (GAP). This is the limitation of existing version of DAVIS and technical focus on solving a fundamental feature bottleneck present in the vast majority of modern CNNs (ResNet, DenseNet, MobileNet, EfficientNet). Our core contribution is precisely based on recovering critical statistical information (variance and dominant activations) that the GAP operation intrinsically discards. We explicitly discuss this architectural scope and the extension to non-GAP models (such as ViTs, which use [CLS] tokens) in Section 5: Limitation and Future Work, where we frame it as a necessary avenue for future research.
>
>
> $\textbf{W3.}$ We thank the reviewer for this observation. We emphasize that DAVIS is not introduced as a new scoring function, but as a feature enrichment mechanism that restores critical statistical signals (maximum activation and variance) that are lost due to global average pooling. Thus, the primary purpose of DAVIS is to strengthen the feature space on which downstream scoring functions operate -- not to replace those scoring functions.
>
> While DAVIS  alone (i.e., using the standard Energy score on DAVIS features) does not always exceed strong standalone baselines such as ASH, our central finding is the consistent and substantial synergistic gains produced when DAVIS features are paired with existing methods. Combinations such as DICE+DAVIS  and ASH+DAVIS  yield the largest and most systematic improvements across architectures and datasets, demonstrating that enriched DAVIS features form a superior input representation for post-hoc scoring.
>
> Importantly, even the simplest instantiation -- Energy applied on DAVIS features -- already shows clear improvements over GAP-based Energy, confirming that the recovered statistics (max, variance) are intrinsically informative. This effect is especially pronounced on architectures such as EfficientNet-B0, where established post-hoc methods (DICE, ASH, SCALE) collapse due to dense activations, whereas DAVIS remains stable and continues to outperform them, even under the default scoring rule.
>
> Overall, the evidence indicates that the strength of DAVIS lies in enhancing the representation, and the most meaningful gains arises, by design, when this enriched representation is combined with existing post-hoc OOD detectors.

---

### Official Review · Reviewer_4SVi · 2025-11-01

**Soundness:** 2
**Presentation:** 2
**Contribution:** 2
**Rating:** 4
**Confidence:** 4

**Summary:**

This paper proposes DAVIS, a simple post-hoc OOD detection method that augments penultimate features with channel-wise maximum activations and variance to recover information lost in global average pooling.
It works plug-and-play with existing methods (Energy, DICE, etc.).

**Strengths:**

- Clear motivation and solid intuition: The paper convincingly identifies information loss in GAP and proposes a simple yet effective fix by incorporating dominant activations and variance.

- Strong empirical results: Consistent and substantial improvements (up to 48% FPR95 reduction) across reported multiple architectures and datasets.

**Weaknesses:**

- Dependence on pretrained classifier head:
The method reuses the original classifier weights for modified features; though empirically justified, this may not always be theoretically optimal.
Some adaptive fine-tuning or learned re-weighting could further stabilize the approach.

- Lack of a unified variant combining both signals:
The paper defines two separate DAVIS variants — DAVIS(m) using channel-wise maxima and DAVIS(µ,σ) incorporating mean plus variance — but does not explore combining both dominant and variance cues jointly.
A unified formulation leveraging both peak intensity and distribution spread could potentially yield further improvement.

- Limited coverage of transformer-based models:
The authors acknowledge DAVIS currently applies only to architectures with spatial aggregation (CNNs).
Extending to ViTs or hybrid architectures would broaden applicability.
Regardless of outcome, such experiments could offer important insights into how attention models encode distributional statistics for OOD detection.

- Hyperparameter tuning validation datasets:
DAVIS tunes γ only on Gaussian-noise validation sets and separately for each dataset and architecture.
No tests on other perturbations or cross-dataset stability are shown, so the claimed robustness is likely distribution-specific rather than general.

- More evaluation setting:
The experiments in DAVIS use dataset splits, metrics, and model sources different from the standardized OpenOOD benchmark.
Evaluating DAVIS under a unified OpenOOD setting would strengthen reproducibility and clarify whether it still outperforms recent strong SOTAs such as AdaSCALE or CombOOD.

**Questions:**

- Could you comment on whether re-training or lightly fine-tuning the classifier head with DAVIS features would further improve separability?

- Would combining DAVIS with logit-based regularization (e.g., Outlier Exposure) yield additive gains, or do the effects saturate?

- Have you tested whether the choice of activation function (beyond SiLU/ReLU) consistently affects the relative importance of maximum vs. variance statistics?

- Could you clarify whether DAVIS(m) and DAVIS(μ,σ) are complementary or mutually exclusive in practice—e.g., has concatenation been tested?

---

> ### Author Response · Authors · 2025-11-26
>
> $\textbf{Questions}$
>
> $\textbf{Q1.}$ Our method is deliberately framed as a strictly post-hoc approach, motivated by scenarios where the backbone is frozen or training data is unavailable. DAVIS is therefore designed to operate without any parameter updates.  With respect to separability, fine-tuning the classifier head using the DAVIS-augmented features would likely provide additional improvements, as the classifier could learn weights that explicitly exploit the added statistical components. However, doing so would move the method outside the post-hoc setting and into a training-based regime. Our focus in this work is to demonstrate that substantial gains are achievable without retraining. We will note this training-based extension as a valuable direction for future work.
>
> $\textbf{Q2.}$ Outlier Exposure (OE) is a training-time regularization technique that modifies the classifier using auxiliary outlier data, whereas DAVIS operates strictly in the post-hoc setting without any training or auxiliary datasets. Because the two methods intervene at different stages of the pipeline, they are compatible in principle, and we expect that OE-trained models would still benefit from DAVIS, as DAVIS enhances the representation used by post-hoc scoring methods regardless of how the classifier was trained. However, studying such combinations lies outside the scope of our work, which focuses specifically on the zero-training post-hoc regime. We will add this as a promising direction for future work, particularly for understanding whether OE’s training-time regularization and DAVIS’s feature-level corrections yield additive or saturating effects.
>
> $\textbf{Q3.}$ Beyond the ReLU and SiLU results reported in the submission, we also evaluated the GELU and Mish activation on ResNet-18 (CIFAR-10/100). The qualitative trend remained consistent -- we witnessed the similar relative importance of maximum vs. variance statistics. In the main paper, we restricted our analysis to ReLU and SiLU because they represent the two dominant activation families in modern CNNs: ReLU-based in classical architectures (ResNet, DenseNet, MobileNet) and SiLU/GELU in recent models (EfficientNet, ConvNeXt). We appreciate the reviewer’s suggestion, and we would be happy to include an additional activation function if one is deemed critical for validation.
>
> $\textbf{Q4.}$ This is a valuable suggestion. We explicitly tested a unified variant using all three signals: [Mean, Variance, Max]. However, our empirical analysis found that this combination yielded negligible performance gains compared to the simpler $DAVIS(m)$ or $DAVIS(\mu, \sigma)$ variants. Therefore, we advocate for the simpler variants, which provide the similar performance. For clarity, we will include this observation in the final version.
>
> $\textbf{Weaknesses}$
>
> $\textbf{W1, W2.}$ Answered in Q1 and Q4 respectively.
>
> $\textbf{W3.}$ As discussed in Section 5 (Scope & Architectural Focus, Limitations & Future Work), DAVIS is scoped to CNNs because its technical contribution specifically targets the information loss introduced by GAP -- the dominant aggregation mechanism in ResNet, DenseNet, EfficientNet, and MobileNet. ViTs use [CLS] token aggregation, requiring non-trivial reformulation (e.g., token-level statistics). Extending DAVIS to ViTs and hybrid models is an important and promising future direction, and we have explicitly noted this in the paper.
>
> $\textbf{W4.}$ We follow the OOD-free tuning protocol from ReAct/DICE, using Gaussian-noised ID samples -- this avoids OOD leakage and is standard for post-hoc methods. Additional perturbations are outside our scope. Cross-dataset transfer (e.g., ImageNet $\gamma$ on CIFAR) still improves over baselines, though dataset-specific $\gamma$ remains optimal.
>
> $\textbf{W5.}$ We also evaluated on the OpenImage-O dataset; DAVIS improves primary baselines (e.g, using DenseNet-121: $DAVIS(\mu,\sigma)$ improves FPR95 by 29.13%, and $DAVIS(\mu,\sigma)+ASH$ by 7.68%). We argue, our protocol is stricter than OpenOOD v1.5 because:
> (1) Full, unaltered datasets: We use canonical OOD benchmarks (SUN, Places, iSUN, LSUN, iNaturalist, full Texture set) omitted in OpenOOD[1,2], giving a more comprehensive assessment.
> (2) Strict post-hoc tuning: OpenOOD allows held-out OOD validation. We follow ReAct/DICE’s OOD-free protocol, tuning only on ID data with Gaussian noise to prevent OOD leakage.
>
> AdaScale and CombOOD comparison.
> On DenseNet-101, DAVIS(m) reduces average FPR95 by 10.47 points over AdaScale-L (40.03→29.56), and DAVIS+DICE achieves 14.97 FPR95 (25.06-point improvement). On CIFAR-100, DAVIS+DICE improves FPR95 by 11.66 points (59.80→48.14). Although CombOOD does not report FPR95, on AUROC (ResNet-50, ImageNet-1k), DAVIS+ASH reaches 97.94%, exceeding CombOOD’s 92.06% on the overlapping Texture and iNaturalist datasets.
>
> [1] ViM: Out-of-distribution with virtual-logit matching.
> [2]penood v1.5: Enhanced benchmark for out-of-distribution detection.

---

> ### Author Response · Authors · 2025-12-02
> **Additional Details: OpenImage-o Evaluations**
>
> For completeness, we also evaluated DAVIS on the OpenImage-O dataset and found that it consistently improves over the primary baselines reported in OpenOOD. Using pre-trained DenseNet-121, $DAVIS(\mu,\sigma)$ improves FPR95 by 29.13 % relative to Energy, and  $DAVIS(\mu,\sigma)+ASH$ further improves ASH by 7.68%. With pre-trained ResNet-50, $DAVIS(\mu,\sigma)$ improves the Energy baseline by 27.09%, and combining DAVIS with SCALE yields a 7.92% improvement. We observe similar trends for MobileNet-v2, and notably, DAVIS mitigates the catastrophic failure of existing methods on pre-trained EfficientNet-B0 by more than 50%. Taken together, these results -- along with our use of full, unaltered OOD datasets and strict OOD-free tuning -- indicate that our evaluation protocol both subsumes and is stricter than the OpenOOD v1.5 setting.

---

### Meta-Review · Area_Chair_tddN · 2025-12-08

**Summary:**

This paper examines the information loss from the commonly-used global average pooling practice in OOD detection, which can be a plug-and-play module for improving OOD detection. The main concerns of reviewers mainly includes:

1.Lack of commonly used benchmarks, such as OpenOOD

2.Need for full dataset usage, which is stricter than standard benchmarks.

3.Evaluation protocol alignment with common practice

4.Hyper-parameter robustness

5.Overlap with existing methods (question about novelty)

**Reviewer Concerns:**

Reviewer concerns addressed:
1.Lack of commonly used benchmarks, such as OpenOOD
2..Need for full dataset usage, which is stricter than standard benchmarks.
3.Hyper-parameter robustness

Reviewer concerns outstanding:
1.Evaluation protocol alignment with common practice
2.Overlap with existing methods (question about novelty)

**Reviewer Scores:**

The original scores are 4,6,2,2. No reviewer claims to change the scores during the discussion.

---

> ### Public Comment · ~Abid_Hassan1 · 2026-03-09
> **Reviewer zDTq updated the Score to 4**
>
> Reviewer zDTq acutally updated the score to 4 in the text.

---

### Decision · Program_Chairs · 2026-01-26

Reject